# Characterising the interaction of tropical and extratropical air masses controlling East Asian summer monsoon progression using a novel frontal detection approach

Ambrogio Volonté[1,2], Andrew G. Turner[1,2], Reinhard Schiemann[1,2], Pier Luigi Vidale[1,2], and Nicholas P. Klingaman[1,2]

[1]Department of Meteorology, University of Reading, Reading, RG6 6ET, UK
[2]National Centre for Atmospheric Science, University of Reading, Reading, RG6 6ET, UK

**Correspondence:** Ambrogio Volonté (a.volonte@reading.ac.uk)

**Abstract.** The East Asian summer monsoon (EASM) is a complex phenomenon, influenced by both tropical and midlatitude dynamics and by the presence of the Tibetan Plateau. The EASM front (EASMF) separates tropical and extratropical air masses as the monsoon marches northwards. Although the different factors behind EASM progression are illustrated in a number of studies, their interactions, in particular between tropical and extratropical air masses, still need to be clarified. In this study we apply Eulerian and Lagrangian methods to the ERA5 reanalysis dataset to provide a comprehensive study of the seasonal progression and interannual variability of the EASM, and we highlight the dynamics of the air masses converging at its front.

A frontal detection algorithm is used to perform a front-centred analysis of EASM progression. The analysis highlights the primary role of the subtropical westerly jet (STWJ) and of the Western North Pacific subtropical high (WNPSH) in controlling the strength and the poleward progression of the EASMF, in particular during *Mei Yu*, the primary stage of EASM progression. These forcings act to steer the southerly advection of low-level moist tropical air, modulated by the seasonal cycle of the Asian monsoon. The *Mei Yu* stage is distinguished by an especially clear interaction between tropical and extratropical air masses converging at the EASMF. The analysis of composites based on the latitude of the EASMF during *Mei Yu* reveals the influence exerted by the STWJ on the cool extratropical flow impacting on the northern side of the EASMF, whose progression is also dependent on the location of the WNPSH. In turn, this affects the extent of the warm moist advection on its southern side and the distribution and intensity of resultant rainfall over China.

This study shows the validity of an analysis of EASM progression focused on its front and on the related low- and mid-level airstreams, at least in the *Mei Yu* stage. The framework highlighted shows how the regional flow over East Asia drives the low-level airstreams that converge at the EASMF, thus controlling the shape of EASM progression. This framework provides a basis for studies of climate variability and extreme events and for model evaluation.

## 1 Introduction

Most of the annual precipitation over China occurs during the East Asian summer monsoon (EASM). The EASM is (like its South Asian counterpart) driven by the differential heating between the Indian and Pacific Oceans and the continental land-

mass, in addition to the seasonal meridional march of maximum solar radiation which governs the global monsoon evolution. The differential heating is enhanced by the presence of a large elevated terrain, the Tibetan Plateau (TP hereafter). Further complexity is caused by the location and broad latitudinal span of East Asia, which causes the EASM to be affected by both tropics and mid latitudes (Webster et al., 1998; Wu et al., 2012), making its seasonal cycle an inherently multi-scale and multi-stage phenomenon. As explained in Chiang et al. (2020), the key to the unique seasonality of EASM rainfall lies in the interaction between two distinct circulations, of tropical and extra-tropical origin, which are jointly driven by tropical and midlatitude dynamics, coupled with the presence of the TP.

The seasonal migration of the rainfall over China can be divided in four distinct stages (*spring*, *pre-Mei Yu*, *Mei Yu*, and *mid-summer*) characterised by quasi-stationary features and abrupt transitions (Chiang et al., 2017; Kong et al., 2017; Wang and LinHo, 2002). The *spring* rains see persistent rainfall over southern China and can be considered a pre-monsoon stage, as it is rather a consequence of the effect of the TP on the regional winter and early spring circulation (Wu et al., 2007). The following *pre-Mei Yu* stage occurs at the same time as the onset of the South Asian monsoon, with convective rainfall surging poleward from the South China Sea towards inland China. The *Mei Yu* stage (*Changma* in Korea and *Baiu* in Japan) starts by mid-June and ends by mid-July, consisting primarily of a substantial northward shift of the rainfall, at this point organised by large-scale frontal convergence in a quasi-stationary rain belt extending from central eastern China to Japan. It is during *Mei Yu* that the East Asian summer monsoon front (EASMF), separating extratropical and tropical air masses throughout the seasonal monsoon progression and characterised by sharp gradients in equivalent potential temperature, becomes more localised and clear. After *Mei Yu*, the rainfall, now mainly caused by local convection, jumps further north for the *mid-summer* stage, which lasts for about a month before the start of the southward retreat (Yihui, 1992; Yihui and Chan, 2005; Kong and Chiang, 2020; Chiang et al., 2020).

The EASM can be seen, at least partially, as a tropical phenomenon, as in general the eastern component of the Asian monsoon is characterised by a multi-stage northward progression starting from tropical latitudes, sustained by low-level southerly and southwesterly winds. During boreal spring the Asian monsoon extends northeastward first over the Indochina peninsula, then over the South China Sea in May, and then, at the end of the month, into the subtropical western North Pacific Ocean, starting the pre-Mei Yu stage of the EASM (Wang and LinHo, 2002). In addition to the southerlies and southwesterlies from South Asia, the poleward transport of moisture over the South China Sea is associated with the Western North Pacific subtropical high (WNPSH). The northward monsoonal flow thus lies between an oceanic high to the east and a monsoonal low to the west, a well-known wave response to subtropical monsoon heating occurring at the eastern edge of a continent (Rodwell and Hoskins, 2001). The WNPSH influences the EASM throughout its progression as: (i) spring rains coincide with the first northward jump of the WNPSH over the Pacific; The onset of the Mei Yu phase coincides with further northward advance of the WNPSH and (iii) Westward extension of the WNPSH helps bring about the end of the Mei Yu phase by replacing the south-westerly flow with that from the southeast (Yihui and Chan, 2005). The location and strength of the WNPSH is therefore a key driver of variability for the EASM, along with the progression of the Asian monsoon and the main modes of tropical variability, such as ENSO (El Niño-Southern Oscillation, Yihui (2004); Chiang et al. (2020)) or convection over western tropical Pacific with the related Pacific-Japan pattern (Nitta, 1986).

The midlatitude influence on the EASM is mainly caused by the subtropical westerly jet (STWJ) and its interaction with the TP. Springtime diabatic heating processes, such as increased heat fluxes over the TP, can reinforce the land-sea thermal contrast and are instrumental in the northward migration of the STWJ. Given the height of the TP, these diabatic processes are able to trigger a reversal of meridional temperature gradient in the upper troposphere between the TP and the Indian Ocean. As a result, they act as drivers of the regional atmospheric circulation and of the EASM onset (Li and Yanai, 1996). However, Chen and Bordoni (2014) show that the influence of the TP is not limited to thermodynamics. Since the extent of the TP and its latitudinal span cause it to interact with the northward migration of the STWJ, dynamic processes are also of primary importance. The latitudinal displacement of the STWJ relative to the TP is thus crucial in the variability of the EASM in all the stages of its progression, as pointed out by Schiemann et al. (2009) and confirmed by Chiang et al. (2017) and Kong et al. (2017). The northward migration of the westerly jet onto the TP corresponds to the onset of *pre-Mei Yu* in southeastern China, while *Mei Yu* begins over central eastern China when the STWJ reaches the northern edge of the TP.

The meridional position of the westerly jet relative to the TP is particularly key for the progression of the EASMF during *Mei Yu*, as stressed by Sampe and Xie (2010). During this stage the STWJ favours advection of warm and moist air from the southeastern flank of the TP, where surface heating is enhanced, towards the EASMF. This advection favours ascent and convection, sustaining the EASMF. The effect of westerlies impinging upon the TP during *Mei Yu* on the downstream circulation is highlighted in Molnar et al. (2010). The modified flow displays an area of moisture convergence in the lee of the TP, where the rainfall occurs. This rain band persists as long as the mechanical forcing of the TP on the STWJ continues. It is when the STWJ moves further north, well away from the TP, that the EASM rapidly weakens and *mid-summer* rainfall dominates northeastern China. The termination of the *Mei Yu* stage is investigated in more detail in Kong and Chiang (2020). Their results show that the EASMF and associated rain band weaken when the axis of the STWJ moves beyond 40°N, leaving the northern edge of the TP. The demise of the front is caused by the weakening of the meridional moisture contrast and meridional wind convergence, which is a consequence of the northward migration of the STWJ switching off the orographic forcing and the associated downstream cyclonic circulation. Differences in the timing and duration of the *Mei Yu* stage are closely linked to the rainfall accumulation observed over eastern China. Chiang et al. (2017) show that an earlier *Mei Yu* termination, and associated transition of the STWJ off the northern edge of the TP, results in the appearance of a "tripole" pattern in rainfall, with lower values over central eastern China and increased amounts over northeastern and southeastern China. It should be noted that the "tripole" mode was originally referring to a leading mode of interannual variability of summer rainfall over East Asia (Hsu and Lin, 2007).

The importance of STWJ latitude in relation to the TP in EASM seasonal progression suggests the possible influence of large-scale modes of variability of the midlatitude flow. Yihui and Chan (2005) explain that the midlatitude influence on EASM progression takes the shape of successive southward cold-air intrusions that provide the temperature gradient sustaining the EASMF. These intrusions are excited by the development and evolution of blocking anticyclones at mid- and high-latitudes over Eurasia, with a dual blocking configuration (over the Ural Mountains and Okhotsk Sea) being the most favourable for long-lasting and heavy Mei Yu rainfall. Internal variability in blocking highs, and in the location of the WNPSH, is thus of primary importance for the progression of the EASM. The influence of eastward-propagating wave trains, such as the Silk

Road pattern trapped along the North African and Eurasian jet streams (Lu et al., 2002), is also apparent. Hong and Lu (2016) outline the correlation between latitudinal STWJ displacements and cyclonic anomalies over East Asia, with Hong et al. (2018) pointing out that this correlation is more robust when the STWJ displacement is poleward.

The studies mentioned thus far illustrate the various factors behind the complexity of EASM progression, influenced by both tropical and midlatitude dynamics, and by the interaction with the TP. However, much still needs to be understood as to how those different dynamical mechanisms interact and lead to the observed complex and multi-stage progression that characterises the EASM. Recent work in the Indian component of the Asian monsoon has suggested that new approaches could be used. Parker et al. (2016) have shown that the northwestward progression of the Indian monsoon onset is controlled by the competition between southwesterly moist tropical flow and a dry intrusion of extratropical air advected from northwest. This dry wedge-shaped layer is moistened from below by shallow convection which gradually reduces its depth, until it becomes shallow enough to allow deep convection to form. In this way, as the advection of moisture from the Arabian Sea becomes stronger and the mid-level dry advection weakens, the monsoon onset advances towards northwest India. Volonté et al. (2020) further characterised Indian summer monsoon progression, by revealing its non-steady nature. They showed that the balance of the moist-flow vs dry-intrusion interaction is closely linked to midlatitude dynamics, e.g. through the southward propagation of potential vorticity streamers and the associated formation of cyclonic circulations in the region where the two air masses interact. Thus, they highlighted that the pace and steadiness of Indian monsoon progression is dependent on the interaction of different air masses (monsoonal vs subtropical) and influenced by a synergy of factors at different scales.

Thus, the approach of Parker et al. (2016), and then Volonté et al. (2020), includes the focus on frontal progression and on the location and evolution of the 3-dimensional boundary between different air masses, and also the use of Lagrangian trajectories to contrast the tropical and mid-latitude origin of air at different stages of monsoon onset. Drawing from this approach and adapting it to the specific properties of the EASM (such as its multi-stage nature), in this study we focus on the role of tropical and extratropical air masses in EASM progression. By doing so, we assess if a front- and airmass-centred Parker et al. (2016)-like framework can be used to outline the key dynamics driving the EASM, clarifying the interactions between the air masses involved. The paper is structured as follows. The description of the reanalysis dataset used for the analysis is contained in Section 2, along with the algorithm of frontal detection developed for the study and any other analysis tools. Results start from an EASM-front-centred perspective in Section 3, focusing on the migration and evolution of the EASMF, and exploring its link with the key features of EASM progression identified through the analysis. Although the main focus is the front over China, we still refer to it as the *EASMF* instead of *Mei Yu front*, since we are considering all stages of EASM progression (other than the final retreat). After taking advantage of the front-centred analysis to outline the key dynamics behind EASM progression, in Section 4 we focus on the dynamics of the interaction between the two airstreams (of tropical and extratropical origin) converging at the frontal region. We use Eulerian and Lagrangian tools to reveal the processes governing their interaction. Analysis is then complemented by assessment of the main factors accounting for variability in EASM progression in in Section 5. We conclude in Section 6.

## 2 Data and Methodology

### 2.1 ERA5

The results of this study are obtained using the ERA5 reanalysis dataset (Hersbach et al., 2020). The data used consist of 6-hourly atmospheric fields (plus hourly precipitation) on a $0.25° \times 0.25°$ horizontal grid and 22 vertical levels between 1000 and 200 hPa for April-July 1979-2018. ERA5 was chosen as the preferred dataset for this study as it allows us to construct 40-year climatologies of quantities of interest over a large domain, necessary to capture the different drivers of EASM progression, and at a finer resolution than its predecessor ERA-Interim (Dee et al., 2011), an advantage in resolving smaller-scale EASMF dynamics. Compared to ERA-Interim, ERA5 has a more advanced assimilation system and includes more sources of data. Early evidence shows that these features lead to better quality of precipitation data. A significant improvement is indeed observed in the representation of deep convection and moisture flux convergence in the tropics, leading to a reduced precipitation bias, as pointed out in Nogueira (2020). Their study contains a global evaluation of ERA5 and ERA-Interim against the Global Precipitation Climatology Project (GPCP) dataset (Adler et al., 2018), showing that over the tropics ERA5 displays lower bias and unbiased root-mean squared error and higher correlations. Focusing the attention on China, Jiao et al. (2021) compare ERA5 precipitation data against a gridded dataset based on observations from more than 2400 stations. Their results highlight the suitability of ERA5 precipitation data for studies in the region, showing that it captures well the annual and seasonal patterns of precipitation, while slightly overestimating summer rainfall and showing larger uncertainty over high-altitude areas.

### 2.2 Front detection algorithm

#### 2.2.1 Front detection algorithms in literature

Several methodologies have been proposed in the literature to identify fronts in different seasons and regions. Thomas and Schultz (2019a, b) provide a comprehensive review of front climatologies based on either manual analyses of synoptic charts or gridded data, and discuss the key choices to be made when devising front identification methods. These choices include the physical quantity used to define the front, the mathematical function operating on said quantity, with the vertical level on which the analysis is performed, the threshold value(s) to be met and the algorithm used to plot the front on a chart.

*a. Choice of physical quantity*

As Thomas and Schultz (2019a) point out, fronts are defined by density discontinuities and hence their identification should be based on air temperature, or potential temperature ($\theta$ hereafter), as in Sanders and Hoffman (2002). However, equivalent potential temperature ($\theta_e$) is also a commonly used quantity, given that it is conserved for moist-adiabatic processes and can detect fronts at all times of day (the same properties apply to $\theta_w$, i.e., wet-bulb potential temperature). Several works used $\theta_e$ or $\theta_w$ (mainly at 850 hPa) to produce front climatologies, mostly at midlatitudes, and analyses of their properties; see e.g., Hewson (1998); Berry et al. (2011a, b); Schemm et al. (2015, 2018); Spensberger and Sprenger (2018); Catto and Raveh-Rubin (2019). As outlined in the comprehensive discussion in Schemm et al. (2018), a key property of $\theta_e$ is its dependence on moisture. Being dependent on moisture content, $\theta_e$ enhances cross-front gradients when both temperature and moisture gradients point

in the same direction (Thomas and Schultz, 2019a). While this property might be regarded as a possible shortcoming in midlatitudes, it makes $\theta_e$ particularly useful for our study. This is because the EASMF forms a boundary between tropical and extratropical air masses, with markedly different average moisture contents, leading to strong moisture gradients on top of weaker temperature gradients (see Li et al. (2018) and references therein). Indeed, this advantage is shown by Thomas and Schultz (2019a) in their comparison maps of frequencies of threshold exceedance using a variety of quantities and functions for frontal identification. A narrow, zonally-oriented local maximum gradient of $\theta_e$ at 850 hPa is found in June between east China and the west Pacific, just south of Japan, while this maximum is not observed in the gradient of $\theta$ (see their Figures 4c and 3c, respectively, and Figure 5c for the difference between the two). A local maximum is present in the same area and season in the specific humidity gradient (their Figure 6c), confirming that the EASMF separates flows with different moisture contents. These results are consistent with the choices made by Tomita et al. (2011) and Li et al. (2018, 2019), which depicted the interannual variability of EASM progression towards Japan and over China, respectively, using $\theta_e$ as the only atmospheric parameter.

Other approaches to front identification, not based on thermodynamic quantities, are documented in the literature. For instance, Schemm et al. (2015) investigate the temporal changes in 10 m wind, while Chen et al. (2003) and Wang et al. (2016) take advantage of potential vorticity inversion, and Dai et al. (2021) apply self-organising map techniques on the 850 hPa wind field. Thomas and Schultz (2019a, b) explore the role of frontogenesis, and Yang et al. (2015) devise a dynamically-defined frontogenesis function. Other studies instead focus on rainfall (e.g., Day et al., 2018; Kong et al., 2017), using a variety of algorithms. The breadth of different methods reflects the variety of subjects tackled by those studies, going from global reviews (Thomas and Schultz, 2019a, b), climatologies of Mei Yu (Dai et al., 2021; Kong et al., 2017; Day et al., 2018) and mid-latitude fronts (Schemm et al., 2015), to single Mei Yu case studies (Yang et al., 2015; Chen et al., 2003; Wang et al., 2016). We recognise that these different techniques, based on dynamics rather than thermodynamics, can be particularly valid when the front under study possesses low baroclinicity or when the thermodynamic boundaries and wind-field boundaries are not collocated. While this can partially be the case for the EASMF, particularly at short timescales, the focus of this study is on the climatological role of the EASMF as a boundary between tropical and extratropical air masses. Therefore, we choose a thermodynamic variable ($\theta_e$), containing information on both temperature and moisture. In the results, we make extensive comparison of the front location, as identified by our algorithm, with the climatological features of rainfall and low-level winds that characterise EASM progression.

*b. Choice of mathematical function*

Regarding the operating function, the studies above used mainly the horizontal gradient of $\theta_e$ (or $\theta_w$) or followed Hewson (1998) in using the thermal front parameter (TFP). The extensive review of frontal analysis methodologies provided in Thomas and Schultz (2019a) compares these two functions, outlining their relative strengths and weaknesses. Their results show how the gradient and TFP are both effective functions, with the gradient of $\theta_e$ working well at identifying airmass boundaries. TFP contains second derivatives and is thus prone to noise when dealing with fields of high spatial variability, such as high-resolution data and/or expressions containing (non-linear dependence on) humidity, such as $\theta_e$. For this reason, (and for the added advantage of identifying the frontal zone, rather than the warm edge of the thermal gradient) the $\theta_e$ gradient is preferred

to TFP in our study. In particular, we will consider the full horizontal $\theta_e$ gradient at 850 hPa, in agreement with most preceding references. However, there are three studies, all focusing on the EASMF (Li et al., 2018, 2019; Tomita et al., 2011), that use only the meridional component of the horizontal gradient of $\theta_e$. They justify their choice by the predominant zonal orientation of the EASMF. Neglecting the zonal component of the gradient helps to filter out local features that are not related with the main front (e.g., terrain or land/sea generated). However, we choose not to ignore the meridionally oriented component of the EASMF and thus compute the full horizontal gradient of $\theta_e$. Finally, it is also worth mentioning the innovative approach presented by Spensberger and Sprenger (2018), in which fronts are defined as three-dimensional coherent volumes where the gradient of $\theta_e$ exceeds a certain threshold (computed between 700 and 950 hPa). For the sake of simplicity, we do not explore this technique here, but the promising results presented in the article encourage a future implementation of the technique in EASM studies.

### 2.2.2 Description of the algorithm used in this study

The algorithm used to detect the EASMF in our analysis was inspired by Li et al. (2018, 2019). However, as explained above, our algorithm is adapted to consider the full horizontal gradient of $\theta_e$ and its code is based on that used in Hart et al. (2017). The input data considered consist of 6-hourly atmospheric fields (temperature and specific humidity) at 850 hPa, on a $0.25°$ $\times\ 0.25°$ horizontal grid, in a domain covering 105°E to 145°E, 15°N to 50°N. A number of preparatory steps are performed before the algorithm is applied and the frontal line is drawn, as summarised below:

1. Computing horizontal $\theta_e$ gradient:

    (a) Compute daily averages of temperature and specific humidity at 850 hPa and of surface pressure ($p_s$) for each grid point;

    (b) Mask points below the ground, i.e. where daily $p_s$ < 850 hPa;

    (c) Compute $\theta_e$ and its horizontal gradient at 850 hPa for each unmasked grid point:

    $$\nabla_h \theta_e = \frac{\partial \theta_e}{r\cos(\phi)\partial\lambda}\hat{\lambda} + \frac{\partial \theta_e}{r\partial\phi}\hat{\phi} \quad , \tag{1}$$

    where $r$ (km) is the Earth's radius, $\lambda$ (rad) is longitude, and $\phi$ (rad) is latitude (see Section 2.5 for more details).

2. Selecting locations exceeding the chosen $|\nabla_h \theta_e|$ threshold:

    (a) Find locations in the domain where the magnitude of the gradient exceeds the chosen threshold (set at 0.05 K km$^{-1}$, consistent with literature presented in Section 2.2.1);

    (b) Exclude points where the meridional component of the gradient is positive, i.e., where $\theta_e$ increases northwards. Given the particular geographical features of the area under study, this step removes coastal and/or terrain-induced local fronts, e.g., around Taiwan, that are not associated with the EASM, but rather semi-permanent features.

3. Restricting the meridional extent of the domain:

(a) Find the mean gradient-weighted latitude ($\overline{\phi_{gw}}$) of all points exceeding the threshold:

$$\overline{\phi_{gw}} = \frac{\sum_{i=1}^{N} \phi_i \cdot |\nabla_h \theta_e|_i}{\sum_{i=1}^{N} |\nabla_h \theta_e|_i} \quad , \tag{2}$$

where N is the number of all unmasked grid points in the domain exceeding the threshold, $\phi_i$ and $|\nabla_h \theta_e|_i$ are the latitude and the magnitude of horizontal $\theta_e$ gradient at each of those points, respectively;

(b) Restrict the latitudinal extent of the domain to a 20° interval centred on $\overline{\phi_{gw}}$. This step is necessary as using the full latitudinal extent of the domain (15°N-50°N) would increase the risk of other large-scale fronts, not associated with the EASM, being detected by the algorithm.

After these preparatory steps, the frontal line can be drawn following the algorithm outlined in the flow chart in Figure 1a. The other panels in Figure 1 show examples of daily frontal lines from the 1979 season, drawn by applying the algorithm and the post-processing step detailed later in this section.

As seen in the flow chart, in this algorithm the latitudes constituting the frontal line are identified one longitude at a time. Therefore, the resulting line could in principle be dependent on the choice of starting longitude, particularly if a frontal band fractures into two or more parts and/or terrain features come into play. However, the algorithm is designed to be independent from this choice, as it is repeated for 5 different starting longitudes, and then, at each longitude in the domain, the modal value of the selected latitudes is computed. Since the existence of small-scale multiple fronts within the EASMF was found more likely over land than ocean (not shown), starting longitudes selected in this study cover the inland longitudinal extent of the region affected by EASM progression, every 3° from 108°E to 120°E. In each of the 5 iterations the line identification starts from a longitude that is inside the domain, and thus, has points both to its east and west. Therefore, the algorithm scans all longitudes between the starting longitude and eastern boundary and then from the starting longitude to the western boundary.

The latitudinal bands identified by the algorithm are sets of at least three contiguous points exceeding the $|\nabla_h \theta_e|$ threshold. This ensures that only bands at least 0.5° wide are considered (e.g., the latitudes of the three contiguous point could be 28°N, 28.25°N, and 28.5°N), excluding isolated above-threshold values that could be related to geographical features rather than associated with a frontal passage.

As indicated in the flow chart, if a point was selected at the previous longitude, the algorithm is initially applied in a restricted portion of the domain (5° wide) centred around the latitude of that point. If no bands are found, then the algorithm is applied to the whole whole latitude range of the domain (20°, centred around the gradient-weighted mean latitude computed in the preparatory steps). The rationale of this choice is to prioritise above-threshold bands with similar latitudes. This is consistent with the aim of this process, i.e., identifying a single coherent large-scale front, rather than jumping between various smaller-scale fronts located at substantially different latitudes.

The only post-processing step that is necessary after having applied the line-drawing algorithm consists of identifying all latitudinal jumps of more than 5 degrees between neighbouring longitudes. If the front longitudinal width between two such jumps is less than 2.5°, then this section of the frontal line is treated as a separate small-scale feature, not part of the main front, and those points are removed from the line. This step, together with the chance that no bands over the gradient threshold

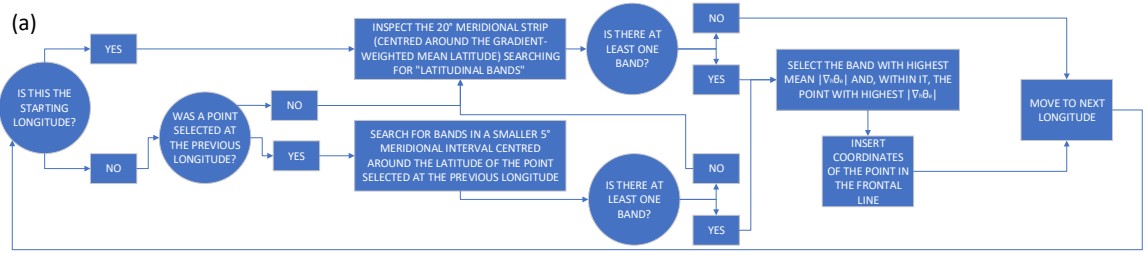

**Figure 1.** (a) Flow chart of the front-detection algorithm applied at each longitude in the domain to draw daily EASM frontal lines. (b)-(e) EASMF daily locations on the indicated dates. Red empty dots indicate the coordinates of the points constituting the frontal line, shading indicates $|\nabla_h \theta_e|$ (K km$^{-1}$), magenta contours enclose regions exceeding $|\nabla_h \theta_e|$ threshold (0.05 K km$^{-1}$), black dashed contours indicate $\theta_e$ (every 3 K, labelled every 6 K only in above-threshold regions). All values are masked out at levels below the ground. Data from ERA5.

are identified at certain longitudes, imply that gaps in the line are possible. This reflects the possibility of the actual gaps in the EASMF.

As explained previously, our front identification algorithm is applied on daily averages of 6-hourly ERA5 data. This results in outputs of daily locations of the front. The frontal lines shown in Sections 3, 4 and 5 are then produced by averaging, for each longitude, all the daily latitude values for the chosen dekad (10-day mean) across all the years considered. Examples of daily frontal lines are shown in Figure 1b-e. Those panels give examples of what these daily front locations look like, and how they change during EASM progression. Figure 1b contains a wide and easily identifiable high-gradient area, resulting in a smooth and extended EASMF. The following panels show a more complex picture, with fractured frontal bands and different northward progression between land and ocean, motivating the use of multiple starting longitudes and the other various filtering steps described in this section. Figure 1e contains a narrow Mei Yu front, fracturing over land. It is worth noting that at some longitudes no clear front is present and therefore a front latitude is not identified. This happens more frequently near the western boundary of the domain, suggesting a terrain influence. Longitude values at which the front latitude is not identified for more than half of the days considered, are excluded from the frontal lines shown in Figures 4-6, 8-9, 14-15.

## 2.3 Use of Lagrangian trajectories

Lagrangian trajectories are a tool that has been widely used to investigate the evolution of air masses, highlighting the processes occurring along them. This study takes advantage of the technique, building on methods recently applied to both tropical and extratropical weather features (see Volonté et al. (2020) for a recent application on the progression of the Indian summer monsoon). Here, trajectories are computed using the LAGRANTO Lagrangian analysis tool (Sprenger and Wernli, 2015). LAGRANTO uses a three-time iterative forward Euler scheme with an iteration step equal to 1/12 of the time spacing of the input data, i.e 30 minutes for the 6-hourly ERA5 data used in this work.

Backward trajectories are computed by using the instantaneous three-dimensional wind field to calculate the positions of the selected air parcels. Local values of the relevant physical quantities are interpolated onto those trajectories. Therefore, trajectories need specific times in which their journey takes place. In this study we are looking at the ERA5 1979-2018 climatology, hence a 'climatological year' needs to be constructed. This is done by taking at each time and each point the average value of all primary fields of the dataset $(u, v, w, q, T, p)$ for that time and that grid point in all years; e.g., $u$ at (115°E, 25°N, 700 hPa) at 01 Jun 00 UTC in the climatological year is constructed by averaging the 40 $u$ values referred at 01 Jun 00 UTC at that grid point over the years 1979-2018.

## 2.4 Measures of moisture content and transport

The identification of areas with high values of moisture content and transport has a primary role in this study. Integrated vapour transport, IVT hereafter, is widely used to identify atmospheric rivers and more generally regions of enhanced moisture

transport (Dacre et al., 2015; Rutz et al., 2014), also over the EASM region (Liang and Yong, 2020). Here, IVT is defined as:

$$IVT = | -\frac{1}{g} \int_{1000hPa}^{500hPa} q\mathbf{V}\,dp\,| \quad , \tag{3}$$

where $g$ is the gravitational acceleration [m s$^{-2}$], $q$ is specific humidity [kg kg$^{-1}$], $\mathbf{V}$ is the total horizontal wind vector [m s$^{-1}$], and $p$ is pressure [Pa]. IVT units are thus kg m$^{-1}$s$^{-1}$. The vertical integration is performed using all 16 output levels between 500 and 1000 hPa available in the ERA5 dataset. At all grid points where 1000 hPa is below the ground, the integration starts at the first above-ground level. In the same way, integrated water vapour (IWV [kg m$^{-2}$], defined below) is used for analysis of whole-column moisture content:

$$IWV = -\frac{1}{g} \int_{1000hPa}^{500hPa} q\,dp \quad . \tag{4}$$

### 2.5 Wind divergence and front deformation

In Section 3 of this study we analyse upper-level horizontal wind divergence and low- and mid-level front deformation. In particular, we compute (horizontal) wind divergence at 200 hPa, commonly defined as:

$$\nabla \cdot \overline{\mathbf{V}} = \frac{\partial u}{\partial x} + \frac{\partial v}{\partial y} \quad . \tag{5}$$

On a spherical coordinate system with latitude equal to zero at the Equator and increasing northwards, this becomes:

$$\nabla \cdot \overline{\mathbf{V}} = \frac{\partial u}{r\cos(\phi)\partial \lambda} - \frac{v\tan(\phi)}{r} + \frac{\partial v}{r\partial \phi} \quad , \tag{6}$$

where r (km) is the Earth's radius, $\lambda$ (rad) is longitude, and $\phi$ (rad) is latitude. Chapter 4 in Hoskins and James (2014) contains more details on this coordinate system and its applications to meteorology.

Moving to front deformation, that we compute at 850 hPa and 500 hPa, a clear and comprehensive description of how fluid flow can be characterised in terms of divergence, rotation and deformation can be found in Chapter 2 of Hoskins and James (2014). In particular, deformation F is defined as:

$$F = \sqrt{F_1^2 + F_2^2} \quad , \tag{7}$$

with F1 and F2 being the two deformation components, with an angle of 45° between each other, defined as:

$$F_1 = \frac{\partial u}{\partial x} - \frac{\partial v}{\partial y} \quad , \tag{8}$$

$$F_2 = \frac{\partial u}{\partial y} + \frac{\partial v}{\partial x} \quad . \tag{9}$$

Using the spherical coordinate system already mentioned, F1 and F2 can be written as:

$$F_1 = \frac{\partial u}{r\cos(\phi)\partial \lambda} - \frac{v\tan(\phi)}{r} - \frac{\partial v}{r\partial \phi} \quad , \tag{10}$$

$$F_2 = \frac{\partial u}{r\partial\phi} + \frac{\partial v}{r\cos(\phi)\partial\lambda} + \frac{u\tan(\phi)}{r} \quad . \tag{11}$$

## 3 EASM climatological progression: a front-centred perspective

In this section the ERA5 climatology (1979-2018) is used to provide a comprehensive description of the seasonal EASM progression and its associated stages. Using the EASMF as a reference, the key dynamics associated with EASM progression are outlined and the links between them (and with the EASMF) are highlighted. These results inform the airstream-focused analysis that follows in Section 4.

### 3.1 Front progression

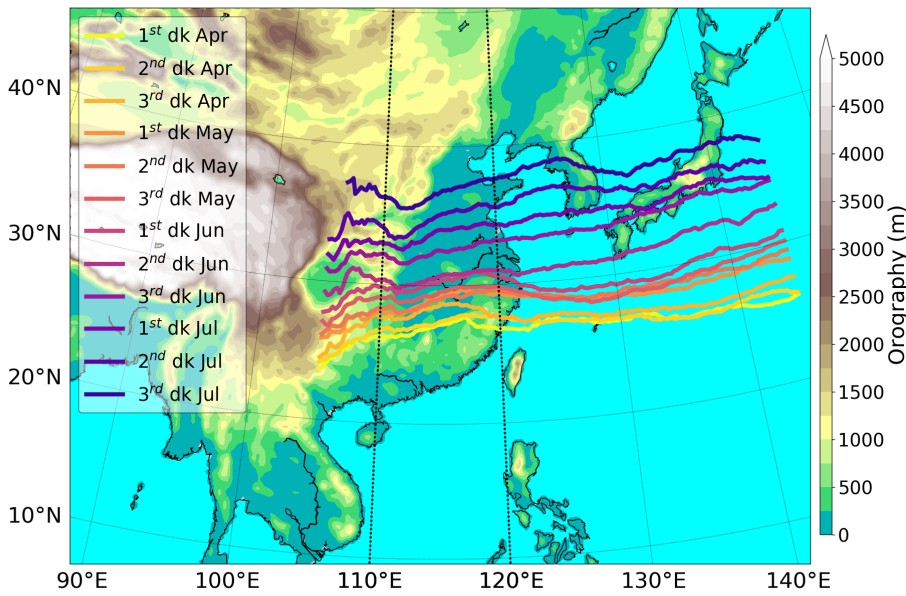

**Figure 2.** Dekadal (10-day) mean location of EASM frontal band at 850 hPa, as detected by the frontal detection algorithm (colour lines, see legend), and orography (shading, m). 110°E and 120°E meridians are highlighted. Data from ERA5, 1979-2018 climatology.

The algorithm developed to identify the EASMF, described in Section 2.2, allows us to assess the location and northward movement of the main front associated with the EASM during its progression. Figure 2 shows dekadal (10-day) means of front location during EASM progression. Focusing mainly over Eastern China, it can be observed that the front initially displays a slow northward progression. The EASMF, a warm front, moves from 26-27°N in the $1^{st}$ dekad of April to 30-31°N in the $2^{nd}$ dekad of June. A clear change in pace in the northward movement of the front then follows. This sudden onset of poleward progression sees the front moving north of 35°N by the $2^{nd}$ dekad of July. The pattern outlined, consisting of a quick northward

progression of the EASMF from mid June which follows a period of much smaller meridional displacement during *spring* is in good agreement with the literature described in Section 1, see Li et al. (2019), Wang and LinHo (2002) and Kong and Chiang (2020) among others.

## 3.2 Rainfall evolution

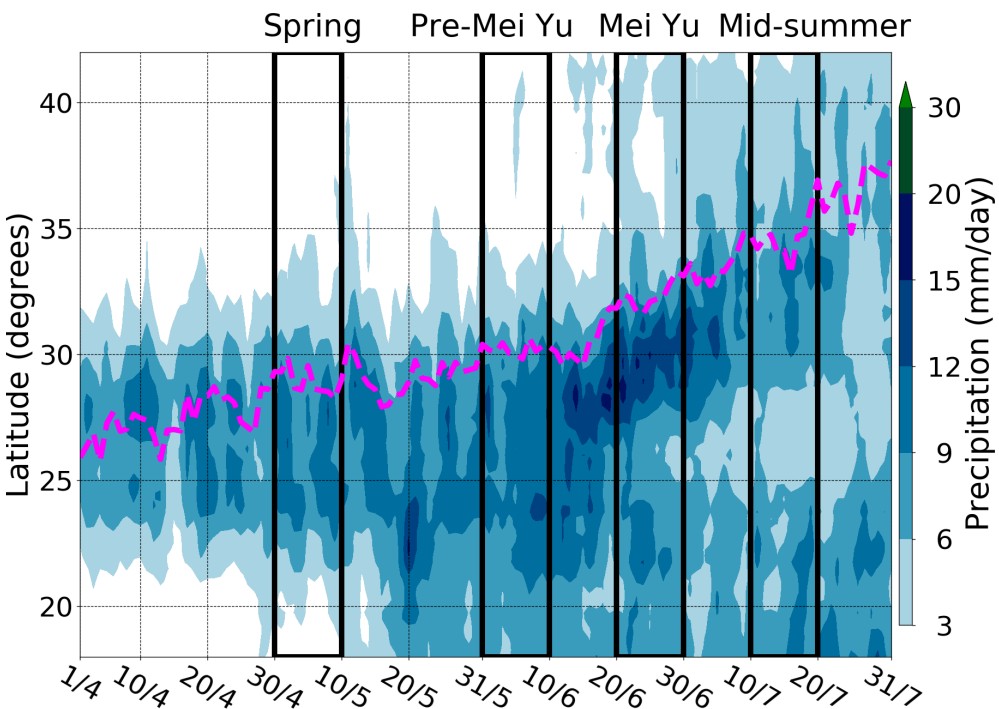

**Figure 3.** Hovmöller plot of climatological precipitation (mm day$^{-1}$) averaged between $110°$E - $120°$E. The magenta dashed line indicates the daily latitude of the EASMF at 850hPa, averaged between $110°$E - $120°$E. Bold black lines indicate the dekads selected as representative of the four stages of EASM progression. Data from ERA5, 1979-2018 climatology.

Figure 3 contains a Hovmöller plot that combines information on precipitation and frontal latitude over eastern China ($110°$E-$120°$E). The seasonal progression of EASM rainfall is characterised by distinct stages, which have been thoroughly described in literature (see Chiang et al. (2017); Kong et al. (2017); Wang and LinHo (2002) and our Introduction for more details). These stages can be identified in the diagram, and are consistent with literature (in particular, see an equivalent diagram in Figure 1 in Kong et al. (2017)). Dekads (periods of 10 days) that are fully contained in each of the stages, of which they 335  are representative in both features and timing, are selected here and will be used throughout the analysis, allowing the comparison of dynamics and weather features between periods of equal length belonging to the four different stages. At the same time, Figure 3 provides a visualisation of the northward progression of the front throughout the season outlined in Figure 2, highlighting the abrupt pace change occurring in the $2^{nd}$ dekad of June.

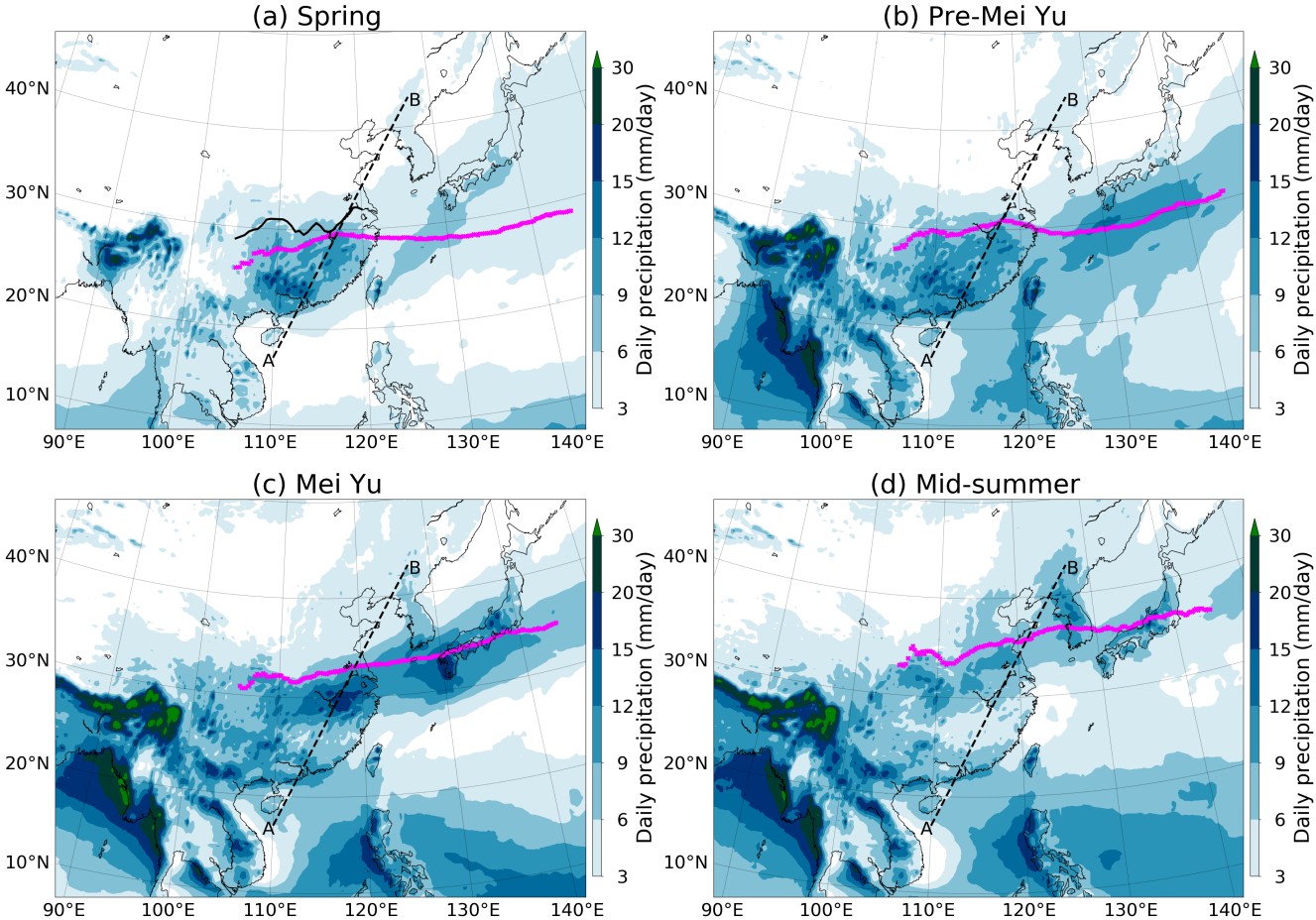

**Figure 4.** Dekadal mean of daily precipitation (shading, mm/day). The magenta line indicates the climatology of the mean dekadal location of the EASMF at 850 hPa. The AB black dashed line indicates the transect of the vertical cross-section in Figure 10. The black solid line in panel (a) indicates the middle and lower course of the Yangtze river. The dekads representing the four stages of EASM progression ($1^{st}$ dk May, $1^{st}$ dk June, $3^{rd}$ dk June, $2^{nd}$ dk July, respectively) are selected according to Figure 3. Data from ERA5, 1979-2018 climatology.

In Figure 3, spring rains are visible until mid May, concentrated over southern China (22-30°N) south of the EASMF location and of the Yangtze river (whose middle and lower course flows from 105°E to 120°E at around or just above 30°N, see black solid line in Figure 4a). Next, the *pre-Mei Yu* stage takes place, with rainfall also occurring over the South China Sea and showing more intense peaks, as larger areas gradually exceed 9 mm day$^{-1}$ and localised short-term maxima go beyond 12 mm day$^{-1}$. *Pre-Mei Yu* gives way to *Mei Yu* around the middle of June, when a distinct and long-lasting band of precipitation heavier than 12 mm day$^{-1}$ (thus, more intense than in earlier stages) starts to progress northward. This band, around 2-3 degrees wide, starts centred around 28°N and reaches around 32°N by July, moving from south to north of the Yangtze river in the process. At the same time, precipitation reduces over southern China. During the first half of July, the *Mei Yu* rainband

decreases in intensity and becomes less clearly defined and separated from other precipitation: this is the final stage in EASM development, *mid-summer*, followed by its retreat (not shown). It is important to note that for the most part of *spring*, and then throughout *pre-Mei Yu* and *Mei Yu* stages, the EASMF finds itself on the northern side of the region of substantial (> 6 mm day$^{-1}$) precipitation. This is consistent with the concepts of spring and pre-Mei Yu rains being due to pre-monsoonal 'warm-sector' precipitation (i.e. to the south of the front) and the *Mei Yu* rainfall being caused by frontal activity (confirmed by rainfall being more intense and tied closer to the front than earlier).

Figure 4 shows the spatial patterns of the evolution of precipitation and front location among the four EASM stages, using the dekads selected in Figure 3. This figure, in good agreement with Kong et al. (2017) and Chiang et al. (2017), which used rain gauge data, highlights the main features of EASM seasonal rainfall evolution. Spring rains are characterised by substantial rainfall being mostly restricted to SE China (east of 105°E and between the South China Sea coast and the Yangtze river region), where the EASMF is located. The transition to *pre-Mei Yu* sees a considerable increase of precipitation throughout South Asia, associated with the formation of a C-shaped precipitation pattern spreading from the Maritime Continent and the Philippines to the northwestern Pacific Ocean and the southern edge of Japan. These variations indicate the effect of the South Asian monsoon onset and are associated with an intensification of rainfall also over China. The *Mei Yu* stage is then characterised by the sharpening and narrowing of a frontal band of precipitation centred over the lower course of the Yangtze river, with the EASMF at its northern edge and dekadal climatological maxima exceeding 15 mm/day. This rainfall band moves further north with the front, and decreases in intensity by *mid-summer*.

Both Figures 3 and 4 show good agreement with similar analysis recently published (see relevant panels in Figure 1 of Kong et al. (2017), constructed using APHRODITE 1951-2007 data, Yatagai et al. (2009)). The agreement is clear with regards to the existence of the progression stages, their timings and the associated precipitation patterns over eastern China.

### 3.3 Large-scale setting and role of the western North Pacific subtropical high

As explained in Section 1, the western North Pacific subtropical high (WNPSH hereafter) is a feature that has been shown to be associated with EASM progression, including: onsets of both the spring rains and Mei Yu phase coinciding with successive northward jumps of the WNPSH, and westward extension of the WNPSH helping bring about the demise of the Mei Yu phase by hindering the south-westerly flow (Yihui and Chan, 2005). In Figure 5 we highlight its role by showing the evolution of geopotential height at 500 hPa, along with the 2-PVU line at 250 hPa giving an indication of the boundary with extra-tropical air. During *spring* and, to some degree, *pre-Mei Yu* stages, geopotential height decreases fairly monotonically with increasing latitude in the East Asia and West Pacific regions, where the EASMF is located. During *Mei Yu*, a clear maximum is instead visible over the West Pacific, between 20° and 30°N, collocated with high sea-level pressure. This is the signature of the WNPSH. This anticyclone helps the low-level flow from the Pacific Ocean turn anticyclonically towards eastern China, where it merges with south-westerly flow of South Asian origin. This suggests that variability in the location and intensity of the WNPSH can affect the pattern of warm-air advection towards the EASMF. At the same time, the boundary of extra-tropical air (2-PVU line) moves northward, with smaller-scale structures indicating a more pronounced local variability in its latitude. This signal is accompanied by the gradual weakening of a climatological trough, present over northeast Asia, that, being in the

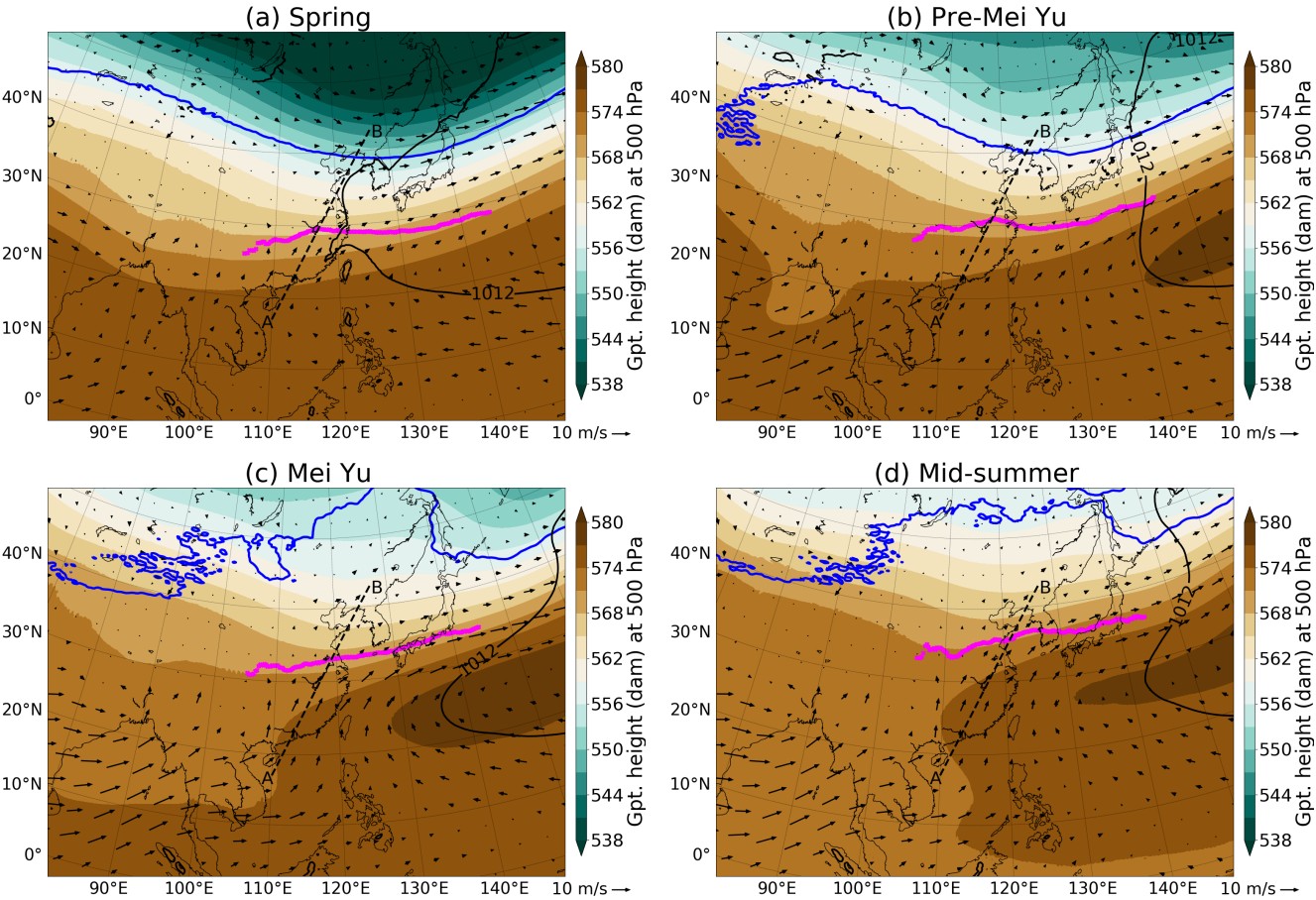

**Figure 5.** Dekadal mean of geopotential height at 500 hPa (shading, dam), mean sea-level pressure (black contour, 1012 hPa) wind vectors at 850 hPa (arrows, m s$^{-1}$) and potential vorticity at 250 hPa (blue contour, 2 PVU). The magenta line indicates the climatological mean dekadal location of the EASMF at 850 hPa. The AB black dashed line indicates the transect of the vertical cross-section in Figure 10. The dekads representing the four stages of EASM progression ($1^{st}$ dk May, $1^{st}$ dk June, $3^{rd}$ dk June, $2^{nd}$ dk July, respectively) are selected according to Figure 3. Data from ERA5, 1979-2018 climatology.

lee of the TP, is at least partially terrain-driven (Kong and Chiang, 2020). This trough, albeit weaker, is still visible during *Mei Yu*. Its axis, indicated by the cyclonic turn in geopotential height contours, lies between eastern Russia and Japan.

Figure 6 shows front deformation (F), defined in Section 2.5, at low levels (850 hPa, shading) and in the middle troposphere (500 hPa, contours). F generally increases with latitude in the early stages of EASMF progression, with higher values and noisier patterns present over the Tibetan Plateau, where the ground is not far from 500 hPa. By the *Mei Yu* stage, a local F maximum in the shape of a near-zonal band is clearly visible at both levels shown (850 and 500 hPa), extending from eastern China towards southern Japan, where the highest values are displayed. This band is collocated with the EASMF and associated

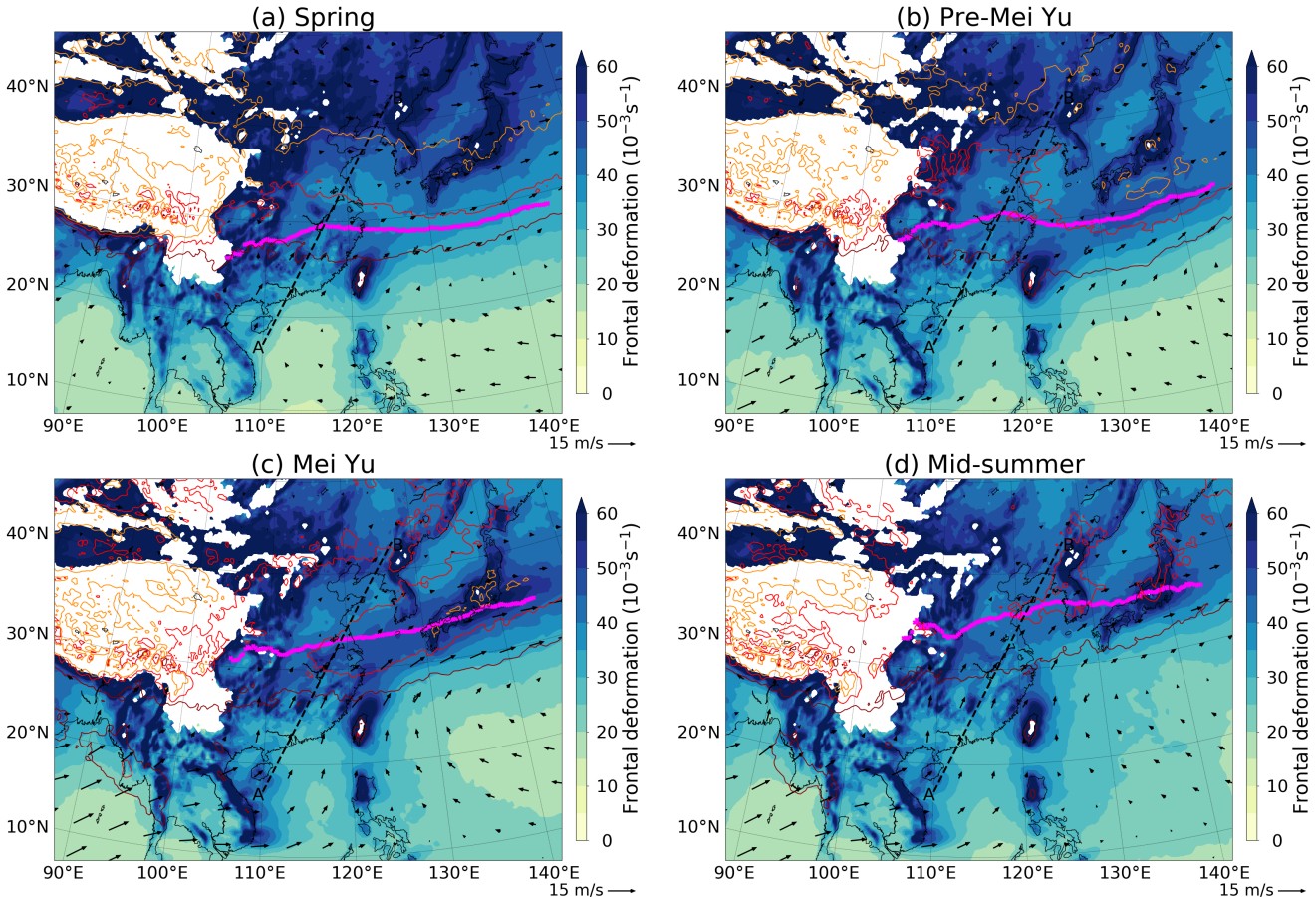

**Figure 6.** Dekadal mean of front deformation at 850 hPa (shading, $10^{-3}$ s$^{-1}$) and at 500 hPa (brown, red and orange contours indicating 35, 42.5, 50 · $10^{-3}$ s$^{-1}$, respectively), wind vectors at 850 hPa (arrows, m s$^{-1}$). The magenta line indicates the climatology of the mean dekadal location of the EASMF at 850 hPa. The AB black dashed line indicates the transect of the vertical cross-section in Figure 10. The dekads representing the four stages of EASM progression ($1^{st}$ dk May, $1^{st}$ dk June, $3^{rd}$ dk June, $2^{nd}$ dk July, respectively) are selected according to Figure 3. All values are masked out at levels below the ground. Data from ERA5, 1979-2018 climatology.

with low-level flow going from southwesterlies to westerlies, on the northern side of the WNPSH, pointing at a near-zonal axis of dilation, confirmed by F1 being dominant over F2 (not shown). Given that the temperature gradient is orientated across the

front (warm to the south and cold to the north, not shown), this situation resembles a classic example of frontogenetic flow, i.e., a flow intensifying temperature gradients at a front (Hoskins and James, 2014). The band then progresses northwards in *mid-summer* while weakening. In summary, the analysis of front deformation highlights the band-shaped maximum occurring during *Mei Yu* on the northern edge of the WNPSH, between this feature and the midlatitude flow, associated with frontogenetic

flow. This result gives further indication that the location and strength of the WNPSH play a key role in the progression of the
395 EASMF.

### 3.4 Role of the subtropical westerly jet and its northward migration

A considerable body of literature highlights the influence of the TP on the flow of the subtropical westerly jet (STWJ hereafter) over Asia, with clear consequences for seasonal progression of the EASM. Kong and Chiang (2020), in particular, show that the demise of the front during *Mei Yu* is accompanied by the northward migration of the STWJ axis away from the northern
edge of the TP, i.e beyond 40°N. This is consistent with the Hovmöller plot in Figure 7a, which contains wind speed at 200 hPa at TP longitudes and the latitude of the 850 hPa EASMF over eastern China. The STWJ axis moves beyond 40°N by the beginning of the $2^{nd}$ dekad of July, i.e., by *mid-summer*. The core of the STWJ is more than 5° further north than the front from the beginning of *pre-Mei Yu* through to *mid-summer*. This is different from the earlier *spring* stage, in which the STWJ core sits at the same latitudes of the EASMF, before weakening while crossing the TP.

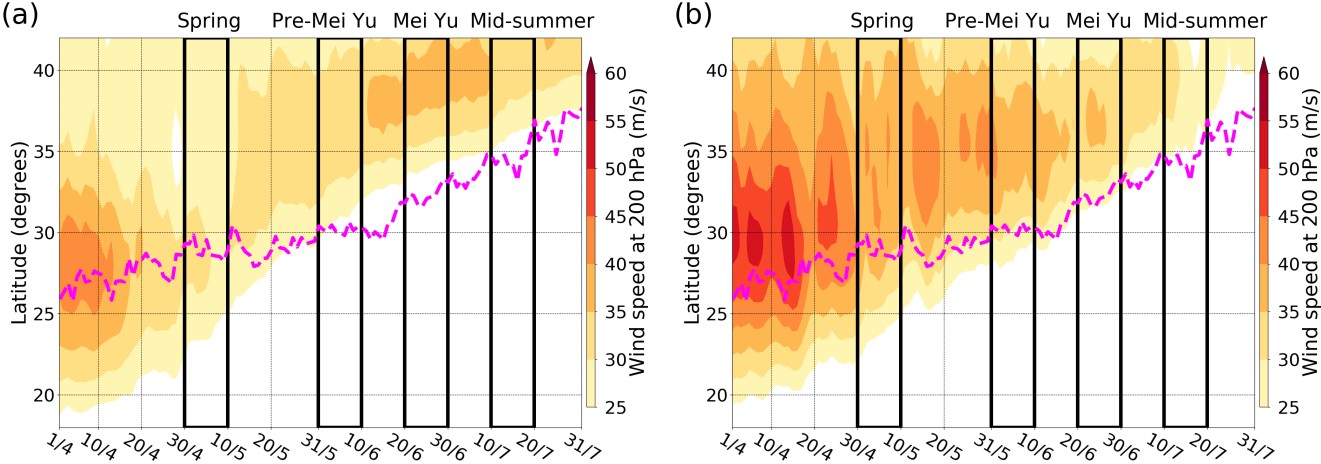

**Figure 7.** Hovmöller plots of wind speed at 200 hPa (m s$^{-1}$) averaged between (a) 80°E-100°E and (b) 110°E-120°E. The magenta dashed line indicates the daily latitude of the EASMF at 850hPa, averaged between 110°E-120°E. The dekads representing the four stages of EASM progression (bold black lines) are selected according to Figure 3. Data from ERA5, 1979-2018 climatology.

In the Hovmöller plot in Figure 7b, STWJ and EASMF both refer to the same longitude interval (110°E-120°E), representative of eastern China. In this case, the core of a strong STWJ is almost on top, just on its northern side, of the EASMF during *spring*. Initially, the STWJ displays a faster northward progression than the EASMF, while gradually weakening. In consequence, the southern edge of the jet, indicated by the 25 m s$^{-1}$ contour, lies on top of the EASMF by mid-June, i.e. at the beginning of the *Mei Yu* stage. At this stage, the northward migrations of the EASMF and of the STWJ have the same
pace. This implies that the latitudes of the front and of the southern edge of the STWJ are the same throughout the duration of *Mei Yu*. The analysis of STWJ latitude over the TP is certainly dynamically meaningful as it highlights the jet-orography

interaction that affects the circulation downstream (Kong and Chiang, 2020). However, the latitude of the STWJ southern edge over eastern China being equal to the latitude of the EASMF is also an important result, as it points at a possible role of the upper-level flow over East Asia in the dynamics of the front.

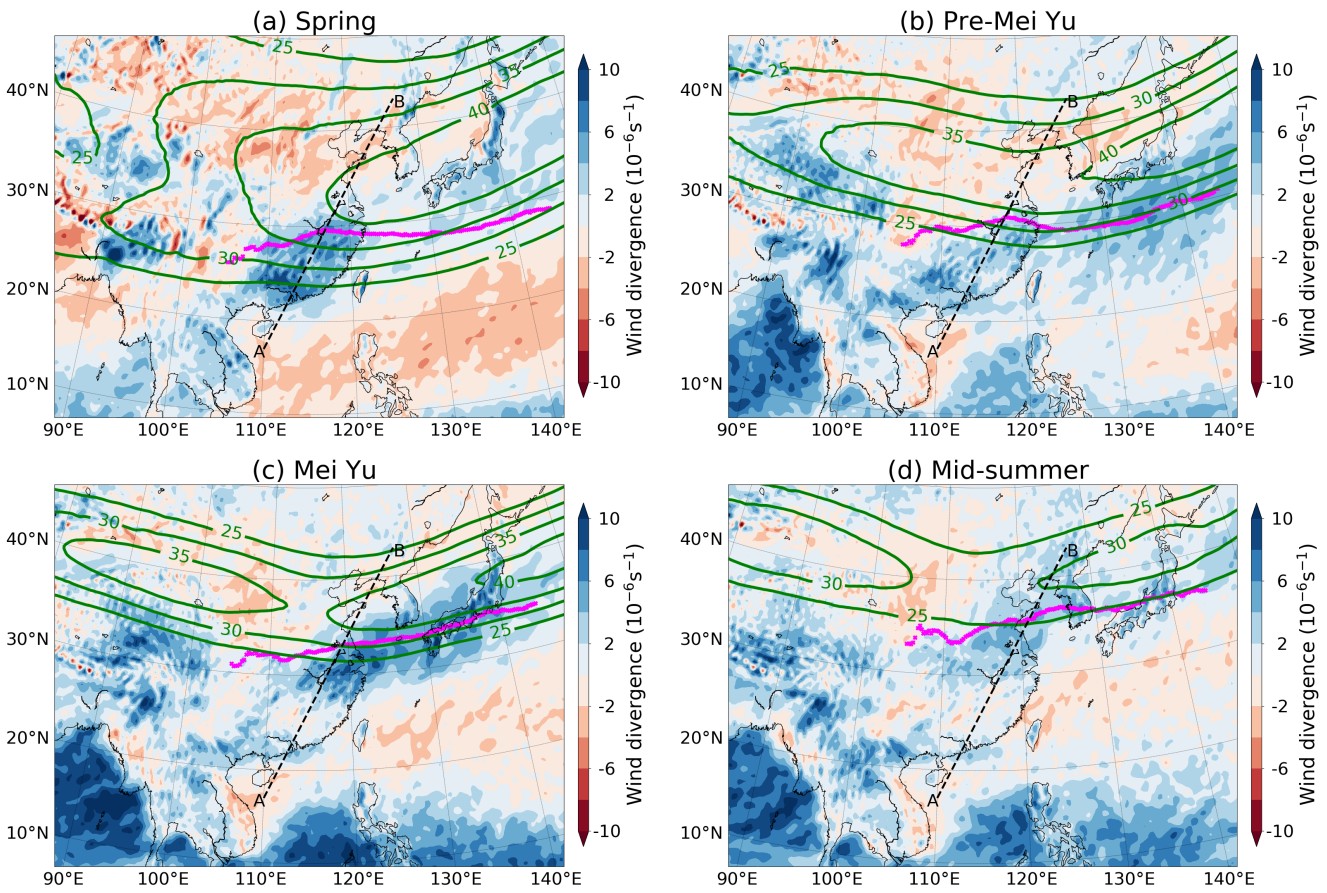

**Figure 8.** Dekadal mean horizontal wind divergence at 200 hPa (shading, $10^{-6}$ m s$^{-1}$) and magnitude of horizontal wind speed at 200 hPa (green contours, every 5 m s$^{-1}$ from 25 m s$^{-1}$ upwards). The magenta line indicates the climatology of the mean dekadal location of the EASMF at 850 hPa. The AB black dashed line indicates the transect of the vertical cross-section in Figure 10. The dekads representing the four stages of EASM progression ($1^{st}$ dk May, $1^{st}$ dk June, $3^{rd}$ dk June, $2^{nd}$ dk July, respectively) are selected according to Figure 3. Data from ERA5, 1979-2018 climatology.

This possible role is investigated in more detail in the maps of Figure 8. These maps highlight the poleward migration of the STWJ, associated with its narrowing and weakening, causing the core of the STWJ to drift away from the EASMF, and the southern edge of the jet to be on top of the front over eastern China from the *pre-Mei Yu* stage onwards. It is also important to note that, particularly during *pre-Mei Yu* and *Mei Yu*, the STWJ displays a trough, whose axis lies over eastern China. At

this location, the STWJ is weaker than over Japan. Therefore, the EASMF lies mainly in the right-entrance region of the downstream jet streak, a region where enhanced ascent and precipitation are favoured (Hoskins and James, 2014). The analysis of horizontal wind divergence at 200 hPa confirms this hypothesis, as it displays an area of positive divergence associated with the right entrance of the downstream jet and collocated with the EASMF in all stages of EASM progression. The region characterised by upper-level divergent flow shows the highest values during the *Mei Yu* stage, when a near-zonal band collocated with most of the EASMF is observed. Strong divergence is also observed over southern China during *spring* and over tropical oceans later in the season, consistent with the location of spring rains and the occurrence of boreal summer tropical convection, respectively. In summary, the presence of widespread upper-level divergence close to the EASMF throughout the stages of EASM progression, and its intensification and organisation in a front-collocated band during *Mei Yu*, indicates the key role of STWJ evolution in driving the dynamics of the EASMF.

### 3.5   Role of low-level advection of moist and warm air from the tropics

The evolution of moisture transport and content associated with EASMF progression is indicated in Figure 9 by the values of IVT and IWV, respectively. These quantities are computed with a vertical integration from 1000 hPa up to 500 hPa. However, their values would be similar if either restricted to lower levels only or extended to the tropopause (not shown), as specific humidity decreases substantially with height, and thus they can be seen as representative of the forcing of low-level moisture advection towards the front. The figure displays a weak relative maximum in IVT over southern China during *spring*, with little large-scale circulation at low levels. As the season progresses, the South Asian monsoon circulation picks up, along with the anticyclonic flow in the neighbouring west Pacific. This leads to the onset of southerlies over the South China Sea and the transport of moist tropical air over eastern China towards the EASMF, with IVT values increasing considerably over the region. A strong IVT maximum develops by *Mei Yu* on an elongated area south-west of Japan, just south of the EASMF, with values comparable to the maximum visible in the Bay of Bengal (BoB). IVT values are higher in this band between China and Japan, where zonal circulation at low levels strengthens, than over eastern China. This is not caused by a higher moisture content in the region, but by stronger winds enhancing the moisture transport, as is made clear by the IWV contours. The 48 kg m$^{-2}$ IWV isoline, limited to the Indochina peninsula in *spring*, progresses east, extending over the South China Sea and the near Pacific, and north, over south-eastern China. Its northern edge reaches the EASMF by the *Mei Yu* stage. By then, local maxima beyond 53 kg m$^{-2}$ are visible not only over South China but also further north, closer to the Yangtze river, consistent with the poleward transport of monsoon air towards the front. The magnitude of these IWV values is comparable to those present in the BoB (apart from the even higher values present over the Ganges-Brahmaputra delta), confirming that the air advected by the low-level southerlies towards the EASMF retains a moisture content typical of South Asian monsoon airmasses.

Figure 9 also shows equivalent potential temperature ($\theta_e$) at 850 hPa. One of the key visible features is the steep meridional gradient in $\theta_e$ over eastern China, with over 20 K difference between southern areas and the Yangtze river delta in *spring*. The progression of the season sees a gradual relaxing of this gradient over southern China, but less so in the region of the Yangtze river. In fact, high-$\theta_e$ air, associated with the aforementioned moist southerly flow (driven by joint forcing from the Asian monsoon and the WNPSH), is transported towards the EASMF. There, the northward march of the southerly flow is prevented

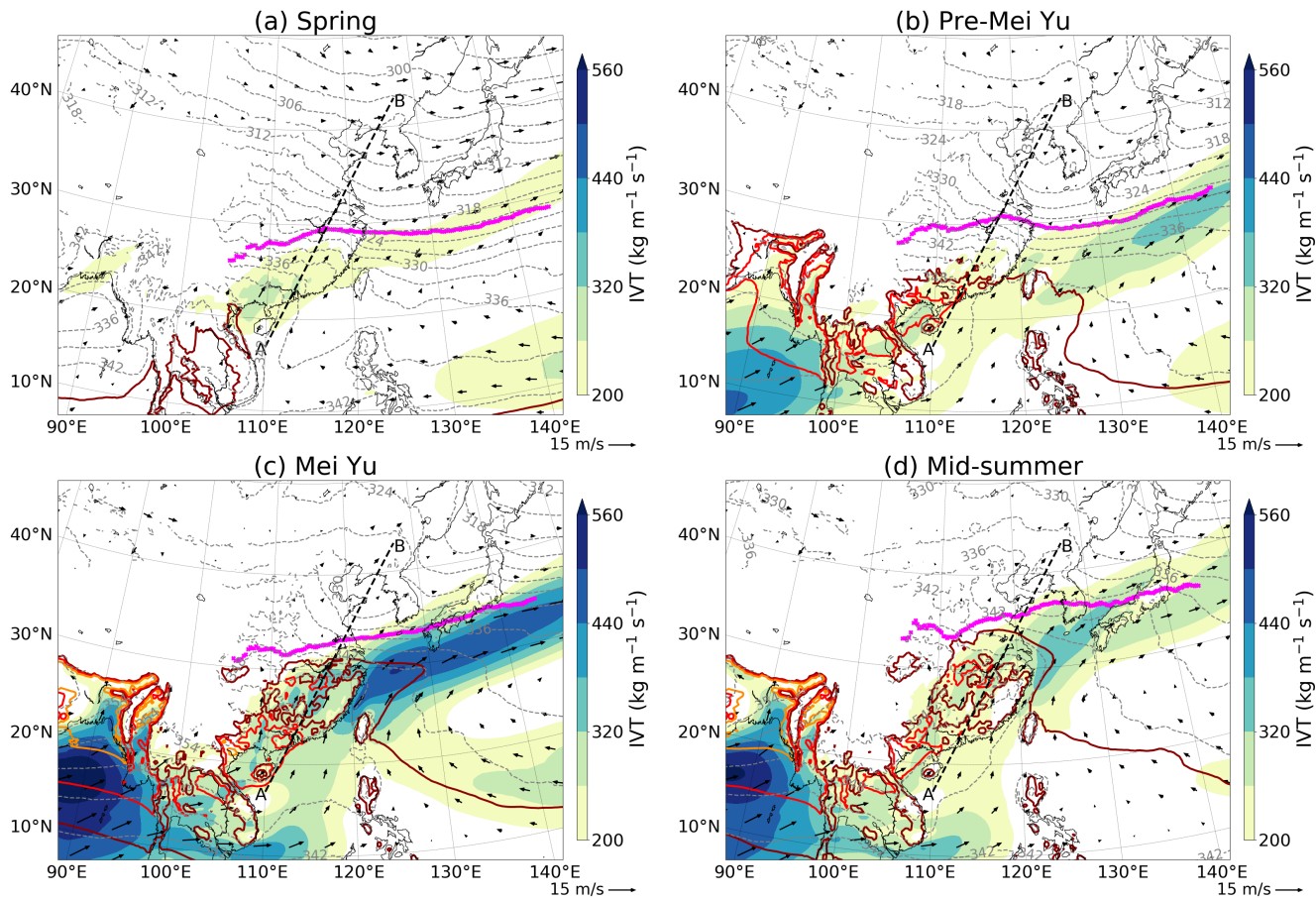

**Figure 9.** Dekadal mean of 500-1000 hPa IVT (shading, kg m$^{-1}$s$^{-1}$) and 500-1000 hPa IWV (brown, red and orange contours indicating 48, 53 and 58 kg m$^{-2}$, respectively), equivalent potential temperature at 850 hPa (grey dashed lines, every 3 K) and wind vectors at 850 hPa (arrows, m s$^{-1}$). The magenta line indicates the climatology of the mean dekadal location of the EASMF at 850 hPa. The AB black dashed line indicates the transect of the vertical cross-section in Figure 10. The dekads representing the four stages of EASM progression (1$^{st}$ dk May, 1$^{st}$ dk June, 3$^{rd}$ dk June, 2$^{nd}$ dk July, respectively) are selected according to Figure 3. All values are masked out at levels below the ground. Data from ERA5, 1979-2018 climatology.

by extratropical mid-level northwesterlies, caused by mechanical forcing of the STWJ by the TP (Chen and Bordoni, 2014; Kong and Chiang, 2020). During the *Mei Yu* stage, the area of steep $\theta_e$-gradient starts to move more substantially northward (as the edge of the STWJ does too, see earlier) and markedly decreases by *mid-summer*, when the high-$\theta_e$ is able to reach midlatitude regions. One further finding from this figure concerns the different pace of poleward progression of the warm air at different longitudes. In fact, during *Mei Yu*, there is a clear negative zonal $\theta_e$ gradient over north-eastern China, just above 30°N. This is consistent with the poleward advection of the highest $\theta_e$ values, found at the southeastern edge of the TP from

*pre-Mei Yu* onwards, thanks to elevated topography and latent heating. The very-high-$\theta_e$ air is transported towards the western part of the EASMF, in agreement with Sampe and Xie (2010), reaching it by *Mei Yu*.

## 4 Analysis of tropical-extratropical airmass interactions

The analysis of EASM progression presented above has highlighted the joint forcing of several features, tropical and extratropical (such as WNPSH, STWJ, moist low-level southerlies, midlatitude flow), on the progression of the EASM and of its front. In this section, we analyse the interaction between these different air masses associated with those features. We use vertical cross-sections to locate and highlight the key airstreams and then Lagrangian trajectories to reveal their path and evolution.

### 4.1 Vertical structure

Figure 10 completes the Eulerian analysis of EASM progression in the ERA5 climatology with vertical cross-sections whose design, inspired by similar figures in Terpstra et al. (2021), is intended to illustrate the evolution of the vertical structure of the troposphere during EASM progression, at the same time highlighting changes in the structure of the EASMF, in the lower-level airmass advection towards it, and in the upper-level STWJ, along with the associated precipitation. These SW-NE cross-sections (transect AB in Figures 4-9) lie parallel to the direction of EASM progression and the low-level warm southerly flow, crossing the Yangtze river and the EASMF over eastern China and extending north near the axis of the midlatitude climatological trough. The panels also serve as further verification of the reliability of the frontal detection algorithm, as they all show that the EASMF is correctly located in the region with maximum values of the horizontal gradient of equivalent potential temperature at 850 hPa.

All panels display high values of moisture flux associated with warm (high-$\theta_e$) low-level southerly advection. In *spring*, *pre-Mei Yu* and *Mei Yu* stages, these high moisture fluxes are confined to the southern side of the EASMF, indicated by a steep along-section gradient in $\theta_e$. $\theta_e$ contours close to the front location are roughly vertical throughout the lower troposphere in *spring*, indicating that the front is deep and it is separating air masses with considerably different thermodynamic properties. As the season progresses, low-level $\theta_e$ contours become more and more slanted. This is caused by the different heights at which warm monsoon air and cooler subtropical or extratropical air impact the front. The warm advection takes places in the lowest levels, at pressures higher than 800 hPa, where the moisture flux and $\theta_e$ values are highest and the southerly flow is clearly visible. Instead, the core of the tropospheric extratropical flow on the northern side of the EASMF is indicated by a minimum in $\theta_e$ and negligible along-section wind at around 700-800 hPa, consistent with this airstream flowing mainly from northwestern quadrants (as shown in greater detail in Section 4.2), perpendicular to the orientation of transect AB. As discussed in the previous section, the increase in intensity of the warm southerly flow is driven by the onset of the Asian monsoon and the anticyclonic flow in the neighbouring western Pacific region while the cool extratropical flow is associated with behaviour of the STWJ. The progression of the EASM season is also characterised by weakening and narrowing of the upper-level STWJ. During *pre-Mei Yu* and *Mei Yu* stages, the southern edge of the STWJ sits on top of the low-level EASMF. By *mid-summer*, $\theta_e$ contours are almost horizontal in the frontal region. The EASMF is considerably weaker throughout the troposphere, as is the

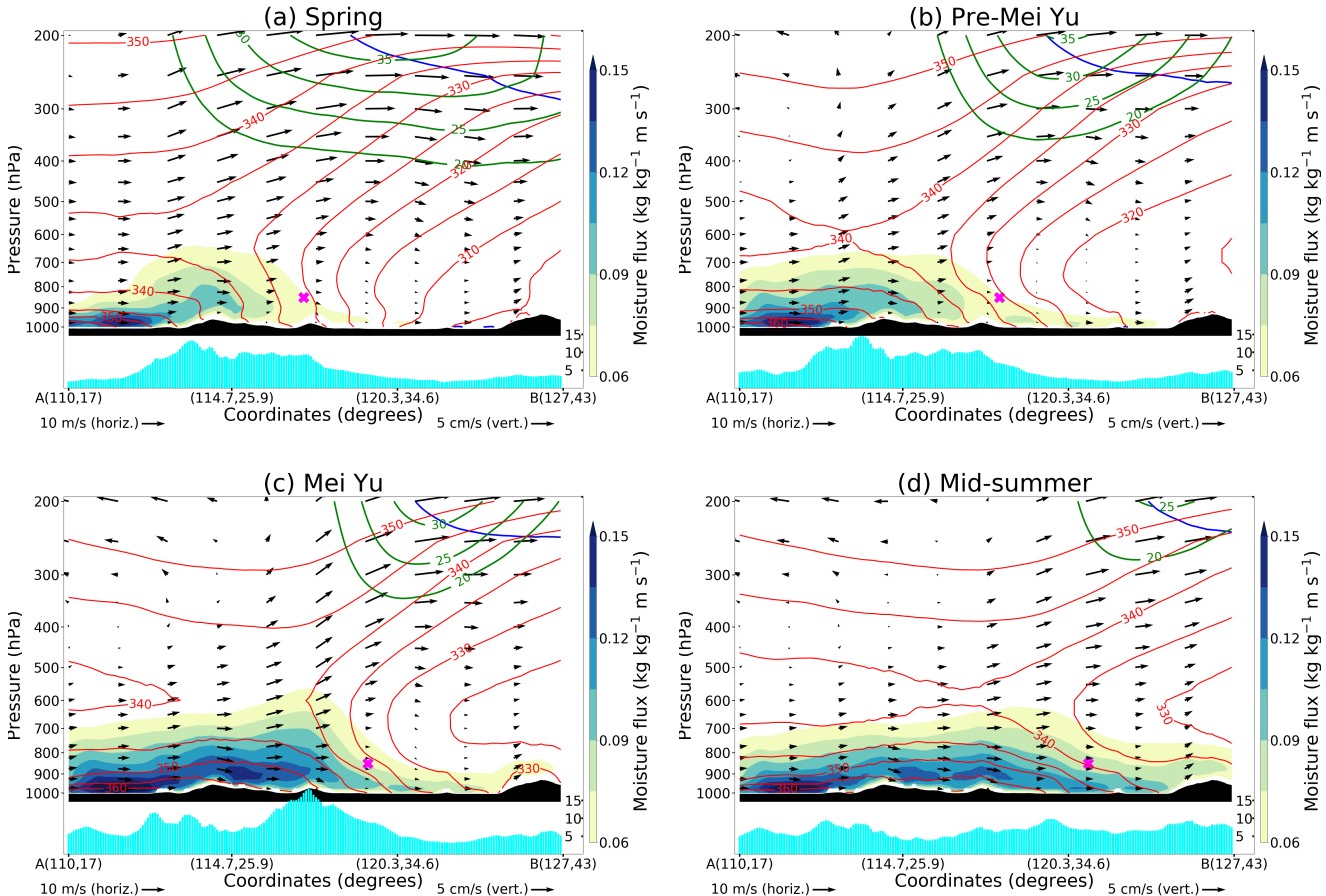

**Figure 10.** Cross-sections, transect AB in Figures 4-9, of dekadal mean moisture flux (shading, m s$^{-1}$), equivalent potential temperature (red contours, every 5 K), magnitude of horizontal wind speed (green contours, m s$^{-1}$), wind vectors (arrows, m s$^{-1}$, computed using the horizontal wind parallel to the section as horizontal component and vertical velocity as vertical component, multiplied by 200 to be consistent with the aspect ratio), potential vorticity (blue contour, 2 PVU) and precipitation (cyan bars, mm day$^{-1}$). The magenta 'x' indicates the location of the EASMF at 850 hPa according to the detection algorithm. The dekads representing the four stages of EASM progression (1$^{st}$ dk May, 1$^{st}$ dk June, 3$^{rd}$ dk June, 2$^{nd}$ dk July, respectively) are selected according to Figure 3. Data from ERA5, 1979-2018 climatology.

upper-level jet. The southerly moist flow is now able to progress further northward, with the meridional $\theta_e$ gradient becoming less important.

Focusing on precipitation, Figure 10 shows broad rain maxima on the southern side of the EASMF in the first two stages, with some degree of orographic enhancement over southern China (around 25°N). During *Mei Yu* the rain maximum moves close to the EASMF and becomes sharper. This is in agreement with the evolution of the moisture flux pattern, as high values are able to get close to the frontal surface. The absence of rain maxima as sharp as before is consistent with the general weakening of the front in *mid-summer*.

## 4.2 Lagrangian analysis of airstreams

Up to this point we applied Eulerian methods of analysis such as maps and vertical cross-sections, identifying the flows impacting the EASMF from the tropical and the extratropical side, and highlighting their differences in terms of height, direction and thermodynamic properties. Using Lagrangian backward trajectories, we can now isolate these airstreams and investigate their origin and path towards the EASMF. As shown previously, the EASMF is characterised by a steep gradient in $\theta_e$, particularly evident at low-levels and over eastern China. The advection of cool extratropical air is centred around 700-800 hPa, while the advection of warm tropical air occurs at lower altitudes. For this reason, the starting points of the backward trajectories are restricted to grid points between 700 and 900 hPa. The selection domain, for each climatological season, consists of the 115°E-120°E longitude interval and of a 1.5° latitude interval centred around the mean latitude of the EASMF within 115°E-120°E (values rounded to the nearest quarter of a degree, i.e., 29.5°N, 30.75°N, 32.75°N, and 35.75°N for *spring*, *pre-Mei Yu*, *Mei Yu*, and *mid-summer* stages, respectively). The 50 points in this domain with highest $\theta_e$ are selected as starting points for the warm airstream, while the 50 points with lowest $\theta_e$ constitute the starting points for the cool airstream. The calculation of the trajectories is based on the flow of the 'climatological year', see Section 2.3 for more details.

Figure 11 shows the path of the trajectories selected in each of the four stages that characterise EASM progression, highlighting the degree of tropical vs subtropical/extratropical interaction at the front. In fact, in *spring* this interaction is still absent, as both the warm and cool airstreams travel towards the front mainly from the South China Sea. The paths and heights of the two airstreams are similar, with the cool airstream slightly ascending and travelling just to the NW of the warm one, but both showing an anticyclonic gyre and a tropical origin. The situation starts to change in the *pre-Mei Yu* stage. The warm air is still mostly travelling from the south, although not recurving anticyclonically as the monsoon forcing starts to grow, while a minority of the cool trajectories, now descending from the mid-troposphere, instead travels from the northwest, suggesting a continental, extratropical origin. The interaction between tropical (monsoon) and extratropical air masses becomes evident in the *Mei Yu* stage, when both airstreams show a much longer path than in previous stages. The warm air travels all the way from the Equatorial Indian Ocean and the cool air, gradually descending, flows along the northern side of the TP, reaching the front from the northwest. Hence, during *Mei Yu* the EASMF is the area in which warm air from South Asia and cool flow from midlatitude Central Asia collide. This is in agreement with Figure 10c, that displays these different air masses converging at the front. As the EASMF weakens by *mid-summer*, its nature of separator of tropics and midlatitudes no longer applies. During this final stage most of the cool air is of local origin, while low-level southerlies persist, driving long-range transport of tropical air masses from South Asia.

Figure 11c thus indicates that the EASMF during *Mei Yu* is characterised by the interaction between extratropical and tropical air masses. To outline the properties of those airstreams we look at Figure 12, containing time profiles of the trajectories considered. The shadings of these trajectories depend on their specific humidity, also shown in Figure 12b. The cool airstream has a substantially lower moisture content than the warm one throughout its travelling time. Therefore, the cool airstream is always identified by blue trajectories, while the warm airstream is constituted by the red ones. A first key difference between the two airstreams is found in their travelling height (Figure 12a). The warm air travels close to the 1000 hPa pressure level

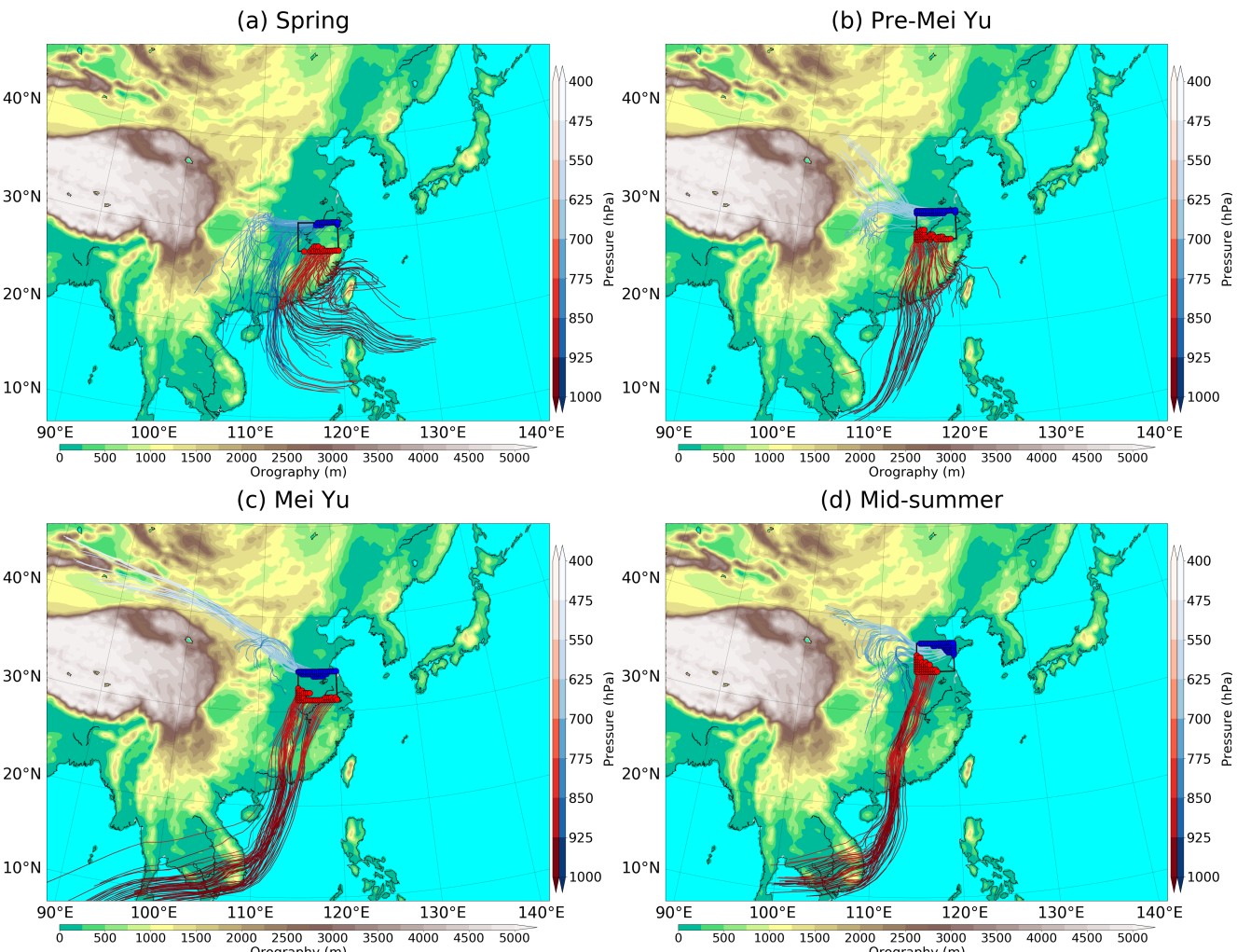

**Figure 11.** Warm (red) and cool (blue) Lagrangian trajectories starting at 0600 UTC on (a) 07 May (b) 07 June, (c) 27 June, (d) 17 July of the 1979-2018 climatology (Section 2.3), computed backward for 168 hours. The black quadrilateral indicates the selection domain (see text for more details on the selection process). Colour shading indicates the pressure of trajectories at each position and red/blue dots within the small domain indicate the starting points of the selected trajectories. The dekads representing the four stages of EASM progression ($1^{st}$ dk May, $1^{st}$ dk June, $3^{rd}$ dk June, $2^{nd}$ dk July, respectively) are selected according to Figure 3. Data from ERA5, 1979-2018 climatology.

before arriving over China, where (probably forced by the southern China terrain) it rises up to around 900 hPa. Conversely, the cool airstream is mainly seen descending gradually from mid-tropospheric heights, impacting the front at around 700-750 hPa.

In addition to pressure and moisture content, the thermodynamic properties of the two airstreams are also markedly different throughout their evolution (Figures 12c and 12d). $\theta_e$ differs of more than 20 K between the cores of the two airstreams for most

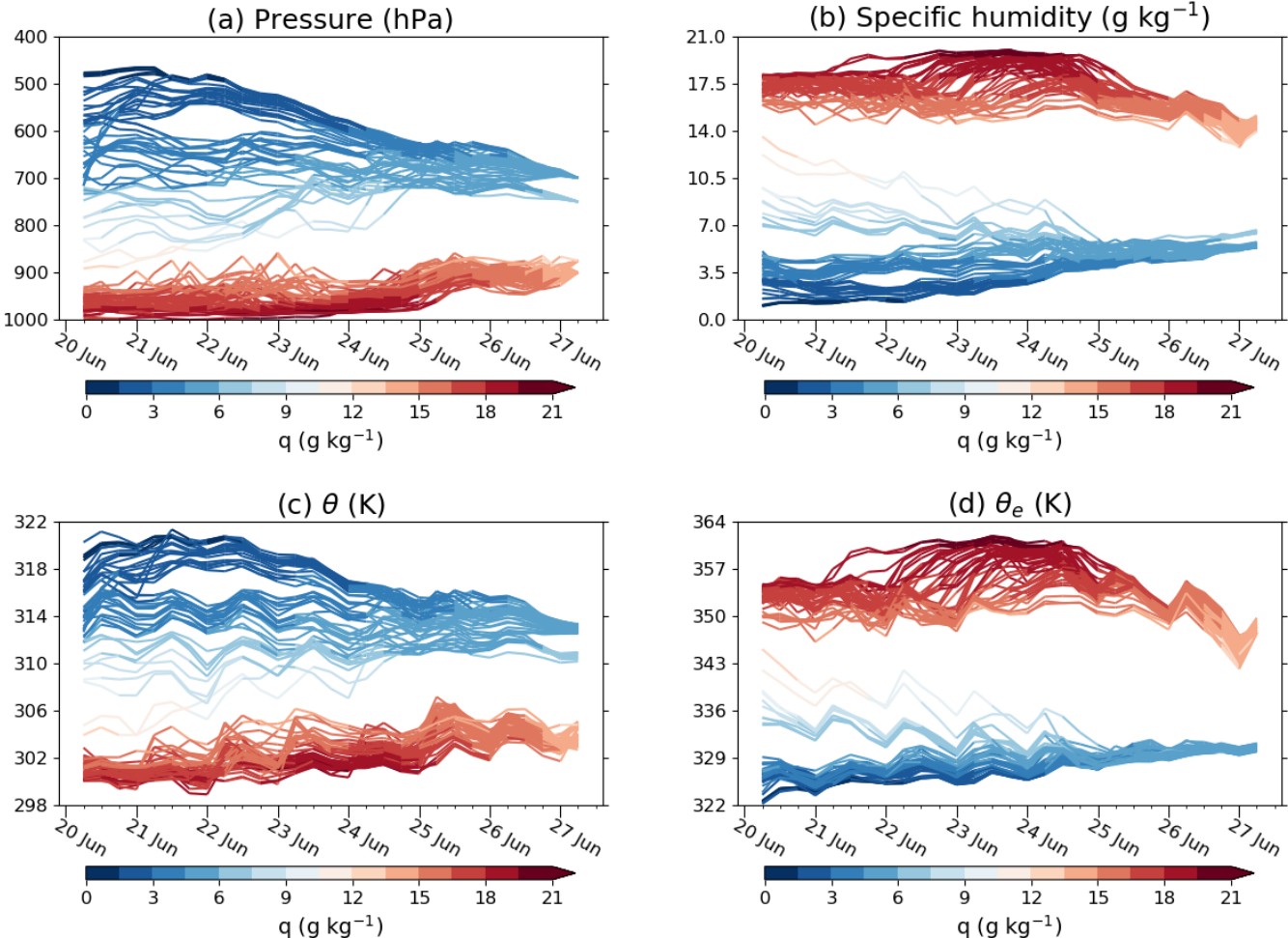

**Figure 12.** Time profiles of (a) pressure, (b) specific humidity, (c) potential temperature and (d) equivalent potential temperature for the *Mei Yu* trajectories (see relevant map in Figure 11c). Colour shading indicates specific humidity. Data from ERA5, 1979-2018 climatology.

of their history, apart from the very final stages, confirming the contrasting nature of the air masses they originate from. At the same time, it needs to be noted that $\theta$ values are higher for the cool airstream than for the warm, highlighting the misleading character of this quantity if used alone to identify air masses with very different moisture contents. The joint analysis of specific humidity, dry and equivalent potential temperatures for the warm airstream suggests the occurrence of condensation during the slow ascent (and/or precipitation from it) together with some mixing with preexisting lower-$\theta_e$ air, given the gradual decrease in specific humidity and in $\theta_e$ with little change in dry potential temperature (possibly due to cancelling effect between the two processes). All these quantities display slow and steady changes for the core of the cool airstream, suggesting the occurrence of a rather undisturbed descending flow towards the frontal zone. It is also important to note that the moisture content of the warm air is fairly constant before its final decrease, indicating the absence of a net moisture increase in 'near' regions (e.g.,

the South China Sea, where possible further moisture intakes are being roughly balanced by the rain present in the region), and therefore pointing at the role of the South Asian monsoon in providing the initial moisture.

# 5 Dynamics of variability in EASMF progression during the *Mei Yu* stage

From the analysis thus far it is clear that most of the northward migration of the EASMF occurs during the *Mei Yu* stage, which is also when the clearest tropical - extratropical interaction takes place. In this section, we investigate the key dynamics behind the variability in the progression of the EASMF during *Mei Yu*, and illustrate the associated changes in circulation and precipitation patterns. To do so, we examine the differences between the composites generated from two subsets of the 40-year ERA5 climatology, based on the latitude of the EASMF during *Mei Yu*. The two subsets, *high-lat* years and *low-lat* years, contain the 10 years with highest and lowest average EASMF latitudes, respectively, in the $3^{rd}$ dekad of June over eastern China (110°E-120°E).

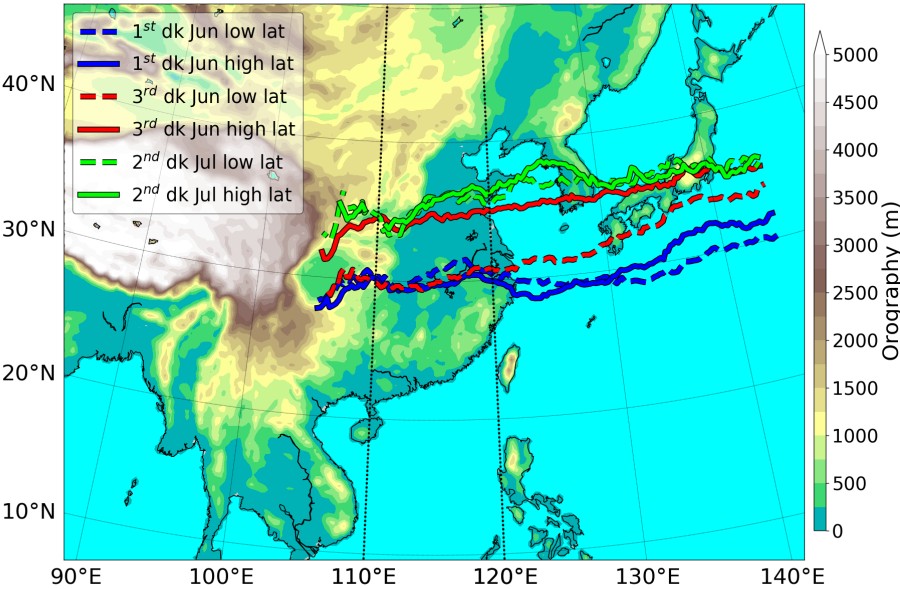

**Figure 13.** Dekadal mean location of EASM frontal band at 850 hPa. As in Figure 2, but for *high-lat* and *low-lat* years and only for selected dekads (see legend).

Figure 13 compares the position of the EASMF for the two composites during the $1^{st}$ dekad of June, the $3^{rd}$ dekad of June, and the $2^{nd}$ dekad of July, dekads representative of climatological *pre-Mei Yu*, *Mei Yu* and *mid-summer* stages, respectively. The map shows that the latitude of the front differs by around 4° to 5° between the two climatological *Mei Yu* composites ($3^{rd}$ dekad of June, red lines), while frontal latitudes are very similar before and after. This points at a lack of clear dependence between the latitude of the EASMF during climatological *Mei Yu* and its position before and after this stage. The similar position (at least over land) in *low-lat* years between $1^{st}$ and $3^{rd}$ dekads of June is suggestive of an initially more stationary

front (Chiang et al., 2017). In fact, this behaviour indicates that in *high-lat* years in which the climatological *Mei Yu* northward migration of the front has already taken place by the end of June, while in *low-lat* years the migration has not yet started by then.

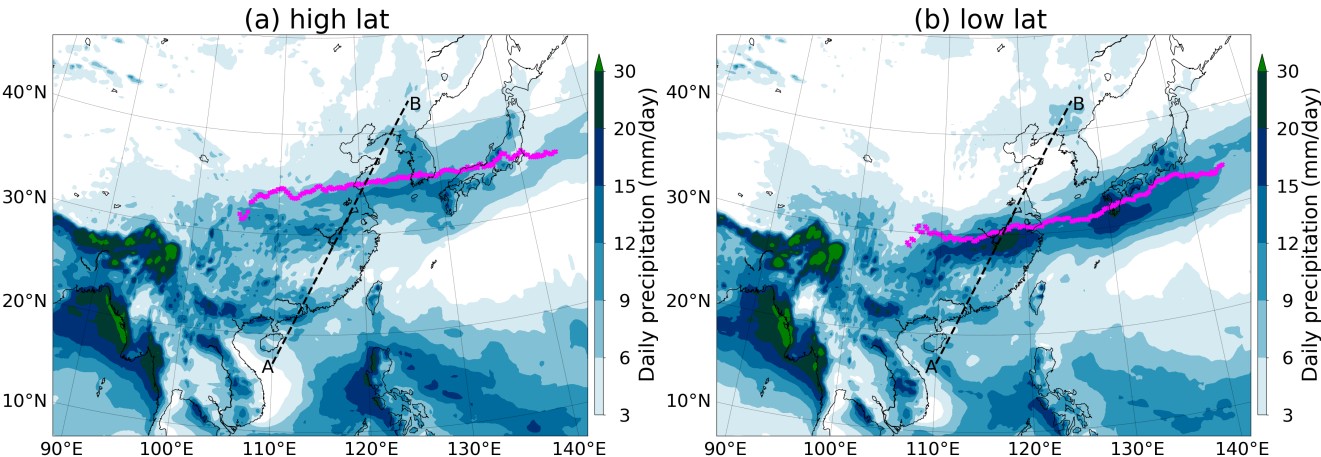

**Figure 14.** Dekadal mean of daily precipitation and of the location of the EASMF at *Mei Yu* stage ($3^{rd}$ dekad of June). As in Figure 4c, but for (a) *high-lat* years and (b) *low-lat* years.

The precipitation patterns during the climatological *Mei Yu* period ($3^{rd}$ dekad of June) associated with the two subsets are presented in Figure 14. The latitudinal variation of front location is accompanied by a migration of the rainfall band, which in both subsets has the front on its northern side. Higher precipitation maxima are observed in *low-lat* years, with a narrow band displaying values exceeding 15 mm day$^{-1}$, between 28°N and 30°N over eastern China, and a smaller region exceeding 20 mm day$^{-1}$ over the lower reaches of the Yangtze river. Instead, in the *high-lat* years the rain band observed is broader,

located mainly north of 30°N and less intense, with values predominantly below 12 mm day$^{-1}$. Hovmöller plots analogous to that in Figure 3 (not shown), indicate that the narrower, more intense and more southerly located band visible in the *low-lat* years is associated with a delay in the northward progression of rainfall (and frontal line). The *low-lat* years rainfall band is also connected to the area of high rainfall values (> 9 mm day$^{-1}$) covering south-eastern China. This is in contrast with the rainfall pattern of the *high-lat* years, showing a low-precipitation area visible over southern China. This latter rainfall pattern

is characterised by rainfall anomalies over central eastern China and Japan (high values) being out of phase with those over northeastern and southeastern China (low values). The pattern resembles one phase of the 'tripole mode' (Hsu and Lin, 2007). Chiang et al. (2017) used observational data to show that the tripole mode results from a significantly earlier *Mei Yu* termination, leading to shorter *Mei Yu* and longer *mid-summer* stage durations. This is consistent with the results in Figure 14, showing the tripole pattern in high-lat years, in which the termination of the *Mei Yu* stage has already taken place by the end of June.

Chiang et al. (2017) also pointed at the link between variations in these rainfall patterns and changes in the poleward migration of the westerlies across the TP. The relationship between the EASMF latitude and the STWJ configuration is investigated in

the remainder of this section, where the key differences between the two climatological *Mei Yu* composites are analysed, in terms of flow and dynamics, at lower and upper levels and on both sides of the front.

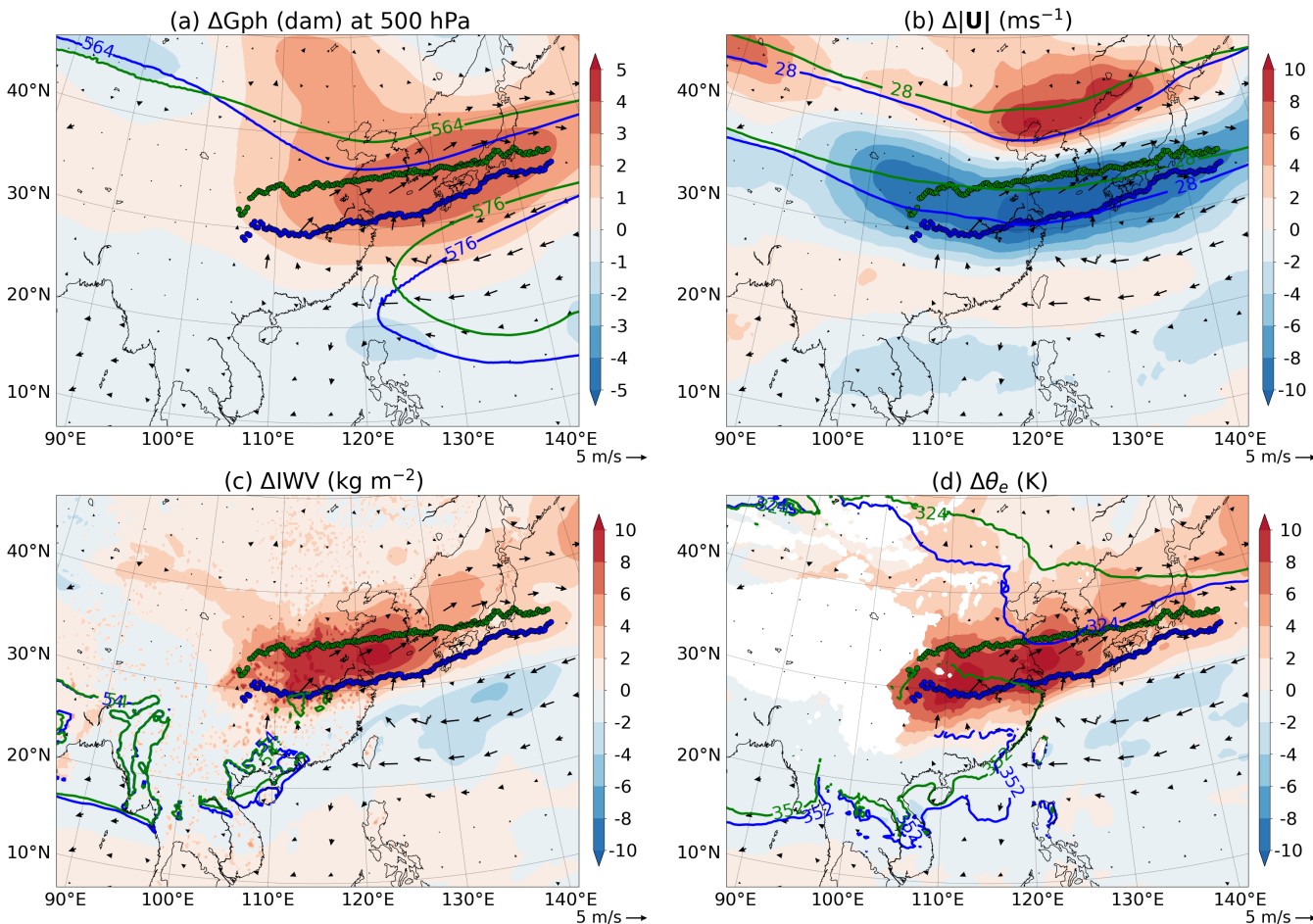

**Figure 15.** *high-lat* years − *low-lat* years anomalies of wind vectors at 850 hPa (arrows, m s$^{-1}$) and of: (a) geopotential height at 500 hPa (shading, dam), (b) wind speed at 200 hPa (shading, m s$^{-1}$), (c) 500-1000 hPa IWV (shading, kg m$^{-2}$), (d) equivalent potential temperature at 850 hPa (shading, K). Selected values of *high-lat* and *low-lat* years composites are indicated separately by green and blue contours, respectively. The contours displayed, indicating the same field the anomalies refer to in each panel, are: (a) 564 dam and 576 dam, (b) 28 m s$^{-1}$, (c) 54 kg m$^{-2}$, (d) 324 K at 700 hPa and 352 K at 925 hPa. The EASMF is indicated in all panels by bold green and blue lines for *high-lat* years and *low-lat* years, respectively. All panels refer to the $3^{rd}$ dekad of June, selected in Figure 3 to represent the *Mei Yu* stage. All values are masked out at levels below the ground. Data from ERA5, 1979-2018 climatology.

Figure 15 highlights the main changes in the ingredients of EASM seasonal progression (during the climatological *Mei Yu* stage) between the two composites, by showing anomalies of relevant quantities (*high-lat* years − *low-lat* years), with frontal locations and anomalous low-level wind vectors also indicated in all panels. The frontal displacement between the composites

over eastern China is accompanied by anticyclonic anomalous flow, visible between China and Japan. The Yangtze river region lies to the western side of this circulation. As a result, that region experiences a more pronounced southerly component of the low-level flow towards the EASMF in *high-lat* years.

The low-level circulation differences between the two composites are associated with changes in the upper-level flow, as indicated by the anomalies in geopotential height at 500 hPa, shown in Figure 15a, with a positive region centred between eastern China and Japan. This anomaly is linked to a different location of the WNPSH, shifted to the north (see the 576 dam contour) in the *high-lat* years, together with a less pronounced midlatitude trough see 564 dam. This difference in the upper-level midlatitude flow is further highlighted in Figure 15b, which shows a more pronounced STWJ trough over East Asia in

*low-lat* years see 28 m s$^{-1}$ contours, with the jet core moving further south and intensifying, as reflected by the dipole in the wind speed anomalies. The northward shift in the location of the WNPSH in *high-lat* years is similar to that displayed by the EASMF. This is consistent with the results presented in Section 3.3, that show how a frontogenetic flow is favoured on the northern side of the WNPSH.

A SE-NW oriented dipole centred at the latitude of the *low-lat* years front is visible in the IWV anomalies (Figure 15c). This

is a consequence of the difference in frontal location between the two composites and of the consequent low-level circulation, as the moist air on the southern side of the front reaches higher latitudes over eastern China in *high-lat* years. This panel also highlights that maximum IWV values beyond 54 kg m$^{-2}$ are locally present at a considerably higher latitude in *high-lat* years, near the Yangtze river region. This is consistent with the stronger southerly flow in *high-lat* years, that advects moist and warm air poleward from the tropics, all the way to the EASMF. Figure 15d contains a $\theta_e$ dipole that is very similar to the IWV one

in Figure 15c, also indicating that moist and warm (i.e., high-$\theta_e$) air travels further north in *high-lat* years. The contour lines in Figure 15d indicate $\theta_e$ values representative of the advection of southerly warm and northerly cool air towards the front; 324 K at 700 hPa and 352 K at 925 hPa, respectively. The pressure levels of those two $\theta_e$ contours differ from each other, and from the level at which $\theta_e$ anomalies are computed (850 hPa), because they refer to the cores of the related airstreams. These contours confirm the larger northward extension of warm air in *high-lat* years (see the 352 K contour extending beyond 30° N)

and also highlight a more intrusive southward advection of cool air in *low-lat* years (see the 324 K contour extending down to 35° N).

In summary, Figure 15 illustrates the key differences between the two composites, highlighting in *high-lat* years the enhanced poleward advection of moisture and high-$\theta_e$ air at low-levels, associated with a low-level anticyclonic anomaly and the northerly displacement of the WNPSH, and in *low-lat* years the more intrusive cool advection on the northern side of the front,

associated with a more pronounced midlatitude upper-level through.

The cross sections in Figure 16 show the vertical structure of the atmosphere, illustrating some of the differences between composites contained in Figures 13-15 and highlighting their links. In particular, the northward displacement (*high-lat* years vs *low-lat* years) is evident for the frontal structure, as well as for both the low-level airstreams impinging on its sides (see the bold 325 K and 350 K $\theta_e$ contours). The cool airstream, centred at around 700 hPa, extends considerably further south in *low-lat*

620    years, reaching 35°N. Conversely, in *high-lat* years it is the southerly warm advection of moist boundary-layer air (see the high values of moisture flux) that shows a larger extension, with its northern edge close to 35°N. Focusing on the upper levels,

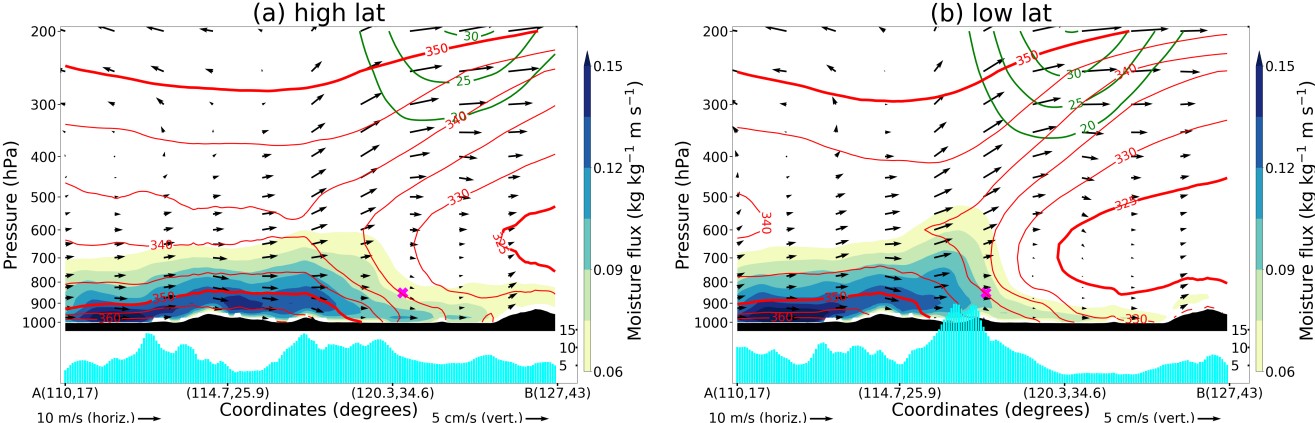

**Figure 16.** Cross-sections of dekadal mean moisture flux, equivalent potential temperature, wind speed, wind vectors and precipitation at *Mei Yu* stage ($3^{rd}$ dekad of June). As in Figure 10c, but for (a) *high-lat* years and (b) *low-lat* years, without 2-PVU lines and with bold 325 K and 350 K $\theta_e$ contours to emphasise the shifting of the thermodynamic structure.

Figure 16 indicates that the meridional displacement already described for the low levels applies also to the STWJ, consistent with the shift of the whole tropospheric frontal structure. The patterns of rainfall along the transect highlight the differences in shape and intensity between the two composites shown by Figure 14, with a narrower, more intense and southerly displaced band of frontal rainfall in *low-lat* years, consequence of the meridional displacement in frontal structure and associated air masses.

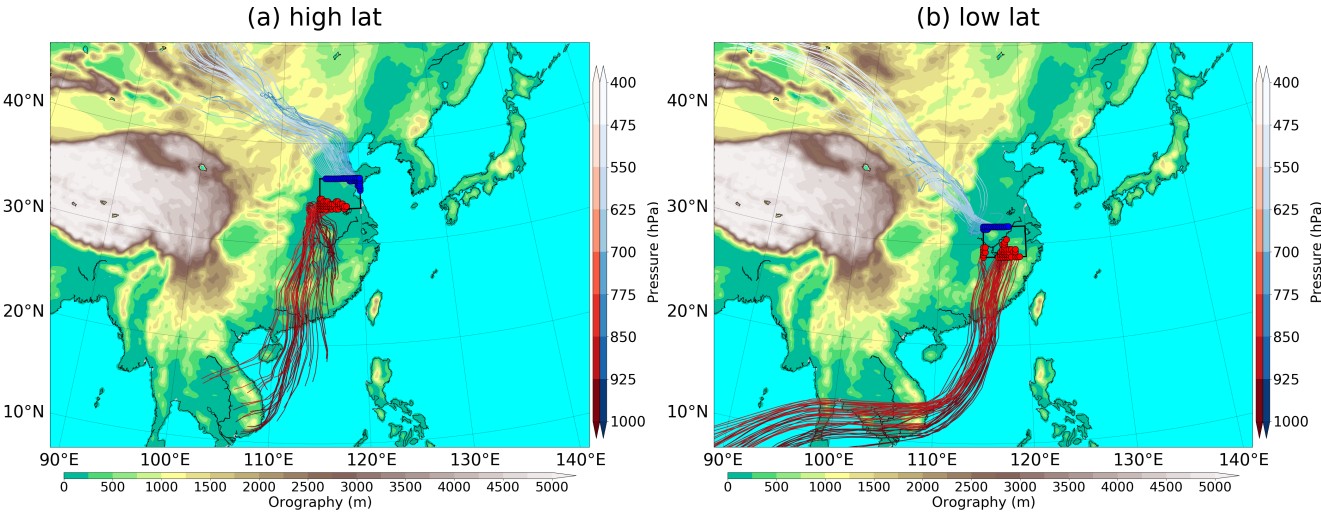

**Figure 17.** Warm (red) and cool (blue) Lagrangian trajectories for the Mei Yu stage. As in Figure 11c, but for (a) *high-lat* years and (b) *low-lat* years. The selection domain is constructed as in Figure 11, with mean frontal latitudes equal to (a) 35°N and (b) 30.25°N.

The trajectories shown in Figure 17 confirm the latitudinal displacement of the EASMF between *high-lat* and *low-lat* years, implying a displacement of the region where the warm and cool airstreams converge, and associated with an overall displacement of the paths of those airstreams. In both composites, the cool airstream descends from mid-tropospheric levels to around 700 hPa when reaching the front. While in *low-lat* years this airstream travels close to the northern edge of the TP, it follows a more north-eastern trajectory in *high-lat* years. Differences between the two composites can be seen also in the path of the warm airstream, that travels northwards beyond the Yangtze river in *high-lat* years, while stopping at around 30°N in *low-lat* years. In *low-lat* years, the flow from the Indian Ocean is more sustained, with the warm airstream displaying a longer and more coherent path towards the front, while a shorter path, with more influence of locally residing air is observed in *high-lat* years. This means that in *low-lat* years the EASMF sees air travelling from both the Equatorial Indian Ocean and continental Asia impacting on each side of the front, resulting in a more direct interaction between deep tropics and midlatitudes. This difference is likely associated with the changes in regional low-level flow resulting from the displacement of the WNPSH, as suggested by the weak anticyclonic gyre of the warm air over China in *high-lat* years (see contours and wind arrows in Figure 15a).

## 6 Conclusions

In this study we presented a comprehensive analysis of the seasonal progression and variability of the EASM, with particular focus on its front and on the interaction of tropical and extratropical air masses. The different factors behind the complex, multi-scale and multi-stage progression of the EASM have been the object of a considerable number of studies, with the main results outlined in Section 1. The interaction between the different dynamical mechanisms, and in particular between tropical and extratropical air masses, still presents open questions. Therefore, using the front- and airmass-centred approach proposed by Parker et al. (2016) for the Indian summer monsoon and adapting it to the EASM, we investigated the interaction between competing air masses as a key factor shaping monsoon progression, with additional focus on *Mei Yu*, the primary stage of EASM progression.

A frontal detection algorithm was developed and used to identify the EASMF. Using a front-centred perspective, we focused on the migration and evolution of the EASM frontal structure throughout the stages of EASM progression. The results of this analysis emphasise the role of the STWJ, whose strength and location are closely associated with that of the EASMF. The STWJ shows a climatological trough over east Asia, and its southern edge and jet-streak right entrance region are collocated with the EASMF as the front progresses poleward. Therefore, the EASMF benefits from upper-level divergence, particularly during the *Mei Yu* stage. Frontogenetic flow is favoured on the northern side of the WNPSH, hence also playing a key role in the progression of the EASMF. These forcings act in conjunction with the low-level southerly advection of moist and warm air. This moist flow, modulated by the seasonal cycle of the Asian monsoon, extends from the tropics up to the Yangtze river and the EASMF.

A clear result of this study is that the interaction between tropical and extratropical air masses converging at the EASMF is especially pronounced during the *Mei Yu* stage. The two air masses are substantially different. The warm and moist air is

of South Asian origin, from where it gets the moisture that is advected at low levels (900-950 hPa) towards the front. Remote moisture sources (e.g., Equatorial Indian Ocean) play a role in providing initial moisture to this airstream, with low-level air just south of the EASMF displaying high values of specific humidity, typical of the South Asian monsoon. The cool air is instead dry and of continental origin; it descends from mid-levels down to 700-800 hPa when travelling north of the TP, before impinging onto the front from the northwest. This tropical-extratropical interaction is not as visible during the other stages of EASM progression, in which air converging at both sides of the front is shown to be mainly of tropical origin. *Mei Yu*, the stage of most rapid poleward migration of the front, is thus characterised by the competition between tropical and extratropical airstreams.

The role played by the STWJ in the interannual variability of EASM northward migration and the associated rainfall patterns is emphasised by the analysis of composites of years with high or low latitude of the EASMF during the climatological *Mei Yu* stage. In detail, the poleward displacement of the EASMF during climatological *Mei Yu* is accompanied by a less evident upper-level trough in the region, along with a northerly displaced WNPSH. This leads to enhanced moisture transport up to the Yangtze river. Conversely, a southward-displaced front, together with a more pronounced upper-level trough and a southward-displaced WNPSH, is characterised by warm air originating further away in the tropics and a more intrusive cool airstream, leading to a more direct influence of warm air travelling from the Equatorial Indian Ocean as well as cool air from continental Asia. As a consequence, the rainfall pattern varies substantially between the two composites. In low-latitude years there is diffuse precipitation over south-eastern China, along with an intense narrow rain band just south of the front. In high-latitude years the pattern changes to a tripole and a less intense frontal band.

This study shows the validity, particularly in the *Mei Yu* stage, of analysing the EASM progression with a Parker et al. (2016)-like approach, a perspective focused on the air masses, tropical and extratropical, that converge at the monsoon front. There are fundamental differences between the Indian summer monsoon and the EASM, caused essentially by the different geographical location of the main actors of the two monsoon systems. However, analogies can be drawn. The progression of the Indian summer monsoon is modulated by the balance between moist low-level tropical advection and mid-level subtropical dry intrusion, in turn influenced by tropical modes and extratropical dynamics, respectively. For the EASM, the interaction between tropical and extratropical airstreams is particularly pronounced in the *Mei Yu* stage, when also most of the northward migration occurs. During this stage, the progression and strength of the EASM are influenced by the state of the STWJ and by the location of the WNPSH, steering the low- and mid-level airstreams that converge at the front and controlling their balance. Variability in the flow thus acts on top of the seasonal progression of the EASM. Figure 18 provides a schematic representation of the key factors of EASM progression during the *Mei Yu* stage. This framework can provide a useful basis for interpreting climate variability, including extreme events, as well as performing model evaluations and interrogating model predictions and projections. However, more research is needed to link the interaction between tropical and extratropical air masses with the smaller-scale transient weather features that constitute the EASMF itself, particularly during *Mei Yu*, and with the large-scale modes of flow variability.

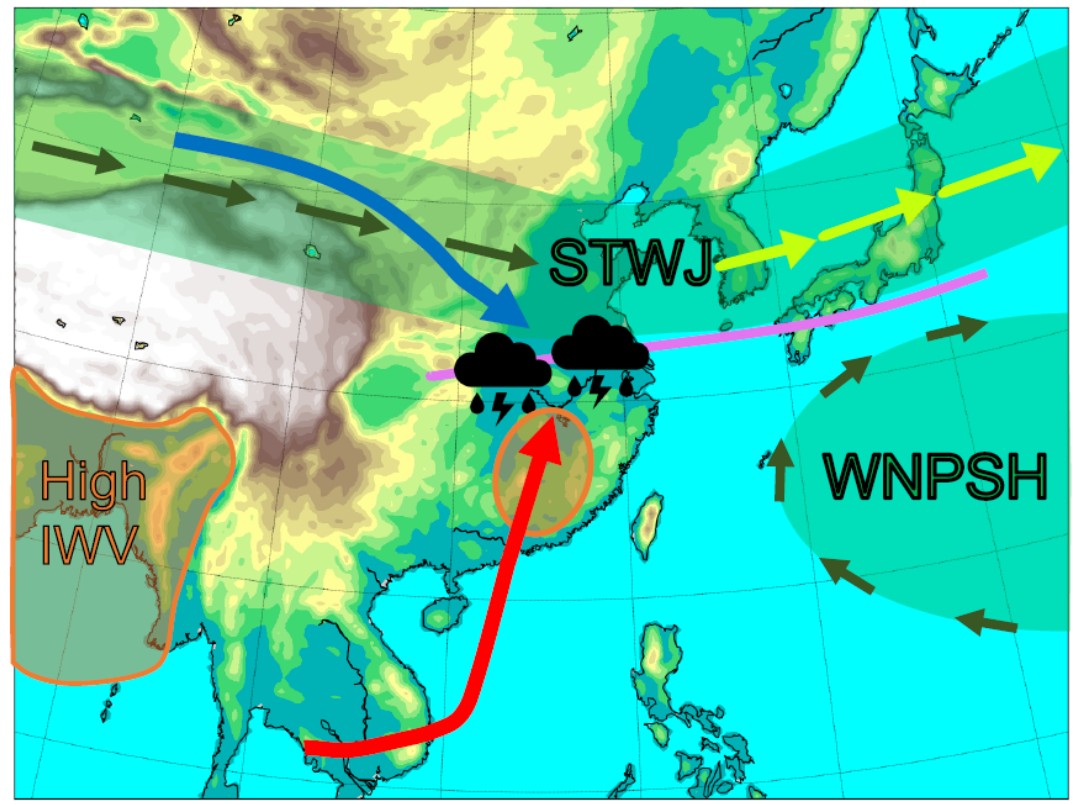

**Figure 18.** Schematic representation of the key factors of EASM progression during the *Mei Yu* stage. The cool (blue) and warm (red) low-level airstreams converge at the EASMF (magenta). Heavy precipitation thus occurs over central-eastern China, where high-moisture-content air is observed, up to values similar to those typical of the South Asian monsoon (orange). Regional flow features (green) control the balance of the airmass advection towards the EASMF, via the pattern and strength of the STWJ (jet streak in yellow) and the location of the WNPSH.

*Code availability.* The software code used for the analysis consists mainly in scripts written for use in a Python environment, normally loading and processing data with Iris (https://scitools-iris.readthedocs.io/en/stable/). It can be made available upon request. LAGRANTO can be donwloaded at https://iacweb.ethz.ch/staff/sprenger/lagranto/.

*Data availability.* The ERA5 reanalysis datasets are publicly available, upon registration, at https://climate.copernicus.eu/climate-reanalysis.

*Author contributions.* AV performed the data analysis and frequently discussed its design and conception with all other authors. AV wrote a full draft of the manuscript, that was then refined and improved by feedback, in various iterations, from other authors, primarily AGT and RS, and also PLV

*Competing interests.* The authors declare that they have no conflict of interest.

*Acknowledgements.* The authors were supported by the COSMIC project through the Met Office Climate Science for Service Partnership (CSSP) China as part of the Newton Fund, contract number P106301. The authors are grateful to an anonymous referee for the review of the original submission, Prof Michael Reeder for the review of the revised submission, an anonymous referee for the reviews of both submissions, Prof David Schultz for his comments, and Dr Shira Raveh-Rubin for serving as co-editor. The authors wish to thank Dr Amulya
Chevuturi for the initial drafting of a frontal detection algorithm and for fruitful discussions. Thanks also to Dr Leo Saffin and Dr Ben Harvey for providing the necessary software for the use of LAGRANTO in a Python environment and with Iris cubes , and to Prof Peter Clark for a Python function used to compute $\theta_e$.

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
