# Peer review of "Characterising the interaction of tropical and extratropical air masses controlling East Asian summer monsoon progression using a novel frontal detection approach"

_Weather and Climate Dynamics, 2021_

## Referee Comment (RC1)

General comments

Unlike traditional monsoon that occurs in the tropics, the East Asian summer monsoon (EASM) is characterized as a subtropical monsoon, which is affected by both the tropical and mid-latitude dynamics. In this work, the authors introduce some methodologies to investigate role of the monsoonal southerlies and the extratropical northerlies in the seasonal progression of the EASM, with a focus on the Mei Yu stage. In particular, the authors employ a frontal detection algorithm that's based on the 850 hPa equivalent potential temperature gradient to identify the front position at 6-hourly frequency. They also introduce a Lagrangian method that illustrates backward trajectories of air masses arriving at East Asia during the seasonal march of the monsoonal rainfall. The authors show that the Mei-Yu stage is unique in that the Lagrangian method shows well separated air mass origin during this stage – cold extratropical air and warm tropical air converging in East Asia and form the Mei-Yu front. By comparing the high-lat years and low-lat years based on the front latitude during the climatological Mei Yu period (3$^{rd}$ dekad of June), the authors show that the extratropical northerlies are crucial in affecting the location and intensity of the Mei Yu front.

This manuscript is overall well-written, and the results are presented in a structured and clear style. The methodologies used in this study provide new angles to the EASM community, especially for studies on extreme events and rainfall variability at high-frequency timescales. What I like most about this paper is the authors provide a comprehensive analysis of key physical factors during the EASM rainfall stages, including atmospheric circulations (such as the subtropical jet and the western Pacific subtropical high), moisture transport, and origin of air masses. That said, I do find some analysis and interpretation a bit confusing or even misleading (see the specific comments for details). In summary, I think this paper could be a potential publication in WCD once the authors address these comments.

Specific comments/major comments

1.  My biggest concern is that I would expect to see more in-depth discussion on the physical mechanisms based on the current title. However, the paper overall is descriptive and lack mechanistic insights. For example, although the authors did show the different air mass origins during each rainfall stages and compared the tropical flow vs. extratropical flow between high-lat and low-lat cases, the authors did not articulate how their results differ from previous findings on mechanism-level. In particular, Chiang et al. (2020) employed modeling approach and provided detailed discussion on interaction of tropical and extratropical flows on the seasonality of the EASM (their section 5 is specifically focused on this). Similarly, Figure 16 in this study looks quite similar to Figure 15 from Sampe and Xie (2010), but I don't see what new insights we learn from it. Maybe the authors could elaborate more. I would like to see more discussion on what new lessons that we learn from this study compared to previous studies asking similar questions such as Chiang et al. (2020) and others.

2.  Section 2.2. Besides the pair of studies cited here (Li et al., 2018, 2019), Day et al. (2018) introduced an East Asian rainband detection algorithm based on precipitation, Dai et al. (2021) used the self-organizing map to define the seasonal rainfall stages in East Asia. And

there might be other methods out there in the literature. To provide more context, it is worthwhile briefly summarizing the algorithms that have been introduced in the literature on the EASM front detection, their pros and cons, and why the algorithm adopted in this study is an ideal approach. Further, could the authors introduce their algorithm in a more structured way in Lines 127-147, such as using numerated bullet points and maybe adding sub-title to illustrate the purpose for each point/step? It would also be helpful to show snapshots of the front detection at 6-hourly frequency for a "true" front case and a "false" front case. These efforts will help the readers to better understand the method.

3. Section 3.1. I'm a bit confused how the front progression shown in Figure 1 is produced. Are the front positions the average of the 6-hourly frontal detection from each dk? Or are they the results of applying the algorithm to the dk averaged theta_e gradient? It's unclear to me how the results from these two approaches would differ. Could the authors clarify?

4. Section 3.2. I agree that the four dekads are representative for the four stages, but how the four rainfall stages were decided at the first place? Is that based on the timing of these stages suggested from previous literature? Please clarify.

5. Section 4.1. Could the authors elaborate on the reasoning of analyzing the vertical cross-section along A-B? I would expect to see more evident signal from the extratropical flow, but they are absent in Figure 8. Supporting evidence is missing for the statement such as in Line 345 that "The advection of cool sub-tropical air is centred around 700-800 hPa"; in Line 364 that "This is in agreement with Figure 8c, displaying these different air masses impacting on the front from opposite sides." I agree that the theta_e contour clearly shows the warm versus cold air, but the wind pattern in the extratropics is too weak to support cold air advection. Do the results change if the authors analyze the cross-section along eastern China, say 105E, instead?

6. Lines 347-349 and Figure 9. Should we expect the position of the frontal position for each stage in Figure 9 match those shown in Figure 3? The spring, in particular, does not seem so. The spring front in Figure 9 is evidently further south compared to Figure 3.

7. Section 5. The comparison between high-lat and low-lat cases is interesting, but the wording here is misleading. The authors selected the high-lat and low-lat cases based on the front latitude during the climatological Mei-Yu period, i.e. the 3$^{rd}$ dekad of June. However, it is very likely that the seasonal evolution of the EASM front is different between the high-lat and low-lat cases (Chiang et al., 2017). Therefore, when describing the high-lat and low-lat results, it might not be accurate to directly refer that as the Mei-Yu in the high-lat and the low-lat. Specific comments related to section 5 are listed below.

   1) Line 393. Suggest rephrasing that these dekads refer to the climatological rainfall stages. Similarly, in Line 401, it might be more accurate to say "during the climatological Mei Yu period" or simply say "The precipitation patterns during the 3$^{rd}$ dekad of June." And in Line 408, suggest replacing "during the Mei Yu stage" with "during the climatological Mei Yu stage" or "during the 3$^{rd}$ dekad of June."

   2) Lines 396-400. A bit concerned about the statement on whether Figure 11 tells us the dependence between the rainfall stages from both cases. An alternative interpretation could be that, the fact that the blue dashed line and the red dashed line stay at the similar latitudes suggests a more stationary front from the low-lat

cases; in other words, the low-lat cases reflect years with longer Mei Yu (Chiang et al., 2017).

3) To further elaborate the point raised in Lines 398-400. It would be helpful to present the high-lat and low-lat versions of Figure 2 (the precipitation hovmoller diagram) in the paper. This will more clearly show the seasonal evolution of the EASM front from the two cases. The authors could also consider expanding their analysis on the STWJ, WPSH, and air mass origins to the four rainfall stages from both cases. This may provide a more complete picture on role of the tropical vs. extratropical flows on the progression of EASM front.

4) Line 410. Again, it is misleading to directly refer the pattern to the 'tripole', because the latter is captured by the leading mode of East Asian summer rainfall. It is more appropriate to say that the rainfall pattern resemble one phase of the tripole mode.

5) Line 413. To clarify, it is more precise to say that the termination of the Mei Yu has already taken place by the end of June.

6) Figures 13-15 suggest that high-lat cases are associated with stronger moisture transport from southerly flow, while the front in the high-lat cases presents a more northward progression and produces less intense rainfall. My take from these results is that the comparison between high-lat and low-lat highlight role of the extratropical air mass in affecting both front latitude (with strong extratropical air → southward or more stationary Mei Yu front) and rainfall intensity (with strong extratropical air → stronger convergence with the southerlies and thus more intense rainfall). The reason why I raise this is I think this is a good example to illustrate the authors original motivation: to figure out how the tropical and extratropical air control the EASM front movement. Further, I would interpret Figure 15 in a different way, compared to what the authors stated in Line 460-462. A more generalized lesson from the comparison between high-lat and low-lat is that it is the strength of the extratropical northerlies that determine the evolution of the front. Thus, compared to the tropical component, the extratropical component is more fundamental (or dominant) in the seasonal evolution, and when the extratropical component is weak, we see the high-lat patterns. No matter whether the authors agree or disagree with these points, I think it would be worth elaborating on what we learn about role of tropical vs. extratropical air from the comparison between the high-lat and the low-lat cases.

8. The authors argue for a crucial role of the South Asian Monsoon in influencing the EASM but did not provide specific references and underlying mechanisms. For example, for the introduction of the tropical influence on the EASM (Line 33 and Lines 41-49), could the authors elaborate more on the effect of the onset and progression of the South Asian monsoon? I find this part is vague and unsure through what physical processes the South Asian monsoon affects the EASM. Further, it is not convincing that the results presented in the current paper suggest the Bay of Bengal, instead of South China Sea, is the primary moisture source for eastern China. The moisture content of the southerly flow is relatively constant might be due to that air parcels both precipitates along the trajectory and picks up moisture from the warm moist ocean surface over the South China Sea. In this case,

there is indeed contribution from the South China Sea, it is just the total amount of water content keep unchanged.

Technical corrections/minor comments

1. End of Line 29. Suggest the authors replace "a large elevated landmass" with "TP", which is more precise and concise in this context.
2. End of Line 32. Add references about the spring rainfall stage, for example, (Wu et al., 2007) and maybe others.
3. Line 38. Add reference on using the gradients of equivalent potential temperature to denote the front, for example, (Tomita et al., 2011). Also clarify whether and how the equivalent potential temperature is calculated, or whether it is directly downloaded from the ERA5 website.
4. Lines 41-49. Convection over western tropical Pacific and related "Pacific-Japan" pattern should not be ignored when considering tropical influence.
5. Line 66. This sentence reads odd. Suggest the authors rephrase the second half of the sentence to something like "..., that the EASM rapidly weakens and mid-summer rainfall dominates northeastern China."
6. Lines 73-75. The mentioning of the "tripole mode" here is a bit misleading. I think it is worth citing the original reference of the tripole mode (Hsu and Lin, 2007) and clarify that the tripole mode was originally referred to the leading mode of East Asian summer rainfall at interannual timescale. Further, it is worth clarifying that Chiang et al. (2017) showed that rainfall anomalies during years with earlier Mei-Yu termination mimic one phase (i.e. more rainfall in northeastern China and southeastern China while less rainfall in central eastern China) of the tripole mode, not results in the tripole mode.
7. Lines 100-109. Could the authors rephrase or reorganize this part? It would make it easier for the readers to follow if the authors can explicitly explain what each section of sections 3-5 focuses on.
8. Line 113. Suggest changing the resolution to $0.25° \times 0.25°$.
9. Section 2.1. The paper by Tarek et al. (2020) specifically focused on its performance in North America. More discussion is needed on whether ERA5 dataset is a more useful or suitable source (compared to ERA-Interim) for studying the EASM.
10. Line 125. Clarify what does "n" refer to. And it should be "The algorithm used in Li et al. retains ..."?
11. Section 2.4. Add units for IVT, IWV and the variables used for both calculations.
12. Line 202. The word "extending" is misleading, which may imply that there is southward rainfall migration when the pre-Meiyu stage starts. However, the rainfall appearing in the South China Sea is just a reflection of enhanced convection during that period.
13. The unit in Figure 3 caption should be mm/day?
14. Lines 296-297. More specifically and directly, it is the extratropical northerlies that prevent the southerly flow to march northward, and the strong extratropical northerlies have been argued to arise from the mechanical forcing from the TP on the SWTJ (Chen and Bordoni, 2014; Kong and Chiang, 2019).

15. Line 342. Rephrase "opposite sides" to "the tropical and the extratropical side." The original wording is vague.
16. Line 466. Suggest adding "(the green contour in Figure 13)" at the end of the sentence.
17. Suggest adding a sub-plot showing low-lat minus high-lat in Figure 14. This will help to highlight the circulation difference in the extratropics.
18. Lines 503-510. I did not understand why the authors tried to draw analogies between the Indian summer monsoon and the EASM. Was the motivation to show that it is valid to apply the method, which was applied to the Indian Summer Monsoon in (Parker et al., 2016), to the EASM? If so, it might be more appropriate to put this part to Line 100.

**References**

Chen, J., and Bordoni, S. (2014). Orographic Effects of the Tibetan Plateau on the East Asian Summer Monsoon: An Energetic Perspective. J. Climate *27*, 3052–3072.

Chiang, J.C.H., Swenson, L.M., and Kong, W. (2017). Role of seasonal transitions and the westerlies in the interannual variability of the East Asian summer monsoon precipitation. Geophys. Res. Lett. *44*, 2017GL072739.

Chiang, J.C.H., Kong, W., Wu, C.H., and Battisti, D.S. (2020). Origins of East Asian Summer Monsoon Seasonality. Journal of Climate *33*, 7945–7965.

Dai, L., Cheng, T.F., and Lu, M. (2021). Define East Asian Monsoon Annual Cycle via a Self-Organizing Map-Based Approach. Geophysical Research Letters *48*, e2020GL089542.

Day, J.A., Fung, I., and Liu, W. (2018). Changing character of rainfall in eastern China, 1951–2007. PNAS 201715386.

Hsu, H.-H., and Lin, S.-M. (2007). Asymmetry of the Tripole Rainfall Pattern during the East Asian Summer. J. Climate *20*, 4443–4458.

Kong, W., and Chiang, J.C.H. (2019). Interaction of the Westerlies with the Tibetan Plateau in Determining the Mei-Yu Termination. J. Climate *33*, 339–363.

Li, Y., Deng, Y., Yang, S., and Zhang, H. (2018). Multi-scale temporospatial variability of the East Asian Meiyu-Baiu fronts: characterization with a suite of new objective indices. Clim Dyn *51*, 1659–1670.

Li, Y., Deng, Y., Yang, S., Zhang, H., Ming, Y., and Shen, Z. (2019). Multi-scale temporal-spatial variability of the East Asian summer monsoon frontal system: observation versus its representation in the GFDL HiRAM. Clim Dyn *52*, 6787–6798.

Parker, D.J., Willetts, P., Birch, C., Turner, A.G., Marsham, J.H., Taylor, C.M., Kolusu, S., and Martin, G.M. (2016). The interaction of moist convection and mid-level dry air in the advance of the onset of the Indian monsoon. Quarterly Journal of the Royal Meteorological Society *142*, 2256–2272.

Sampe, T., and Xie, S.-P. (2010). Large-Scale Dynamics of the Meiyu-Baiu Rainband: Environmental Forcing by the Westerly Jet*. Journal of Climate *23*, 113–134.

Tarek, M., Brissette, F.P., and Arsenault, R. (2020). Evaluation of the ERA5 reanalysis as a potential reference dataset for hydrological modelling over North America. Hydrology and Earth System Sciences *24*, 2527–2544.

Tomita, T., Yamaura, T., and Hashimoto, T. (2011). Interannual Variability of the Baiu Season near Japan Evaluated from the Equivalent Potential Temperature. Journal of the Meteorological Society of Japan. Ser. II *89*, 517–537.

Wu, G., Liu, Y., Zhang, Q., Duan, A., Wang, T., Wan, R., Liu, X., Li, W., Wang, Z., and Liang, X. (2007). The Influence of Mechanical and Thermal Forcing by the Tibetan Plateau on Asian Climate. J. Hydrometeor *8*, 770–789.

---

## Author Response (AR1)

**Response to reviews of "The interaction of tropical and extratropical air masses controlling East Asian summer monsoon progression" by Volonté et al.**

wcd-2021-12 - A. Volonté on behalf of all authors, 14 January 2022

Dear Editor,

We thank the two reviewers for their thorough reviews and for the positive and constructive comments on our manuscript. We also thank Prof Schultz for his detailed and helpful short comment. We believe that addressing these comments has given us the chance to improve our study. Key changes made to the original version of the manuscript include a redesigned front detection algorithm (with revised sections describing it step-by-step and presenting the relevant literature) and new diagnostics such as front deformation and horizontal wind divergence. Front location data are used in all figures in the results' sections. Therefore, figures 1-15 in the original manuscript have all been replaced by revised versions with new front data. Below we give point-to-point responses to each of these reviews in turn, replying also to the short comment. Our responses are written in *green italic* font, while all edits in the manuscript are in red.

**Reviewer 1**

**General comments**

Unlike traditional monsoon that occurs in the tropics, the East Asian summer monsoon (EASM) is characterized as a subtropical monsoon, which is affected by both the tropical and mid-latitude dynamics. In this work, the authors introduce some methodologies to investigate role of the monsoonal southerlies and the extratropical northerlies in the seasonal progression of the EASM, with a focus on the Mei Yu stage. In particular, the authors employ a frontal detection algorithm that's based on the 850 hPa equivalent potential temperature gradient to identify the front position at 6-hourly frequency. They also introduce a Lagrangian method that illustrates backward trajectories of air masses arriving at East Asia during the seasonal march of the monsoonal rainfall. The authors show that the Mei-Yu stage is unique in that the Lagrangian method shows well separated air mass origin during this stage – cold extratropical air and warm tropical air converging in East Asia and form the Mei-Yu front. By comparing the high-lat years and low-lat years based on the front latitude during the climatological Mei Yu period (3rd dekad of June), the authors show that the extratropical northerlies are crucial in affecting the location and intensity of the Mei Yu front. This manuscript is overall well-written, and the results are presented in a structured and clear style. The methodologies used in this study provide new angles to the EASM community, especially for studies on extreme events and rainfall variability at high-frequency timescales. What I like most about this paper is the authors provide a comprehensive analysis of key physical factors during the EASM rainfall stages, including atmospheric circulations (such as the subtropical jet and the western Pacific subtropical high), moisture transport, and origin of air masses. That said, I do find some analysis and interpretation a bit confusing or even misleading (see the specific comments for details). In summary, I think this paper could be a potential publication in WCD once the authors address these comments.
*Many thanks for this detailed analysis and for the positive judgement on our work. See below our replies to the specific comments.*

**Specific comments/major comments**

1. My biggest concern is that I would expect to see more in-depth discussion on the physical mechanisms based on the current title. However, the paper overall is descriptive and lack mechanistic insights. For example, although the authors did show the different air mass origins during each rainfall stages and compared the tropical flow vs. extratropical flow between high-lat and low-lat cases, the authors did not articulate how their results differ
from previous findings on mechanism-level. In particular, Chiang et al. (2020) employed modeling approach and provided detailed discussion on interaction of tropical and extratropical flows on the seasonality of the EASM (their section 5 is specifically focused on this). Similarly, Figure 16 in this study looks quite similar to Figure 15 from Sampe and Xie (2010), but I don't see what new insights we learn from it. Maybe the authors could elaborate more. I would like to see more discussion on what new lessons that we learn from this study compared to previous studies asking similar questions such as Chiang et al. (2020) and others.

*Thanks for this comment. We agree that Section 3, that provides an overview of the features and processes jointly driving the evolution of the EASM and of its front, could be regarded as the most descriptive in the manuscript. In its original version, this section was possibly too descriptive. However, in the revised version of the manuscript large parts of Section 3 have been rewritten, particularly in Sections 3.3, 3.4, and 3.5, as new analyses and diagnostics, such as front deformation and upper-level horizontal wind divergence, have been included following the various suggestions in the reviewers' comments. The discussion of EASMF progression variability (Section 5) has also been modified in parts, according to comment 7 from the reviewer and related sub-comments. We hope that the reviewers will agree that these additions have increased the mechanistic insight of our analysis. In addition to this, the front detection algorithm has been redesigned addressing reviewers' concerns (for more details see the reply to the next comment). Given that minor differences were observed in the EASM front progression, small changes have been made to the description of the results and in the conclusions, giving us the opportunity to improve the clarity and sharpness of the text (all edits in the manuscript in red). For what concerns the novelty of the results presented in the manuscript, the general comment opening this review acknowledges the introduction of Lagrangian methodologies showing, among other results, the uniqueness of the Mei Yu stage in terms of convergence of tropical-extratropical air masses. In the same comment, the reviewer acknowledges that this study provides new angles to the EASM community. Expanding on this, although we appreciate that parts of our analysis overlap to those in the studies referenced in this comment, we would argue that the combined Eulerian/Lagrangian perspective used, and the range of diagnostics applied (particularly in the revised manuscript) show the novelty of our study, setting it aside from previous literature.*

2. Section 2.2. Besides the pair of studies cited here (Li et al., 2018, 2019), Day et al. (2018) introduced an East Asian rainband detection algorithm based on precipitation, Dai et al. (2021) used the self-organizing map to define the seasonal rainfall stages in East Asia. And there might be other methods out there in the literature. To provide more context, it is
worthwhile briefly summarizing the algorithms that have been introduced in the literature on the EASM front detection, their pros and cons, and why the algorithm adopted in this study is an ideal approach. Further, could the authors introduce their algorithm in a more structured way in Lines 127-147, such as using numerated bullet points and maybe adding sub-title to illustrate the purpose for each point/step? It would also be helpful to show snapshots of the front detection at 6-hourly frequency for a "true" front case and a "false" front case. These efforts will help the readers to better understand the method.

*Many thanks for this comment. As it can be seen later in this document, concerns on the front detection algorithm are echoed by Reviewer 2 and, particularly, by Prof Schultz. We therefore*

*decided to redesign the algorithm, now based on the full horizontal gradient of equivalent potential temperature. Section 2.2 now contains an initial subsection that reviews the state-of-the-art literature on front identification and a following subsection that describes our revised algorithm, including examples of daily front maps. For the sake of brevity, we do not copy the new Section 2.2 in this document. While the main properties of EASM front progression outlined by the original algorithm are retained by this updated version, minor differences can be seen throughout all stages, and particularly in the earlier part of the season. Therefore, small changes have been made to the description of the results and in the conclusions, all indicated by red text.*

3. Section 3.1. I'm a bit confused how the front progression shown in Figure 1 is produced. Are the front positions the average of the 6-hourly frontal detection from each dk? Or are they the results of applying the algorithm to the dk averaged theta_e gradient? It's unclear to me how the results from these two approaches would differ. Could the authors clarify?
*The front identification algorithm is applied on daily averages of the 6-hourly ERA5 data used in this study, resulting in daily locations of the front. The frontal lines in Figure 1 (and following) are produced by averaging, for each longitude, all the daily latitude values for the chosen dekad across all years. This clarification has now been added at the end of Section 2.2.2.*

4. Section 3.2. I agree that the four dekads are representative for the four stages, but how the four rainfall stages were decided at the first place? Is that based on the timing of these stages suggested from previous literature? Please clarify.
*The four rainfall stages are described in detail in the Introduction. The choice of these four representative dekads is indeed based on features and timings suggested by previous literature. This has been clarified in the text.*

5. Section 4.1. Could the authors elaborate on the reasoning of analyzing the vertical cross section along A-B? I would expect to see more evident signal from the extratropical flow, but they are absent in Figure 8. Supporting evidence is missing for the statement such as in Line 345 that "The advection of cool sub-tropical air is centred around 700-800 hPa"; in Line 364 that "This is in agreement with Figure 8c, displaying these different air masses impacting on the front from opposite sides." I agree that the theta_e contour clearly shows the warm versus cold air, but the wind pattern in the extratropics is too weak to support cold air advection. Do the results change if the authors analyze the cross-section along eastern China, say 105E, instead?
*Transect AB lies parallel to the direction of EASM progression and the low-level warm southerly flow, crossing the Yangtze river and the EASMF over eastern China and extending north near the axis of the midlatitude climatological trough. This has now been clarified in the manuscript, at the beginning of section 4.1. The core of the cool flow on the northern side of the EASMF is indeed indicated by a minimum in $\theta_e$ and by negligible along-section wind at around 700-800 hPa. This is consistent with this airstream flowing mainly from northwestern quadrants (as shown in greater detail by Lagrangian analysis in Section 4.2), perpendicular to the orientation of transect AB (this clarification has also been added to Section 4.1). In section 4.2, the expression "impacting on the front from opposite sides" has been replaced by "converging at the front". See below a cross-section along western China (we presume this is what the reviewer means, given the 105°E indication). This section, shifted 10° to the west with respect to the original transect AB, passes over a more complex terrain. As a result, the low-level moist advection is less evident, as is the $\theta_e$ gradient associated with the EASMF, while the more-elevated cool flow is less affected by the transect change. In general, we do not think that using this alternative transect choice would provide added value to the manuscript, although we acknowledge the importance of checking it.*

[Figure]

Figure A1: Cross-section, on a transect shifted 10° to the west with respect to transect AB in Figures 4-9 of the manuscript, of dekadal mean moisture flux (shading, m s−1), equivalent potential temperature (red contours, every 5 K), wind speed (green contours, m s−1), wind vectors (arrows, m s−1, computed using the horizontal wind parallel to the section as horizontal component and vertical velocity as vertical component, multiplied by 200 to be consistent with the aspect ratio), potential vorticity (blue contour, 2 PVU) and precipitation (cyan bars, mm day−1). The magenta 'x' indicates the location of the EASMF at 850 hPa according to the detection algorithm. The dekad chosen (3rd dk June) represents the Mei Yu stage of EASM evolution. Data from ERA5, 1979-2018 climatology.

6. Lines 347-349 and Figure 9. Should we expect the position of the frontal position for each stage in Figure 9 match those shown in Figure 3? The spring, in particular, does not seem so. The spring front in Figure 9 is evidently further south compared to Figure 3.

*This was a mistake; we apologise for it and we thank you for spotting it. Frontal positions in Figure 9 must indeed match those in Figure 3. This has been now rectified while generating a new version of Figure 9 that contains the updated frontal latitudes (see replies to major comment 2). As specified in the text, for each stage the selection domain of the trajectories is centred around the mean frontal latitude between 115°E and 120°E, with values (rounded to the nearest quarter of a degree) of 29.5◦N, 30.75◦N, 32.75◦N, and35.75◦N for spring, pre-Mei Yu, Mei Yu, and mid-summer stages, respectively. A visual comparison can confirm that these values now match the frontal lines displayed in Figure 3.*

7. Section 5. The comparison between high-lat and low-lat cases is interesting, but the wording here is misleading. The authors selected the high-lat and low-lat cases based on the front latitude during the climatological Mei-Yu period, i.e. the 3rd dekad of June. However, it is very likely that the seasonal evolution of the EASM front is different between the high-lat and low-lat cases (Chiang et al., 2017). Therefore, when describing the high-lat and low-lat results, it might not be accurate to directly refer that as the Mei-Yu in the high-lat and the low-lat. Specific comments related to section 5 are listed below.

*We replaced "Mei Yu" with "climatological Mei Yu" where applicable, in Section 5 and in the associated paragraph of the conclusions (and in the locations pointed out in the sub-comment below).*

1) Line 393. Suggest rephrasing that these dekads refer to the climatological rainfall stages. Similarly, in Line 401, it might be more accurate to say "during the climatological Mei Yu period" or simply say "The precipitation patterns during the 3rd dekad of June." And in Line 408,

suggest replacing "during the Mei Yu stage" with "during the climatological Mei Yu stage" or "during the 3rd dekad of June."

*We agree that these suggestions improve precision, and have thus added "climatological" in each of the suggested cases. (Note that we assume the reviewer meant Line 418 rather than 408, since we could find no phrase there as quoted.)*

2) Lines 396-400. A bit concerned about the statement on whether Figure 11 tells us the dependence between the rainfall stages from both cases. An alternative interpretation could be that, the fact that the blue dashed line and the red dashed line stay at the similar latitudes suggests a more stationary front from the low-lat cases; in other words, the low-lat cases reflect years with longer Mei Yu (Chiang et al., 2017).

*We thank the reviewer for the suggestion. We do not think that this alternative interpretation, pertaining to the similar latitudes of the low-lat years for 1st and 3rd dekads of June (red and blue dashed lines) over land points, conflicts with the statements in the original manuscript. Therefore, a sentence has been added to the revised manuscript with reference to Chiang et al. (2017), while original statements have been clarified. Note that the figure referred to is now Figure 13.*

3) To further elaborate the point raised in Lines 398-400. It would be helpful to present the high-lat and low-lat versions of Figure 2 (the precipitation hovmoller diagram) in the paper. This will more clearly show the seasonal evolution of the EASM front from the two cases. The authors could also consider expanding their analysis on the STWJ, WPSH, and air mass origins to the four rainfall stages from both cases. This may provide a more complete picture on role of the tropical vs. extratropical flows on the progression of EASM front.

*Thanks for these suggestions. Figure A2, included here, contains the (a) high-lat years and (b) low-lat years Hovmöller plots for precipitation and frontal latitude. Containing only ten seasons for each panel, these plots are noisier than their all-years counterpart (Figure 2 in the original manuscript, now Figure 3). The key differences between the two plots pertain to the climatological Mei Yu period. There is northward progression in June in high-lat years, while it is delayed to July in low-lat years. Substantially larger precipitation values are also visible in low-lat years. These points have been added to Section 5, in the paragraph describing high-lat vs low-lat rainfall, and we decided not to add Figure A2 to the manuscript, which in its updated version contains 18 figures. Regarding a possible expansion of the Section 5 analysis to the other stages of EASM progression, we showed in Section 4 (and particularly in Figure 11) how Mei Yu is the stage in which tropical – extratropical interactions are most evident. Therefore, we think that, while extending the analysis in Section 5 to the other three stages would certainly be interesting, it would be beyond the scope of the present study.*

[Figure]

Figure A2: Hovmöller plot of climatological precipitation (mm day−1) and daily latitude of the EASMF at 850hPa, both averaged between 110°E - 120°E. As in Figure 3, but for (a) high-lat years and (b) low-lat years, as defined in Section 5 of the manuscript. Data from ERA5, 1979-2018 climatology.

4) Line 410. Again, it is misleading to directly refer the pattern to the 'tripole', because the latter is captured by the leading mode of East Asian summer rainfall. It is more appropriate to say that the rainfall pattern resemble one phase of the tripole mode.

*Thanks (see also our response to minor comment 6). We have adjusted the sentence to state that the pattern resembles one phase of the tripole mode, as suggested, and added a reference to Hsu and Lin (2007). We have also removed a further mention of the "tripole mode", originally contained in the description of the vertical cross-sections (now Figure 16).*

5) Line 413. To clarify, it is more precise to say that the termination of the Mei Yu has already taken place by the end of June.

*We have reworded this sentence as suggested.*

6) Figures 13-15 suggest that high-lat cases are associated with stronger moisture transport from southerly flow, while the front in the high-lat cases presents a more northward progression and produces less intense rainfall. My take from these results is that the comparison between high-lat and low-lat highlight role of the extratropical air mass in affecting both front latitude (with strong extratropical air **-->** southward or more stationary Mei Yu front) and rainfall intensity (with strong extratropical air **-->** stronger convergence with the southerlies and thus more intense rainfall). The reason why I raise this is I think this is a good example to illustrate the authors original motivation: to figure out how the tropical and extratropical air control the EASM front movement. Further, I would interpret Figure 15 in a different way, compared to what the authors stated in Line 460-462. A more generalized lesson from the comparison between high-lat and low-lat is that it is the strength of the extratropical northerlies that determine the evolution of the front. Thus, compared to the tropical component, the extratropical component is more fundamental (or dominant) in the seasonal evolution, and when the extratropical component is weak, we see the high-lat patterns. No matter whether the authors agree or disagree with these points, I think it would be worth elaborating on what we learn about role of tropical vs. extratropical air from the comparison between the high-lat and the low-lat cases.

*This is a very interesting take on the results of Section 5. We agree on extratropical air having a key role in the EASM front progression, as stated in various places in the section, as well as in Section 4 and in the conclusion. However, we do not think that there is enough evidence from these results to support the idea of extratropical air being dominant / more fundamental than its tropical counterpart in the seasonal evolution. Rather, these results suggest a synergy between tropical low-level northward advection, midlatitude circulation and associated cool descending airstream and the location and strength of the WNPSH, as summarised in the Conclusions. The assessment of the relative importance of these mechanisms in controlling EASM progression would certainly be interesting but is beyond the scope of this paper.*

8. The authors argue for a crucial role of the South Asian Monsoon in influencing the EASM but did not provide specific references and underlying mechanisms. For example, for the introduction of the tropical influence on the EASM (Line 33 and Lines 41-49), could the authors elaborate more on the effect of the onset and progression of the South Asian monsoon? I find this part is vague and unsure through what physical processes the South Asian monsoon affects the EASM. Further, it is not convincing that the results presented in the current paper suggest the Bay of Bengal, instead of South China Sea, is the primary moisture source for eastern China. The moisture content of the southerly flow is relatively constant might be due to that air parcels both precipitates along the trajectory and picks up moisture from the warm moist ocean surface over the South China Sea. In this case, there is indeed contribution from the South China Sea, it is just the total amount of water content keep unchanged.

*Many thanks for pointing this out. We agree on a lack of clarity in the original manuscript on both points raised in this comment. We have changed line 33, removing the causal link and limiting the sentence to stating the contemporaneity between the onset of the South Asian Monsoon and the pre-Mei Yu EASM stage. We clarified the sentences linking the EASM with the progression from the tropics of the general Asian monsoon, removing possibly misleading mentions of South Asian Monsoon and Indian Ocean and adding a reference to Wang and LinHo (2002). Throughout the manuscript we replaced "South Asian Monsoon" with a more general "Asian Monsoon", where appropriate. In addition to this, we replaced "Indian Ocean" with "Equatorial Indian Ocean" as the origin area of Mei Yu warm trajectories in Figure 11c (see Section 4.2). Describing the trajectories' time series in Figure 12, we now highlight the role of the South Asian monsoon in providing the "initial" (rather than "main") moisture to the airstream. Contextually, we note that the moisture content of the airstream is fairly constant as it passes over the South China Sea, implying a balance between moisture intake and precipitation. Similar edits have been made to the Conclusions.*

**Technical corrections/minor comments**

1. End of Line 29. Suggest the authors replace "a large elevated landmass" with "TP", which is more precise and concise in this context.
*Expression replaced.*

2. End of Line 32. Add references about the spring rainfall stage, for example, (Wu et al., 2007) and maybe others.
*Thanks. We added this reference and included their statement on the spring stage being a result of the TP effect on winter-spring regional circulation, rather than a proper summer-monsoon stage.*

3. Line 38. Add reference on using the gradients of equivalent potential temperature to denote the front, for example, (Tomita et al., 2011). Also clarify whether and how the equivalent potential temperature is calculated, or whether it is directly downloaded from the ERA5 website.
*This reference, among others, is now included in the revised Section 2.2.1. The description of the front detection algorithm in Section 2.2.2 clarifies that equivalent potential temperature is computed by us, using ERA5 temperature and specific humidity at 850 hPa as input data. An acknowledgement to Peter Clark for providing the Python function used for the calculation has now been added.*

4. Lines 41-49. Convection over western tropical Pacific and related "Pacific-Japan" pattern should not be ignored when considering tropical influence.
*Mention added, with reference to Nitta (1987).*

5. Line 66. This sentence reads odd. Suggest the authors rephrase the second half of the sentence to something like "…, that the EASM rapidly weakens and mid-summer rainfall dominates northeastern China."
*Thanks for this suggestion to improve clarity; it has been implemented.*

6. Lines 73-75. The mentioning of the "tripole mode" here is a bit misleading. I think it is worth citing the original reference of the tripole mode (Hsu and Lin, 2007) and clarify that the tripole mode was originally referred to the leading mode of East Asian summer rainfall at interannual timescale. Further, it is worth clarifying that Chiang et al. (2017) showed that rainfall anomalies during years with earlier Mei-Yu termination mimic one phase (i.e. more rainfall in northeastern China and southeastern China while less rainfall in central eastern China) of the tripole mode, not results in the tripole mode.

*Thanks for this comment. We have altered the sentence to suggest that there is an appearance of a "tripole pattern" in rainfall, and the citation to Hsu and Lin (2007) has been added to indicate that the original depiction of a tripole mode was in reference to interannual variability.*

7. Lines 100-109. Could the authors rephrase or reorganize this part? It would make it easier for the readers to follow if the authors can explicitly explain what each section of sections 3-5 focuses on.
*Thanks for the suggestion, the lines mentioned have been rewritten to avoid repetition.*

8. Line 113. Suggest changing the resolution to 0.25° × 0.25°.
*Changed.*

9. Section 2.1. The paper by Tarek et al. (2020) specifically focused on its performance in North America. More discussion is needed on whether ERA5 dataset is a more useful or suitable source (compared to ERA-Interim) for studying the EASM.
*We replaced that reference with two studies that are more relevant to our work, evaluating the performances of ERA5 globally and over China, respectively. As explained in the manuscript, these new references highlight the improved performance of ERA5 precipitation data compared to ERA-Interim over the tropics and its good agreement with low-altitude observation stations over China, making it suitable for studying the EASM.*

10. Line 125. Clarify what does "n" refer to. And it should be "The algorithm used in Li et al. retains …"?
*As explained in response to previous comments, the section in which Line 125 was included has been completely rewritten.*

11. Section 2.4. Add units for IVT, IWV and the variables used for both calculations.
*Units of IVT, IWV, and of all variables involved in the calculations have been added to the text in Section 2.4.*

12. Line 202. The word "extending" is misleading, which may imply that there is southward rainfall migration when the pre-Meiyu stage starts. However, the rainfall appearing in the South China Sea is just a reflection of enhanced convection during that period.
*Thanks for this suggestion to improve precision. We have replaced "rainfall extending equatorward" with "rainfall also occurring".*

13. The unit in Figure 3 caption should be mm/day?
*We agree and we changed it to mm/day, thanks for spotting it.*

14. Lines 296-297. More specifically and directly, it is the extratropical northerlies that prevent the southerly flow to march northward, and the strong extratropical northerlies have been argued to arise from the mechanical forcing from the TP on the SWTJ (Chen and Bordoni, 2014; Kong and Chiang, 2019).
*Thanks for this clarification of the mechanism. We have revised the sentence in Section 3.5 to reflect this argument, as well as simplifying some of the grammar. The sentence now reads, "In fact, high-$\theta_e$ air, associated with the aforementioned moist southerly flow (caused by joint forcing from the Asian monsoon and the WNPSH), is transported towards the EASMF. There, the northward march of the southerly flow is prevented by extratropical mid-level northwesterlies, caused by mechanical forcing of the STWJ by the TP (Chen and Bordoni, 2014; Kong and Chiang, 2020)."*

15. Line 342. Rephrase "opposite sides" to "the tropical and the extratropical side." The original wording is vague.

*Sentence rephrased.*

16. Line 466. Suggest adding "(the green contour in Figure 13)" at the end of the sentence.
*Thanks for this suggestion. The containing paragraph (at the end of Section 5) has been thoroughly revised; it now includes a cross-reference back to the relevant map, which is now Figure 15a.*

17. Suggest adding a sub-plot showing low-lat minus high-lat in Figure 14. This will help to highlight the circulation difference in the extratropics.
*Thanks for this suggestion. The high-lat - low-lat differences between the various elements of circulation and thermodynamics/moisture content (jet stream, winds and $\theta_e$ at 850 hPa, IVW) are shown in the panels of Figure 15 (Figure 13 in the original manuscript). The circulation difference at low-levels, resulting in southwesterlies between northern China and Japan, is clearly shown in the panels. Therefore, we believe that it would be best to keep Figure 16 (Figure 14 in the original manuscript) as a 2-panel figure, focusing the attention on the individual features of the vertical cross-sections in high-lat and low-lat years.*

18. Lines 503-510. I did not understand why the authors tried to draw analogies between the Indian summer monsoon and the EASM. Was the motivation to show that it is valid to apply the method, which was applied to the Indian Summer Monsoon in (Parker et al., 2016), to the EASM? If so, it might be more appropriate to put this part to Line 100.
*We approached this problem from the perspective that part of our group had earlier been involved in the Parker et al. (2016) study, and therefore we attempted to adapt and test its hypothesis of interaction between the moist tropical and dry midlatitude airmasses in monsoon onset – in this case for the EASM. Naturally the ISM and EASM have a different relationship to the major regional orography, and the EASM is further complicated by proximity to the western north Pacific high and of course the Mei Yu front.*

*As such, Parker et al. (2016) was already introduced in the Introduction (at line 89 in the original manuscript). There, we have given an example of a property of the EASM that required adapting the original Parker-like approach (i.e., its multi-stage nature) and further clarified that our focus is on outlining the key dynamics driving the EASM, clarifying the interactions between the air masses involved. Here, in the final conclusions, we return to the Indian-related hypothesis and explain the validity of having taken this approach and the fundamental differences between the EASM and the monsoon in India. We clarify again that such an approach implies a perspective focused on the air masses, tropical and extratropical, that converge at the monsoon front.*

References
Chen, J., and Bordoni, S. (2014). Orographic Effects of the Tibetan Plateau on the East Asian Summer Monsoon: An Energetic Perspective. J. Climate 27, 3052–3072.
Chiang, J.C.H., Swenson, L.M., and Kong, W. (2017). Role of seasonal transitions and the westerlies in the interannual variability of the East Asian summer monsoon precipitation. Geophys. Res. Lett. 44, 2017GL072739.
Chiang, J.C.H., Kong, W., Wu, C.H., and Battisti, D.S. (2020). Origins of East Asian Summer Monsoon Seasonality. Journal of Climate 33, 7945–7965.
Dai, L., Cheng, T.F., and Lu, M. (2021). Define East Asian Monsoon Annual Cycle via a SelfOrganizing Map-Based Approach. Geophysical Research Letters 48, e2020GL089542.
Day, J.A., Fung, I., and Liu, W. (2018). Changing character of rainfall in eastern China, 1951–2007. PNAS 201715386.
Hsu, H.-H., and Lin, S.-M. (2007). Asymmetry of the Tripole Rainfall Pattern during the East Asian Summer. J. Climate 20, 4443–4458.

Kong, W., and Chiang, J.C.H. (2019). Interaction of the Westerlies with the Tibetan Plateau in Determining the Mei-Yu Termination. J. Climate 33, 339–363.

Li, Y., Deng, Y., Yang, S., and Zhang, H. (2018). Multi-scale temporospatial variability of the East Asian Meiyu-Baiu fronts: characterization with a suite of new objective indices. Clim Dyn 51, 1659–1670.

Li, Y., Deng, Y., Yang, S., Zhang, H., Ming, Y., and Shen, Z. (2019). Multi-scale temporal-spatial variability of the East Asian summer monsoon frontal system: observation versus its representation in the GFDL HiRAM. Clim Dyn 52, 6787–6798.

Parker, D.J., Willetts, P., Birch, C., Turner, A.G., Marsham, J.H., Taylor, C.M., Kolusu, S., and Martin, G.M. (2016). The interaction of moist convection and mid-level dry air in the advance of the onset of the Indian monsoon. Quarterly Journal of the Royal Meteorological Society 142, 2256–2272.

Sampe, T., and Xie, S.-P. (2010). Large-Scale Dynamics of the Meiyu-Baiu Rainband: Environmental Forcing by the Westerly Jet*. Journal of Climate 23, 113–134.

Tarek, M., Brissette, F.P., and Arsenault, R. (2020). Evaluation of the ERA5 reanalysis as a potential reference dataset for hydrological modelling over North America. Hydrology and Earth System Sciences 24, 2527–2544.

Tomita, T., Yamaura, T., and Hashimoto, T. (2011). Interannual Variability of the Baiu Season near Japan Evaluated from the Equivalent Potential Temperature. Journal of the Meteorological Society of Japan. Ser. II 89, 517–537.

Wu, G., Liu, Y., Zhang, Q., Duan, A., Wang, T., Wan, R., Liu, X., Li, W., Wang, Z., and Liang, X. (2007). The Influence of Mechanical and Thermal Forcing by the Tibetan Plateau on Asian Climate. J. Hydrometeor 8, 770–789.

**Reviewer 2**

**General Comments**

The paper examines the structure and evolution of EASM front and the associated synoptic features. The paper is mostly well written and makes a very worthwhile contribution to the topic. However, in the Introduction the paper discusses the recent work by Parker et al. on the progression of the Indian monsoon and says it will adapt their approach to the EASM to study how competing topical and mid-latitude airmass control the progression of the EASM. I don't think the paper really achieves this. It doesn't tackle the effects of the airmasses in controlling the way in which shallow convection deepens etc. I was expecting more along these lines. Nonetheless, I found the paper as it stands insightful and original.

*Many thanks for this careful review and for the positive judgement on our work. See below our replies to the specific comments.*

**Significant Comments**

1) Section 2.1. Using the rainfall from reanalyses can be a problem as the rainfall is the model-generated rainfall. Have you compared the ERA5 rainfall to GPCP (or something similar)? You say that the patterns of ERA5 rainfall are similar to Kong et al. 2017, but what about the cross-sections plotted in Figs. 8 and 14, and the arguments made about the tri-pole structure?

*A study comparing ERA5 and ERA-Interim against GPCP dataset, and showing significant improvements in the tropics, is now referenced in Section 2.1. There are no equivalent precipitation cross-sections in Kong et al. (2017) and, therefore, no direct comparison can be made. However, the similarity between the maps in our manuscript and in their study would suggest that cross-sections would also be similar, given that the section transect is included in the domain of the maps. The arguments about the tripole structure in Section 5 have now been revised (see replies to major comment 7 by Reviewer 1 and related sub-comments), and a reference to Chiang et al. (2017) has been added.*

2) Section 2.2. The EASM front is defined by the meridional gradient in the equivalent potential temperature. But later in the paper, you relate the front to the upper jet through thermal wind. Of course, the thermal wind relates horizontal gradients in the potential temperature (not the equivalent potential temperature) to the wind shear. So, what are the implications of defining the front by equivalent potential temperature rather than potential temperature? How much of the frontal gradient comes from gradients in the moisture and how much from gradients in the temperature?

*Thanks for this comment. We agree that references to thermal wind balance would be difficult to verify, without including dry potential temperature in the analysis, and could be misleading. Therefore, we removed those references. The relative role of moisture and temperature gradients in the EASMF is mentioned in Section 2.2.1, where background research on frontal identification methods is presented. In particular, Li et al. (2018) point out that the EASMF forms a boundary between tropical and extratropical air masses, with markedly different average moisture contents, leading to strong moisture gradients on top of weaker temperature gradients. Expanding on this, we show here (Figure A3) vertical cross-sections that are equivalent to those in Figure 10 of the manuscript, but with dry potential temperature ($\theta$) instead of equivalent potential temperature ($\theta_e$). Dry isentropes are sloped, with cold air to the north and warm air to the south. However, the complex vertical $\theta_e$ structure is totally absent, as are the clear signatures of the cores of the two opposing flows. This confirms the choice of using $\theta_e$, and not $\theta$, to describe the dynamics of the*

*airflows converging at the EASMF (and to remove the references to thermal wind, as it is directly related to θ and not to θ$_e$).*

[Figure]

Figure A3:  Cross-sections, transect AB in Figures 4-9 of the manuscript, of dekadal mean moisture flux (shading, m s−1), dry potential temperature (red contours, every 2 K), wind speed (green contours, m s−1), wind vectors (arrows, m s−1, computed using the horizontal wind parallel to the section as horizontal component and vertical velocity as vertical component, multiplied by 200 to be consistent with the aspect ratio), potential vorticity (blue contour, 2 PVU) and precipitation (cyan bars, mm day−1). The magenta 'x' indicates the location of the EASMF at 850 hPa according to the detection algorithm. The dekads representing the four stages of EASM evolution (1st dk May, 1st dk June, 3rd dk June, 2nd dk July, respectively) are selected according to Figure 3. Data from ERA5, 1979-2018 climatology.

3) Section 2.2. The front detection methods is based on the meridional gradient in the equivalent potential temperature. This builds in the assumption that the front is oriented zonally. How true is this? Are there any implications for your study? That the fronts are not zonal shows up in Fig. 7 (as you note).
*This point has now been addressed, as the updated front detection methods is based on the full horizontal gradient of equivalent potential temperature. For a more detailed reply see comment 2 from Reviewer 1 and the additional comment from Prof Schultz.*

4) Section 3.1, Fig. 1. You discuss the position of the front with time, but how does the strength of the front and the slope change with time? You don't explicitly say this, but Fig. 1 shows the EASM to be a warm front.
*As correctly pointed out in this comment, we describe the northward progression with time of the EASMF in Section 3.1. We have now specified in that section that the EASMF is indeed a warm front. Its strength is mainly discussed, in terms of θ$_e$ gradient, in Sections 3.5 and in 4.1. In the latter section, changes to the slope of θ$_e$ contours are also discussed. We considered including an analysis of frontal slope as defined in Papritz and Spengler (2015), but in the end decided not to. A more detailed discussion on this matter can be found in the reply to comment 10.*

5) Figure 2. How reliable is ERA5 for precipitation? How much would this figure change if you used CPCP or something similar? I suspect it would change quite a lot. Also, there are relatively regular bursts in the rainfall prior to June with periods of around 10 days. What are these? I find is surprising that monsoon burst show up in the composite mean.

*As mentioned in the Answer to comment 1, a study comparing ERA5 and ERA-Interim against the GPCP dataset, and showing significant improvements in the tropics, is now referenced in Section 2.1. Those results, together with the comparison against Kong et al. (2017), which used rain gauge data, (see Section 3.2) suggest that the overall picture would be similar, with agreement on the key features of EASM progression, such as the characteristic EASM rainfall stages. However, we would expect to see differences at smaller scales, given the much coarser resolution of GPCP with respect to ERA5. For what concerns the 'burst', these regular and intermittent periods of precipitation above 9 mm/day in Spring and pre-Mei Yu (we assume this is what the reviewer refers to) indicate a non-constant nature of rainfall in these stages. The overall pattern of precipitation is composed by individual events with variable intensity, smoothed out by the averaging process over the 40 years considered. This is consistent with the noisier pattern seen in Figure A.2, where the diagrams only contain 10 seasons.*

6) Line 230-231. "… a feature that has been shown to be associated with the EASM progression …". Associated in what way? Be more explicit.

*Further detail from the Yihui and Chan (2005) study has been added to the introduction, and referenced in Section 3.3, to make this argument explicit. The WNPSH relates to EASM in several aspects: (i) The spring rains coincide with the first northward jump of the WNPSH over the Pacific; (ii) The onset of the Mei Yu phase coincides with further northward advance of the WNPSH and (iii) Westward extension of the WNPSH helps bring about the end of the Mei Yu phase by replacing the south-westerly flow with that from the southeast.*

7) Lines 240-241. You talk about a climatological trough over the Korean Peninsula, but the PV = 2 contour is displaced far poleward in panels c and d. Explain what's going on here.

*Thanks for pointing out the lack of clarity here. A climatological trough is present over northeast Asia, gradually weakening as the season progresses. This trough, albeit weaker, is still visible during Mei Yu. Its axis, indicated by the cyclonic turn in geopotential height contours, lies between eastern Russia and Japan (even though the 2 PVU line is further north). This has been now clarified in the text.*

8) Lines 243-251. Presumably the EASM front moves because the dilatation axis moves. And the dilatation axis is determined by the strength and location of the WNPSH, among other things. I'd like to see a plot of the deformation and dilatation axes as it would link the EASM to the theory for midlatitude fronts.

*Thanks for this suggestion. We added an analysis of front deformation, starting from the theory described in Hoskins and James (2014) in Section 3.3. This analysis highlights the importance of WNPSH strength and location in determining front deformation, as correctly hypothesised in this comment, and favouring frontogenetic flow in the EASMF region. The axis of dilation is very close to zonal during the whole season and we decided not to show it in Figure 6 to avoid adding more complication.*

9) Lines 255-262. Why not simply calculate the thermal wind assess quantitatively how well it holds?

*As explained in the reply to comment 2, we have now removed the references to thermal wind.*

10) Last paragraph of Section 3.3. As the front weakens does it slope more? The frontal slope diagnostics of Papritz and Spengler 2015 might be useful here.

*Thanks for this suggestion. Changes to the slope of θ$_e$ contours are discussed in Section 4.1. However, the frontal slope diagnostic (S) developed in Papritz and Spengler (2015) is based, correctly, on θ. In particular, S is defined as the magnitude of the horizontal gradient of geopotential height, taken along (dry) isentropic surfaces. We considered including an analysis of S in Section 3 of the manuscript but in the end decided not to, for both technical and scientific reasons. As all the ERA5 atmospheric data used in this study is on pressure levels, a conversion from isobaric to isentropic or iso-height coordinates would have been needed. In case of negligible slope of isobars (compared to isentropes), an approximated version of S defined on pressure levels could have arguably been used. In any case, the main reason behind our decision not to include an analysis of frontal slope is that the evolution of EASMF is much better described by θ$_e$ than by θ. Comparing the vertical cross-sections in Figure A3 with those in Figure 10 of the manuscript, we see that the complex vertical structure of θ$_e$ is absent for θ, as are the clear signatures of the cores of the two opposing flows. Dry isentropes are sloped, with cold air to the north and warm air to the south, but the slope is more uniform throughout the individual sections and sees little change over time, apart from a gradual slope relaxation at southern latitudes. Therefore, we decided not to include S in the new version of Section 3 and to focus on other quantities, such as front deformation and horizontal wind divergence. We would argue that they can be more useful in identifying the properties of the EASMF and the dynamics driving its evolution.*

11) Line 267. Is there really causality here? The thermal wind relation is diagnostic.
*As explained in the reply to comment 2, we have now removed the references to thermal wind.*

12) Line 269-270. The front is also in the left exit of the upstream jet, which is even more favourable for precipitation.
*Thanks for pointing this out. Starting from this comment, we decided to add an analysis of horizontal divergence at 200 hPa to Section 3.4. This analysis emphasises "the presence of widespread upper-level divergence close to the EASMF throughout the stages of EASM progression, and its intensification and organisation in a front-collocated band during Mei Yu". In particular, Figure 8 shows clear divergence in the right entrance region of the downstream jet. However, an equivalently clear signal cannot be identified in the exit region of the upstream jet, that we, therefore, do not mention.*

13) Line 272. 1000 hPa is below the ground in many places. Isn't this a problem?
*For all variables used in this study, values are masked out at all levels that are below the ground. This information is now included in the caption of all relevant figures. Thus, the lower limit of the integral used to compute IVT is either 1000 hPa or the lowest level above the ground, whichever value is the smallest (the same applies to the computation of IWV).This has now been clarified in Section 2.4 of the manuscript.*

14) Section 4.2. You start the warm and cold back-trajectories at different heights (900 and 700 mb). Hence much of the difference in the thermodynamic properties of the parcels at the initial time is due to the difference in elevation as opposed to horizontal differences in the properties of the air masses. Why not begin the parcels at a single height, say 850 mb, which is of course the height at which you define the front? How sensitive are you results to this choice?
*Warm and cold trajectories start at different heights not because of an a-priori choice, but as a result of the different elevations of cores of the respective airstreams. As explained in Section 4.2, starting points are selected by identifying the 50 lowest and the 50 highest θ$_e$ values in the starting domain. The vertical extent of this domain goes from 700 hPa and 900 hPa, in order to cover the cores of both airflows (the cool advection is centred around 700-800 hPa, while the warm advection occurs closer to the ground, see Figures 10 and 16). For example, considering Mei Yu trajectories computed using the full 40-year climatology (map in Figure 11c), Figure 12c indicates that warm back-trajectories*

*start at 875 hPa and 900 hPa, while most of the cold trajectories start at 700 hPa, with very few at 750 hPa (ERA5 data not available at 725 hPa). Whilst we acknowledge the reasons behind using a single starting level (e.g., 850 hPa), we would argue that considering all available levels between 700 hPa and 900 hPa enables us to better characterise the dynamics of both airstreams, the main goal of that section of the manuscript, while still restricting starting points to the lower troposphere. The use of a single starting level produces trajectories that are less separated in terms of initial thermodynamic properties, not capturing the full extent of the differences between the two airstreams, both in terms of path and properties (not shown).*

15) Line 492. The STWJ doesn't really **control** the variability - does it?
*We agree with the reviewer in that the nature of our analysis does not allow to distinguish "control" and "response". We have reformulated as follows:*
*"The results of this analysis emphasise the role of the STWJ, whose strength and location are closely associated with that of the EASMF. The STWJ shows a climatological trough over east Asia, and its southern edge and jet-streak right entrance region are collocated with the EASMF as the front progresses poleward."*

**Technical Corrections**

1) Line 70. Insert "which is a" before "consequence".
*Words added.*

2) Line 125. What is n? Presumably the number of points identified. Say so.
*As explained in response to previous comments, the section in which Line 125 was included has been completely rewritten.*

3) Line 202 and numerous other places. You refer to the Yangtze River and the valley. It would be helpful to mark this on one of the map. I (and presumably others) only know the general location of the Yangtze River and a not the specific geography.
*Thanks for pointing this out. We realised that the use of "Yangtze river valley" was too vague. Therefore, we replaced the term "valley" with "region", "course" or removed it altogether in the various places it was used. We have also added a black solid line to Figure 3a highlighting the middle and lower course of the river, mentioned in Section 3.2.*

4) Figure 3 caption. mm per what?
*Changed it to mm/day, thanks for spotting it.*

5) Figure 10. Use the same colour scheme as Fig. 9: warm = red and cool = blue. In my pdf, red is too close to brown and blue to close to green.
*The colour scheme used in the updated version of Figure 10 is now red-blue as in Figure 9.*

**Short Comment from Prof D. Schultz**

*Many thanks for this detailed comment. As it can be seen in this document, concerns on the front detection algorithm are echoed by the two reviewers. We therefore decided to re-develop the algorithm, which we describe in detail in the new Section 2.2.2, now based on the full horizontal gradient of $\theta_e$. Section 2.2.1 now contains a review of the state-of-the-art literature in the field, discussing the studies mentioned in this comment, and assessing their relevance to our work. In particular, we justify the choices of $\theta_e$ as quantity and of the magnitude of its horizontal gradient as function, in light of our focus on the role of the EASMF as separator between tropical and mid-latitude air masses. In general, while the changes made have undoubtedly improved the front detection algorithm, the key results from the original version of the manuscript still hold. See also the response to comment 2 from Reviewer 1.*

I have a primary concern about this manuscript, which is how the authors define a front and use the term "front" throughout the manuscript.

1. The authors use a definition of a front developed by Li et al. (2019). However, in neither Li et al. (2019) nor in the present manuscript is the suitability of this quantity assessed. Such a critical assessment in relation to previous definitions of fronts, in general, and the Mei-Yu front, specifically, needs to be carried out.

For example, at the most general, quite a few other studies have examined automated frontal detection methods, but that prior knowledge is not discussed in this manuscript as it would pertain to justifying the authors' choices in the present manuscript. Some examples include Hewson (1998), Berry et al. (2011), Schemm et al. (2018), Thomas and Schultz (2019a,b), and Catto and Raveh-Rubin (2019). The readers would benefit from a detailed discussion of the advantages and disadvantages of various approaches of automated frontal detection and a justification for these specific choices by the authors. Specifically, the following items need to be discussed.

2. Choice of theta-e. In atmospheric dynamics literature (e.g., Hoskins and Bretherton 1972), fronts are defined by the horizontal gradients in density (expressed through temperature changes). However, the present manuscript uses theta-e, which is a function of temperature and moisture. Thomas and Schultz (2019a,b) have discussed the implications of choosing theta-e over a temperature-based quantity (such as potential temperature). See in particular, Table 2 of Thomas and Schultz (2019b), which presents the advantages and disadvantages of using potential temperature versus theta-e. In the present manuscript, however, the authors did not justify their choice of theta-e over other thermodynamic quantities that would not be affected by moisture. Indeed, Yang et al. (2015) write, "mei-yu rainbands typically consist of a much stronger moisture gradient than temperature gradient". This statement (and others can be found in other articles, as well) is why a more clear definition of "front" is needed in this manuscript.

3. More specifically, what is the context for the choice of theta-e in terms of the Mei-Yu front? The discussion of the previous literature on Mei-Yu frontal identification is limited. Although theta-e is a useful diagnostic in some studies, in others, it is not appropriate. In fact, Chen et al. (2003) argued the following:
"As the mechanism for frontogenesis was almost unrelated to baroclinity in our mei-yu front case, traditional definitions of front and frontogenesis in terms of horizontal temperature gradient become inappropriate."
In Wang et al. (2016), because the thermodynamic boundary and the wind-field boundary were often not collocated, the authors diagnosed the wind field (through the vorticity equation) and frontogenesis field (using theta-e) separately.

That agreement on how to diagnose the Mei-Yu front is not apparent from just a small sampling of the literature raises the issue of the appropriateness of the frontal diagnostic used in the present manuscript. Thus, this previous literature raises the issue of how cleanly the airstream boundaries line up (or don't line up, as the case may be) with gradients in the thermodynamic fields. How do the authors reconcile their picture of the Mei-Yu front that is smooth and simple compared to the previous literature on this topic?

4. Choice of meridional gradient. As detailed in their Table 1b, Thomas and Schultz (2019b) showed studies have used the full gradient of temperature as a frontal diagnostic (e.g., Sanders and Hoffman 2002; Spensberger and Sprenger 2018; Thomas and Schultz 2019a,b). The choice of only using the meridional gradient rather than the full gradient is an unusual one. In neither Li et al. (2019) nor in the present manuscript is the use of only part of the full gradient discussed. Given that the Mei-Yu front may not be purely oriented in a west–east orientation on any given weather map, the authors would not be capturing the full magnitude of the front by only using the meridional gradient. This choice needs to be better justified in the manuscript.

5. In summary, by designation of the front as the meridional gradient of theta-e, the authors obtain results that are overly smooth compared to previous literature that describes the complexity of the Mei-Yu front. Therefore, statements such as those below need to be better qualified.
"The EASM front neatly separates tropical and extratropical air masses" (line 2).
"The Mei Yu stage is distinguished by an especially clear inter- action between tropical and mid-latitude air masses converging at the EASM front" (lines 11–12).

References
Berry G., C. Jakob, and M. J. Reeder, 2011b: Recent global trends in atmospheric fronts. Geophys. Res. Lett., 38, L21812.
Catto, J. L., Raveh-Rubin, S., 2019: Climatology and dynamics of the link between dry intrusions and cold fronts during winter. Part I: global climatology. Clim Dyn 53, 1873–1892.
Chen, G. T., Wang, C., & Liu, S. C. (2003). Potential Vorticity Diagnostics of a Mei-Yu Front Case, Monthly Weather Review, 131(11), 2680-2696.
Hewson, T. D., 1998: Objective fronts. Meteor. Appl., 5, 37–65.
Hoskins, B. J., and F. P. Bretherton, 1972: Atmospheric frontogenesis models: Mathematical formulation and solutions. J. Atmos. Sci., 29, 11–37.
Sanders, F., and E. G. Hoffman, 2002: A climatology of surface baroclinic zones. Wea. Forecasting, 17, 774–782.
Schemm, S., M. Sprenger, and H. Wernli, 2018: When during their life cycle are extratropical cyclones attended by fronts? Bull. Amer. Me- teor. Soc., 99, 149–165.
Spensberger, C., and M. Sprenger, 2018: Beyond cold and warm: An objective classification for maritime midlatitude fronts. Quart. J. Roy. Meteor. Soc., 144, 261–277.
Thomas, C. M., and D. M. Schultz, 2019a: Global climatologies of fronts, airmass boundaries, and airstream boundaries: Why the definition of "front" matters. Mon Wea. Rev., 147, 691–717, doi: 10.1175/MWR-D-18-0289.1.
Thomas, C. M., and D. M. Schultz, 2019b: What are the best thermodynamic quantity and function to define a front in gridded model output? Bull. Amer. Meteor. Soc., 100, 873–895, doi: 10.1175/BAMS-D-18-0137.1.
Yang, S., Gao, S. & Lu, C., 2015: Investigation of the mei-yu front using a new deformation frontogenesis function. Adv. Atmos. Sci. 32, 635–647.
Wang, C., Tai-Jen Chen, G., & Ho, K. (2016). A Diagnostic Case Study of Mei-Yu Frontal Retreat and Associated Low Development near Taiwan, Monthly Weather Review, 144(6), 2327-2349.

---

## Referee Report (RR1)

**Comments on WCD-2021-12-R1**

**Summary**

Thanks to the authors for their efforts in revising the original manuscript. The revised version has been largely improved and the authors' responses have cleared most of my concerns. Particularly, the authors provided a more thorough literature review on frontal detection methods and expanded their analysis for a more comprehensive survey of insights revealed from the frontal detection algorithm proposed in this work. My comments in this round are mainly clarifications, focusing on making sure the message is clear and direct to the readers. I recommend the paper for publication in the journal after the authors address these comments.

1. I think the authors should highlight the most novel element in the title of their manuscript. In my view, the most novel thing in this work is that they proposed a frontal detection algorithm and validated its fidelity in capturing the climatology and interannual variability of the EASM. So, I suggest changing the title to something like "Understanding the seasonal evolution and interannual variability of the East Asian summer monsoon through a novel frontal detection algorithm"

2. Line 6. Please clarify here you meant "interannual variability"

3. Line 9. I think you should highlight that Mei-Yu is the primary stage of the EASM.

4. Line 10. This sentence should be revised and clarified. What does "These forcings" mean? The authors seem to argue that the low-level southerly flow is modulated by the seasonal evolution of the South Asian Monsoon system, not in part due to WNPSH? In my view, both the South Asian origin and those along the edge of the WNPSH contribute to the southerly flow.

5. Line 13. Clarify "the midlatitude flow impacting on the northern side", in the spirit of being concise and clear, suggest changing to "extratropical northerly flow"

6. Line 17. Clarify "the regional flow"; suggest being specific and clear which regional circulation systems drive the airmasses converging at the front; also, is it true that only "the low-level airstreams" matter? The authors also showed in the paper that the northerly flow from higher levels is needed for the formation of the Mei Yu front.

7. Line 38. Provide the full name of EASMF as it is the first time it appears in the main text.

8. Line 42-44. This statement is oversimplified and might be misleading. The whole sentence is a paraphrase of the texts below Figure 4 from Wang and LinHo (2002). And if I understand correctly, Wang and LinHo (2002) is not about how the whole Asian monsoon (including the Indian Summer monsoon, East Asian summer monsoon, and the WNP summer monsoon) affects the EASM, rather, they discuss how the Asian monsoon system as a whole progress across the season. So, I don't think it is reasonable using this single

paper as a reference for arguing tropical influence on EASM. The authors mentioned other tropical influences before closing that paragraph; it is more appropriate to include those factors (ENSO, tropical convection) into the opening sentence of the paragraph. And you can elaborate how the EASM is related to other components in the Asian monsoon system after that.

9. Line 45-46. Suggest removing the redundant information and rephrase to "…, the poleward transport of moisture over the South China Sea is associated with the Western North Pacific subtropical high (WNPSH)."

10. Line 53. Provide full name of "ENSO"

11. Line 56-57. I don't disagree with the authors that diabatic heating over the orography can affect the meridional position of the STWJ, but the current description is too vague. Please provide references (if there is any) that have discussed effect of the springtime diabatic heating on the northward migration of the STWJ. Also, the seasonal cycle of insolation and the resulted changes in the meridional temperature gradient should be more instrumental in driving the northward progression of the jet from spring to summer.

12. Line 57. I think the authors meant the diabatic heating over the TP, but please be more specific and clarify what are "These processes" as the preceding sentences have covered both STWJ and the diabatic heating.

13. Line 85-86. Can the authors clarify the blocking anticyclones over which regions are of particular importance in causing the dry air intrusion that may affect EASM?

14. Line 87-89. The original papers of the Silk Road Pattern should be cited (see references in Hong et al). Suggest adding one sentence to briefly introduce what is the Silk Road Pattern, otherwise readers who are not familiar with this pattern might have a hard time appreciate the summarized studies of Hong et al.

15. Line 90-92. Did the author mention the interaction between the tropical dynamics and the TP in the introduction section?

16. Line 105-106. Suggest rephrasing to "…we focus on the role of … masses in the EASM progression"

17. Line 106 and in other related lines. Is it possible to replace "a Parker et al. (2016)-like framework" with a more physical description? The author cited Parker et al. (2016) as one motivation for understanding the dynamics of the EASM from the perspective of the interaction of tropical and extratropical air masses. But this study used very different approach or methods throughout, so I don't suggest the authors to use "Parker et al. 2016-like framework or Parker et al. 2016-like approach" to describe this motivation.

18. Section 2.2.1. Since the main purpose of this section is to introduce the previously proposed algorithms and justify the choice of using the horizontal gradient of theta_e in the EASMF detection, I think it would make more sense to first introduce the methods based on the baroclinicity/SOM/rainband summarized in Line 160-170, then introduce the thermodynamic definitions as mentioned in Line 140-159, and then continue in Line 171-187 to justify the parameter used in this work. Other minor comments in this section include
    a. Suggest using a title such as something like "Existing front detection algorithms in the literature"
    b. Line 135. Suggest rephrasing to "… have been proposed in the literature to identify …"
    c. Line 147-150. To make it more concise, suggest replacing "While this… Being dependent on moisture content, " with "As a result," because the description of Line 152-155 have covered similar information;  also suggest replacing "this front" in Line 149 with "the EASMF"
    d. Besides providing the list of references in Line 160-165, can the author add more information for each study (or each group of studies), no need to be long, simply describing XXX used XXX methods to study fronts or airmass boundaries over which region or to study which problem (something like this) would be helpful.
    e. Line 165-166. I did not get the point here. Can the authors reword?
    f. Line 168. Suggest deleting "subtropical, or even" because the focus in this study is on the interaction between the tropical and extratropical airmasses.
    g. Line 170. I think the authors also discussed higher level circulations including the STWJ and the extratropical northerly flow.

19. Line 198. Did you use a uniform 850 hPa as the threshold? Not sure whether this is a reasonable estimation because the terrain in East Asia is not that uniform, though I agree that the terrains in East China are not that high.

20. Line 203. How sensitive are the results obtained from this algorithm to the selected threshold? If this has already been discussed in previous literature, suggest the authors add that information here as well.

21. Line 214. I am curious about how these captured "other large-scale fronts" look like. I am asking because though these fronts might not look like a typical EASMF that we expect from the climatological rainband progression, but this reasoning might not be convincing enough to exclude their role in contributing to the EASM rainfall. Can the authors comment on this?

22. Another question related to the algorithm. Figure 1c shows frontal line over land along the northern front, while the southern land-based front is not considered as front though it appears to be part of the large-scale zonally coherent fronts extending from the land to the ocean. What is the reasoning for this?

23. The current shading scale in Figure 1 makes it a bit difficult to see the black contours of theta_e. I wonder whether regions where the theta_e gradient is lower than 0.03 can be blanked out.

24. Equations (3) and (4): suggest replacing "1000 hPa" with "surface"

25. Please provide references for ERA-Interim, GPCP, and APHRODITE when these datasets are mentioned.

26. For the frontal deformation analysis (Figure 6 and Lines 370-380), can the authors show the F1 and F2 separately as well, and discuss how they tell us about the front and the role of tropical and extratropical air masses?

27. Line 433. Please clarify what it means by saying these southerlies are "monsoonal"

28. Figure 10 caption, clarify it is zonal wind speed shown in the green contour

**References**

Parker, D.J., Willetts, P., Birch, C., Turner, A.G., Marsham, J.H., Taylor, C.M., Kolusu, S., and Martin, G.M. (2016). The interaction of moist convection and mid-level dry air in the advance of the onset of the Indian monsoon. Quarterly Journal of the Royal Meteorological Society *142*, 2256–2272.

Wang, B. and LinHo (2002). Rainy Season of the Asian–Pacific Summer Monsoon*. J. Climate *15*, 386–398.

---

## Author Response (AR2)

**Response to referee comments on the first revision of "The interaction of tropical and extratropical air masses controlling East Asian summer monsoon progression" by Volonté et al.**

wcd-2021-12-R1 - A. Volonté on behalf of all authors, 7 April 2022

Dear Editor,

We thank Prof Michael Reeder and an anonymous referee for their reviews and for the positive and constructive comments on our manuscript. We believe that addressing those comments has given us the chance to further improve our study. Below we give point-to-point responses to the comments by Referee #1. Our responses are written in *green italic* font, while all edits in the tracked-changes version of the manuscript are in red.

**Comments on WCD-2021-12-R1**

**Summary**

Thanks to the authors for their efforts in revising the original manuscript. The revised version has been largely improved and the authors' responses have cleared most of my concerns. Particularly, the authors provided a more thorough literature review on frontal detection methods and expanded their analysis for a more comprehensive survey of insights revealed from the frontal detection algorithm proposed in this work. My comments in this round are mainly clarifications, focusing on making sure the message is clear and direct to the readers. I recommend the paper for publication in the journal after the authors address these comments.

1. I think the authors should highlight the most novel element in the title of their manuscript. In my view, the most novel thing in this work is that they proposed a frontal detection algorithm and validated its fidelity in capturing the climatology and interannual variability of the EASM. So, I suggest changing the title to something like "Understanding the seasonal evolution and interannual variability of the East Asian summer monsoon through a novel frontal detection algorithm"

*Many thanks for this suggestion and for acknowledging the novelty in this study. We agree on including a reference to the frontal detection approach used. However, we would argue that the focus on the interaction between airmasses is a prominent feature of this paper and should be retained in the title, arguably at the expense of mentioning "seasonal evolution and interannual variability". Therefore, we changed the title to "Characterising the interaction of tropical and extratropical air masses controlling East Asian summer monsoon progression using a novel frontal-detection approach".*

2. Line 6. Please clarify here you meant "interannual variability"

*Done, thanks. The same change has been applied to the relevant sentence in the conclusions (see text highlighted in red) and the same applies to our replies to comments 3-6. Contextually, we have also replaced "EASM evolution" with "EASM progression" in the abstract and in the rest of the manuscript, where applicable.*

3. Line 9. I think you should highlight that Mei-Yu is the primary stage of the EASM.

*Done.*

4. Line 10. This sentence should be revised and clarified. What does "These forcings" mean? The authors seem to argue that the low-level southerly flow is modulated by the seasonal evolution of the South Asian Monsoon system, not in part due to WNPSH? In my view, both the South Asian origin and those along the edge of the WNPSH contribute to the southerly flow.

*The sentence has been changed to "These forcings act to steer the southerly advection of low-level moist tropical air, modulated by the seasonal cycle of the Asian monsoon." STWJ and WNPSH are mentioned in the previous sentence and are thus the forcings we are referring to. We think that the updated sentence is consistent with the results in this study, showing that both STWJ and WNPSH locations and patterns are instrumental in the path of the low-level southerly advection of tropical air, which is indeed modulated by the overall cycle of the South Asian monsoon (see for example the origin and path of "warm" trajectories in Figures 11 and 17 and the low-level wind field in many other figures).*

5. Line 13. Clarify "the midlatitude flow impacting on the northern side", in the spirit of being concise and clear, suggest changing to "extratropical northerly flow"

*We replaced "midlatitude flow" with "cool extratropical flow". We would avoid inserting the term "northerly", as our study (particularly the Lagrangian analysis) shows that this airstream has an important zonal component, particularly when travelling on the northern side of the Tibetan Plateau.*

6. Line 17. Clarify "the regional flow"; suggest being specific and clear which regional circulation systems drive the airmasses converging at the front; also, is it true that only "the low-level airstreams" matter? The authors also showed in the paper that the northerly flow from higher levels is needed for the formation of the Mei Yu front.

*Thanks for pointing this out. We added "over East Asia" after "the regional flow" (this is a summarising sentence, so we don't think we should mention again STWJ and WNPSH). We also changed "low-level airstreams" to "low- and mid-level airstreams" to include the "cool" flow.*

7. Line 38. Provide the full name of EASMF as it is the first time it appears in the main text.
*Done, thanks for spotting this.*

8. Line 42-44. This statement is oversimplified and might be misleading. The whole sentence is a paraphrase of the texts below Figure 4 from Wang and LinHo (2002). And if I understand correctly, Wang and LinHo (2002) is not about how the whole Asian monsoon (including the Indian Summer monsoon, East Asian summer monsoon, and the WNP summer monsoon) affects the EASM, rather, they discuss how the Asian monsoon system as a whole progress across the season. So, I don't think it is reasonable using this single paper as a reference for arguing tropical influence on EASM. The authors mentioned other tropical influences before closing that paragraph; it is more appropriate to include those factors (ENSO, tropical convection) into the opening sentence of the paragraph. And you can elaborate how the EASM is related to other components in the Asian monsoon system after that.

*We have clarified that the eastern component of the Asian monsoon is characterised by a multi-stage northward progression starting from tropical latitudes, sustained by low-level southerly and southwesterly winds. This makes the EASM a tropical phenomenon, at least partially. This progression is described in Wang and LinHo (2002) and thus we would argue that this reference is appropriate. We would also argue that tropical modes of variability like ENSO should not be moved to the opening sentence of this paragraph, as the Asian monsoon (including the EASM) would exist (and progress) even in the absence of ENSO.*

9. Line 45-46. Suggest removing the redundant information and rephrase to "..., the poleward transport of moisture over the South China Sea is associated with the Western

North Pacific subtropical high (WNPSH).”
*Done.*

10. Line 53. Provide full name of “ENSO”
*Done.*

11. Line 56-57. I don't disagree with the authors that diabatic heating over the orography can affect the meridional position of the STWJ, but the current description is too vague. Please provide references (if there is any) that have discussed effect of the springtime diabatic heating on the northward migration of the STWJ. Also, the seasonal cycle of insolation and the resulted changes in the meridional temperature gradient should be more instrumental in driving the northward progression of the jet from spring to summer.
*We tried improving the clarity of this discussion by referencing the work by Li and Yanai (1996) and specifying that, given the height of the TP, these diabatic processes are able to trigger a reversal of meridional temperature gradient in the upper troposphere between the TP and the Indian Ocean, therefore acting on the regional circulation.*

12. Line 57. I think the authors meant the diabatic heating over the TP, but please be more specific and clarify what are “These processes” as the preceding sentences have covered both STWJ and the diabatic heating.
*We replaced “These processes” with “These diabatic processes” (mentioned in the previous sentence).*

13. Line 85-86. Can the authors clarify the blocking anticyclones over which regions are of particular importance in causing the dry air intrusion that may affect EASM?
*Yihui and Chan (2005), referenced in the manuscript at the beginning of the sentence considered in this comment, state that “The continuous southward intrusion of cold air and accompanying frontal systems (the so-called Meiyu-Baiu front) is excited by the development and prevailing of blocking highs in the mid-and high latitudes over Eurasia. The dual blocking high situation, one located over the Ural Mountains and another located over the Okhotsk Sea, is the most favorable situation for prolonged Meiyu-Baiu heavy rainfall”. We have added a few more words at the end of the sentence, explicitly mentioning this preferred blocking pattern.*

14. Line 87-89. The original papers of the Silk Road Pattern should be cited (see references in Hong et al). Suggest adding one sentence to briefly introduce what is the Silk Road Pattern, otherwise readers who are not familiar with this pattern might have a hard time appreciate the summarized studies of Hong et al.
*Thanks for pointing this out; a reference to Lu et al. (2002) has been added. Starting from the explanation given in the conclusions of the article referenced, we also specified that the Silk Road Pattern is a wave train trapped along the North African and Eurasian jet streams.*

15. Line 90-92. Did the author mention the interaction between the tropical dynamics and the TP in the introduction section?
*We appreciate that the role of the TP is mainly discussed in relation to the STWJ. Therefore, we modified the sentence, that now reads “… the complexity of EASM progression, influenced by both tropical and midlatitude dynamics, and by the interaction with the TP”.*

16. Line 105-106. Suggest rephrasing to “…we focus on the role of … masses in the EASM progression”
*Done, thanks.*

17. Line 106 and in other related lines. Is it possible to replace "a Parker et al. (2016)-like framework" with a more physical description? The author cited Parker et al. (2016) as one motivation for understanding the dynamics of the EASM from the perspective of the interaction of tropical and extratropical air masses. But this study used very different approach or methods throughout, so I don't suggest the authors to use "Parker et al. 2016-like framework or Parker et al. 2016-like approach" to describe this motivation.

*The quote (now at line 114) now reads "a front- and airmass-centred Parker et al. (2016)-like framework", and the same addition has been made in the first paragraph of the Conclusions.*
*We would argue that there are substantial methodological similarities between Parker et al. (2016) (and Volonté et al., 2020) and this study, such as the focus on frontal progression and on the location and evolution of the 3-dimensional boundary between different air masses. Furthermore, Parker et al. (2016) used a trajectory approach to contrast the tropical and mid-latitude origin of air at different stages of the ISM onset, as in this study. We now explicitly mention this at lines 110-112. In this study, we have assessed the validity of such an approach (front- and airmass-centred) to the progression of the EASM. We copy here a couple of sentences from the conclusions that summarise this assessment. "There are fundamental differences between the Indian summer monsoon and the EASM, caused essentially by the different geographical location of the main actors of the two monsoon systems. However, analogies can be drawn. The progression of the Indian summer monsoon is modulated by the balance between moist low-level tropical advection and mid-level subtropical dry intrusion, in turn influenced by tropical modes and extratropical dynamics, respectively. For the EASM, the interaction between tropical and extratropical airstreams is particularly pronounced in the Mei Yu stage, when also most of the northward migration occurs."*

18. Section 2.2.1. Since the main purpose of this section is to introduce the previously proposed algorithms and justify the choice of using the horizontal gradient of theta_e in the EASMF detection, I think it would make more sense to first introduce the methods based on the baroclinicity/SOM/rainband summarized in Line 160-170, then introduce the thermodynamic definitions as mentioned in Line 140-159, and then continue in Line 171-187 to justify the parameter used in this work.

*We understand this point of view, but we would argue that the description of methodologies based on thermodynamic functions should be placed first, given that those are the methodologies most commonly used. To better highlight the two main choices to be made when devising a front identification method (physical quantity and mathematical function) their descriptions are now placed in two different sub-sections.*

Other minor comments in this section include:

a. Suggest using a title such as something like "Existing front detection algorithms in the literature"
*We changed the title of Section 2.2.1 to "Front-detection algorithms in literature". Contextually, we modified the title of Section 2.2.2 to "Description of the algorithm used in this study".*

b. Line 135. Suggest rephrasing to "… have been proposed in the literature to identify …"
*Done.*

c. Line 147-150. To make it more concise, suggest replacing "While this… Being dependent on moisture content, " with "As a result," because the description of Line 152-155 have covered similar information; also suggest replacing "this front" in Line 149 with "the EASMF"
*Although we did not implement the changes exactly as suggested, the modifications we have made follow the spirit of this comment. We also made a few small changes earlier in the paragraph to improve its clarity.*

d. Besides providing the list of references in Line 160-165, can the author add more information for each study (or each group of studies), no need to be long, simply describing XXX used XXX methods to study fronts or airmass boundaries over which region or to study which problem (something like this) would be helpful.

*We added a sentence at the end of this part, specifying that the breadth of different methods reflects the variety of subjects tackled by those studies, ranging from global reviews, climatologies of Mei Yu and mid-latitude fronts, to single Mei Yu case studies (and we included relevant references in the manuscript text).*

e. Line 165-166. I did not get the point here. Can the authors reword?
*Sentences reworded, thanks.*

f. Line 168. Suggest deleting "subtropical, or even" because the focus in this study is on the interaction between the tropical and extratropical airmasses.
*Done.*

g. Line 170. I think the authors also discussed higher level circulations including the STWJ and the extratropical northerly flow.
*We agree with this observation. However, in this case the sentence refers specifically to rainfall and low-level winds because those are the key quantities to the "other approaches to front identification" described in the paragraph.*

19. Line 198. Did you use a uniform 850 hPa as the threshold? Not sure whether this is a reasonable estimation because the terrain in East Asia is not that uniform, though I agree that the terrains in East China are not that high.
*We confirm that the front detection algorithm was applied at the 850 hPa pressure level, following most of the literature presented in Section 2.2.1, including studies focusing on the EASMF. In addition to being the most practical option (ERA5 data on pressure levels), we would argue that the 850 hPa level is far enough from the terrain in East China to allow a safe use of the algorithm, as opposed to (e.g.) the 925 hPa level. The topography of the region is presented in Figure 2, along with dekadal climatological frontal lines, and in Figures 11,13, 17 and 18. These figures show that only the region west of 110°E display a non-negligible terrain influence. This point is made at the end of Section 2.2.2, along with an explanation of how many missing values determine longitudes being skipped.*

20. Line 203. How sensitive are the results obtained from this algorithm to the selected threshold? If this has already been discussed in previous literature, suggest the authors add that information here as well.
*The gradient threshold chosen in this manuscript is consistent with the relevant studies referenced in Section 2.2.1. To our knowledge, the key points of discussion on frontal identification concern the quantity to be used and the mathematical function to be applied on it, rather than the actual threshold value. However, a preliminary investigation on selected case studies has shown that 0.05 K km$^{-1}$ is close to the 90$^{th}$ percentile of the horizontal gradient of equivalent potential temperature and this seems to agree with Figure 4 in Thomas and Schultz (2019a).*

21. Line 214. I am curious about how these captured "other large-scale fronts" look like. I am asking because though these fronts might not look like a typical EASMF that we expect from the climatological rainband progression, but this reasoning might not be convincing enough to exclude their role in contributing to the EASM rainfall. Can the authors comment on this?
*The statement on excluding "other large-scale fronts" is related to the choice of restricting the latitudinal domain to a 20° latitude interval centred on the mean gradient-weighted latitude of all points exceeding the gradient threshold on that day. For example, a daily mean gradient-weighted*

*latitude of 30° N would exclude all points below 20° N and above 40° N. Thus, this could exclude fronts related to tropical activity in the South China Sea and/or fronts associated with higher-latitude Rossby waves, when the EASM front is over the Yangtze river. Figure 1c provides a partial example, with above-threshold values present at around 15° N and not considered by the algorithm. It is clear that those frontal regions are not part of the EASM front and do not contribute to EASM rainfall. In the same figure, the EASM front splits into two bands over land, with the northern one selected by the algorithm. Those bands are not removed by the algorithm step just discussed and their treatment is explained in detail in the next comment.*

22. Another question related to the algorithm. Figure 1c shows frontal line over land along the northern front, while the southern land-based front is not considered as front though it appears to be part of the large-scale zonally coherent fronts extending from the land to the ocean. What is the reasoning for this?

*In Figure 1C the front, clearly oriented on a single zonal band over the ocean, splits into two bands over land. The southern "land band" lies around the same latitude of the "ocean band". However, our algorithm identifies the northern "land band", a few degrees further north, as front. The reasoning behind this is that the northern band has a higher mean $\theta_e$ gradient than the southern one and thus, as the algorithm starts drawing the line over land, this is the band that will be picked up first. If instead the algorithm were to start from the ocean, the southern land band would be chosen, as (at all points after the start) the search for bands is firstly performed in a 5° interval centred on the previous latitude identified. As explained in Section 2.2, since the existence of small-scale multiple fronts within the EASMF was found more likely over land than ocean, starting longitudes selected in this study cover the inland longitudinal extent of the region affected by EASM progression, every 3° from 108°E to 120°E. The modal value of the latitudes identified is then taken at each longitude. In summary, in cases like the one shown in Figure 1c, the algorithm will prioritise the most intense band over land, as long as this band is located within the 20° latitude interval centred on the mean gradient-weighted latitude of all points exceeding the gradient threshold on that day. This is consistent with our aim of identifying and characterising the general progression of the monsoon front. While the identification of a coherent band is a key feature of this study (and therefore we inserted the 5° interval step), we are choosing to prioritise intensity rather than minimising all latitudinal jumps, which would come at the cost of following less intense (and arguably secondary) frontal bands.*

23. The current shading scale in Figure 1 makes it a bit difficult to see the black contours of theta_e. I wonder whether regions where the theta_e gradient is lower than 0.03 can be blanked out.

*Thanks for pointing this out. After some experimenting with contours and colour maps, we decided to increase the width of equivalent potential temperature contours (now dashed) to improve their visibility. We did not blank out low-gradient regions because we wanted to keep them distinct from masked and out-of-domain areas.*

24. Equations (3) and (4): suggest replacing "1000 hPa" with "surface"

*We see the merit of this comment, given that the vertical integration starts from the surface at all grid points where surface pressure is equal to or less than 1000 hPa. However, in all other grid points the integration starts from 1000 hPa, and the choice of levels used for vertical integration is explained in the text. Therefore, we make no changes.*

25. Please provide references for ERA-Interim, GPCP, and APHRODITE when these datasets are mentioned.

*References added, thanks.*

26. For the frontal deformation analysis (Figure 6 and Lines 370-380), can the authors show the F1 and F2 separately as well, and discuss how they tell us about the front and the role of tropical and extratropical air masses?

*As mentioned in Section 2.5, F1 and F2 are the two deformation components, with an angle of 45° between each other. Given that the dilation axis is nearly zonal, F1 = ∂u/∂x − ∂v/∂y is dominant over F2. This point has now been added to the discussion.*

27. Line 433. Please clarify what it means by saying these southerlies are "monsoonal"

*This has been clarified by specifying that the air considered "retains a moisture content typical of South Asian monsoon airmasses".*

28. Figure 10 caption, clarify it is zonal wind speed shown in the green contour

*The quantity indicated by the green contours in Figure 10 is not zonal wind speed, but the magnitude of horizontal wind speed. This is now specified in the caption (and in the caption of Figure 8). We inspected the text of Section 4.1 to find possible misleading sentences on the matter but couldn't find any. Therefore, no other changes have been made.*

**References**

Parker, D.J., Willetts, P., Birch, C., Turner, A.G., Marsham, J.H., Taylor, C.M., Kolusu, S., and Martin, G.M. (2016). The interaction of moist convection and mid-level dry air in the advance of the onset of the Indian monsoon. Quarterly Journal of the Royal Meteorological Society 142, 2256–2272.

Wang, B. and LinHo (2002). Rainy Season of the Asian–Pacific Summer Monsoon*. J. Climate 15, 386–398.